# Video Generation Models: A Survey of Post-Training and Alignment

**Chaoyu Li**[*,†]                                    *chaoyuli@asu.edu*
*Arizona State University*

**Xiaoyi Gu**[*]                                      *xiaoyigu1809@gmail.com*
*Twitch*

**Yogesh Kulkarni**                                   *ykulka10@asu.edu*
*Arizona State University*

**Eun Woo Im**                                        *eunwooim@asu.edu*
*Arizona State University*

**Mohammadmahdi Honarmand**                           *mhonar@stanford.edu*
*Stanford University*

**Zeyu Wang**                                         *zwang14@ebay.com*
*eBay*

**Juntong Song**                                      *juntong.song@newsbreak.com*
*NewsBreak*

**Fei Du**                                            *dufei@microsoft.com*
*Microsoft*

**Xilin Jiang**                                       *xj2289@columbia.edu*
*Columbia University*

**Kexin Zheng**                                       *kexinzhe@usc.edu*
*University of Southern California*

**Tianzhi Li**                                        *litianzhi98@gmail.com*
*Carnegie Mellon University*

**Fei Tao**                                           *fei.tao@newsbreak.com*
*NewsBreak*

**Pooyan Fazli**[†]                                   *pooyan@asu.edu*
*Arizona State University*

[*]*Equal Contribution*    [†]*Corresponding Author*

**Github Repo:** `https://github.com/people-robots/Awesome-Video-Generation-Post-Training`
**Reviewed on OpenReview:** `https://openreview.net/forum?id=YlUEWLESIu`

## Abstract

Video generation has rapidly progressed from short, low-quality clips to high-resolution, long-duration sequences with complex spatiotemporal dynamics. Despite strong generative priors learned through large-scale pretraining, pretrained video models often fail to reliably follow human intent, maintain temporal coherence, or satisfy physical and safety constraints.

Compared with image and text generation, alignment in video generation presents unique challenges, including error accumulation over time, motion-appearance coupling, multi-objective trade-offs, and limited supervision for temporal properties. These challenges motivate systematic post-training strategies that adapt pretrained models without retraining them from scratch. In this survey, we present the first comprehensive review of post-training and alignment in video generation models. We frame post-training as a unifying framework and distinguish between **implicit alignment** and **explicit alignment** based on how alignment signals are enforced. From this perspective, we organize existing approaches into four broad categories: (1) **supervised fine-tuning methods**, (2) **self-training and distillation methods**, (3) **preference- and reward-based methods**, and (4) **inference-time methods**. This taxonomy provides a coherent view of how alignment signals shape model behavior across both training and deployment. Beyond methodological advances, we review commonly used datasets, benchmarks, and evaluation practices, and discuss open challenges such as scalable reward design, long-horizon temporal consistency, stability-expressiveness trade-offs, and safety-aware generation. This survey aims to provide a structured conceptual foundation and practical guidance for advancing controllable and reliable video generation models.

# 1 Introduction

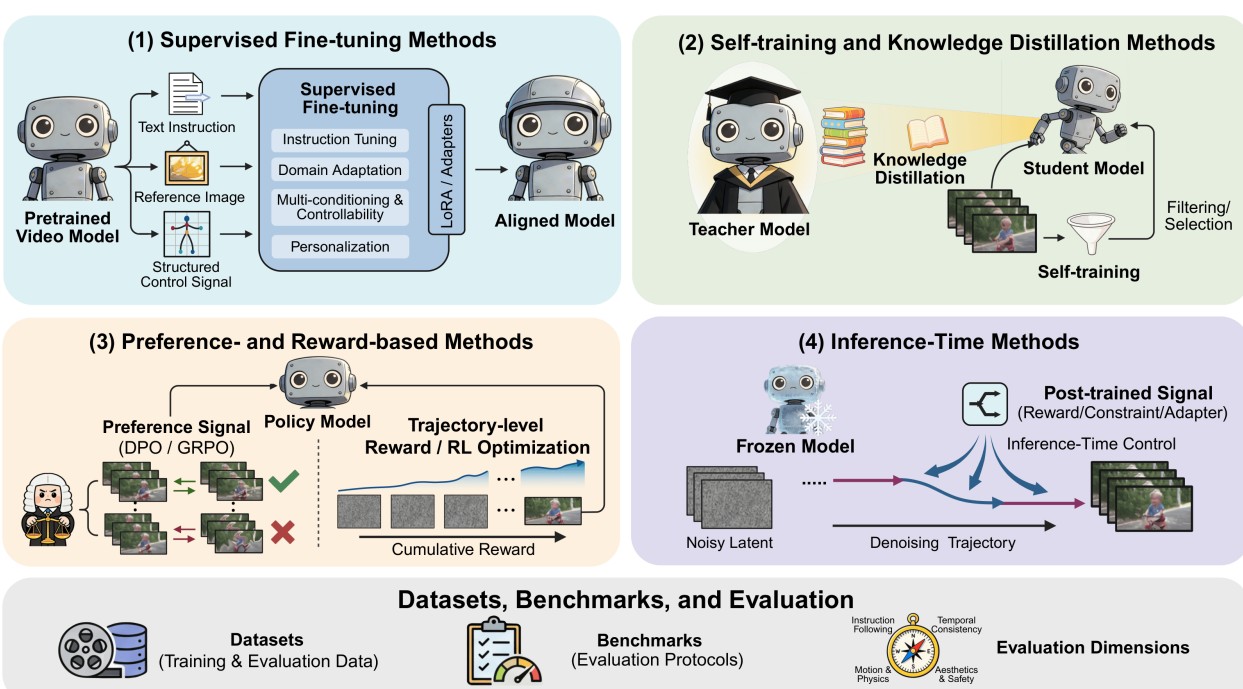

Figure 1: An overview of the post-training and alignment in video generation, and the scope of this survey.

Video generation has advanced rapidly, evolving from low-quality, short clips to high-definition, minute-long sequences with increasingly complex dynamics (Brooks et al., 2024; Zhuo et al., 2024). Despite this progress, generating realistic videos remains highly challenging. Models must preserve spatiotemporal coherence, ensure physical plausibility, and maintain fine visual details simultaneously. As a result, video generation stands among the most demanding problems in generative AI, requiring both strong generative priors and precise control mechanisms. The evolution of video generation models has followed several major trends. Early research mainly relies on generative adversarial network (GAN)-based methods and probabilistic generative models, focusing on unconditional generation or class-specific synthesis (Vondrick et al., 2016; Xue et al.,

2025; Denton & Fergus, 2018). However, these approaches often suffer from limited diversity and training instability, making it difficult to model complex video distributions (Goodfellow et al., 2020; Ho et al., 2020).

With the growth of large-scale datasets and computational resources, research shifts toward foundation-style pre-training on massive video-text corpora. This paradigm enables models to learn general visual representations and align visual content with semantic descriptions, substantially improving generalization and text-conditioned generation performance (Blattmann et al., 2023a; Guo et al., 2024b). Early diffusion-based video generation models typically adopt U-Net-style backbones Ho et al. (2022), while more recent large-scale systems have increasingly transitioned to Diffusion Transformers (DiTs) Peebles & Xie (2023), which have emerged as the dominant paradigm in modern video generation (Kong et al., 2024; Lin et al., 2024a; Jin et al., 2025; Yang et al., 2025b; Ma et al., 2025b). By combining scalable transformer backbones with diffusion-based denoising, operating in compressed latent spaces, and incorporating multimodal conditioning, DiT-based models show strong scaling behavior and impressive generalization across diverse video generation tasks. Despite the powerful generative priors obtained through large-scale pre-training, such models do not inherently guarantee that generated videos adhere to user intent, physical constraints, or fine-grained control signals. In practice, failures often appear as identity drift in long videos, physically unrealistic object interactions, unstable motion, or incomplete alignment with complex text prompts (Huang et al., 2024; Bansal et al., 2025). These problems are not just empirical weaknesses. They reflect a deeper mismatch between the objectives used during pre-training and the behavioral requirements of real-world applications (Yuan et al., 2024a; Dai et al., 2024; Zheng et al., 2025a).

From a learning perspective, post-training represents a shift in objectives. In pre-training, the model optimizes likelihood over large-scale, noisy web data to learn a broad generative distribution. While this enables general visual competence, such data rarely captures fine-grained physical constraints or consistent human aesthetic preferences. As a result, important behavioral constraints remain underrepresented. Post-training addresses this gap by introducing higher-quality and more structured supervision. These signals steer the model toward specific behavioral goals. Instead of relearning general visual concepts, post-training reshapes existing representations to better satisfy practical requirements. This transition, from broad distribution modeling to targeted behavioral refinement, forms the conceptual foundation of alignment in video generation.

Aligning video generation models with desired behaviors poses fundamentally different challenges from those in image or text generation. In videos, small frame-level errors can accumulate over time and interact in complex ways, leading to artifacts that may not be evident in short clips (Zheng et al., 2025a). Moreover, alignment objectives, such as motion realism, temporal coherence, identity consistency, and physical plausibility, span multiple dimensions and can conflict with one another, creating trade-offs between stability and expressiveness (Atzmon et al., 2025). Reliable supervision for temporal properties is also scarce and expensive, leading to reliance on proxy metrics or learned evaluators that may introduce bias (Qian et al., 2025). Together, these challenges call for alignment strategies specifically designed to handle temporal dynamics, multi-objective trade-offs, and limited supervision (Liu et al., 2025c; Zhang et al., 2026a).

Motivated by these challenges, recent research has increasingly focused on post-training and alignment techniques that adapt pretrained models through additional optimization stages applied after large-scale pretraining. Figure 1 provides an overview of this landscape, highlighting major post-training paradigms and representative methods discussed in this survey. Rather than modifying core architectures or relying solely on scaling, these approaches refine model behavior through targeted post-training optimization (Wu et al., 2023; Liu et al., 2025h). Across the literature, post-training techniques have expanded along multiple dimensions. As shown in Figure 2, research on post-training alignment for video generation has grown rapidly since 2022, with expanding diversity in supervision paradigms and deployment strategies.

One line of work focuses on supervised adaptation and parameter-efficient tuning, such as LoRA (Hu et al., 2022), to improve controllability, personalization, and domain transfer (Lin et al., 2025a; Wang et al., 2023b; Zhang et al., 2025m). Another direction incorporates evaluative signals derived from human preferences or verifiable proxies to better align generation with semantic intent, physical plausibility, or safety constraints (Xu et al., 2025b; Ji et al., 2025b; Yuan et al., 2024a). Meanwhile, some approaches leverage model-generated data and teacher supervision to enable iterative refinement and improve inference efficiency (Zhang et al., 2025g; Bi et al., 2025; Yin et al., 2025). In parallel, a growing body of work explores how post-trained signals

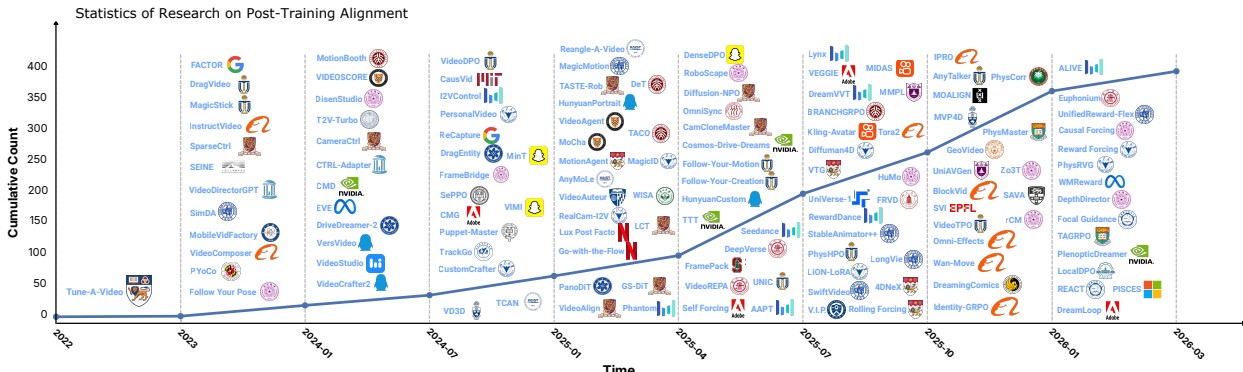

Figure 2: Research trends in post-training and alignment for video generation models (2022–Feb. 2026).

can be reused at deployment time to steer generation without further parameter updates (Liu et al., 2025a; Chen et al., 2025i; Deng et al., 2024). Together, these directions mark a shift from purely scaling-driven improvements toward modular, signal-driven alignment strategies, forming a flexible toolbox for addressing the unique temporal and multi-objective challenges of video generation.

In this survey, **post-training** refers broadly to any optimization, adaptation, or control procedure applied after large-scale pretraining that modifies the behavior of a video generation model without retraining it from scratch. These methods operate on pretrained foundation models and aim to shape model behavior in downstream use. We use the term **alignment** to describe the extent to which a video generation model's behavior conforms to desired objectives at deployment. These objectives include accurately following human intent, maintaining temporal and identity consistency, respecting physical and causal constraints, and avoiding unsafe or undesirable outcomes. In this sense, alignment concerns **behavioral correctness** and reliability, rather than visual quality or data fit alone.

Within this post-training framework, we distinguish between **implicit alignment** and **explicit alignment** based on how alignment signals are applied. **Implicit alignment methods** shape model behavior indirectly. They rely on mechanisms such as supervised adaptation, model-generated or teacher-provided signals, or structured controllability mechanisms, without explicitly evaluating whether generated outputs satisfy alignment objectives. In contrast, **explicit alignment methods** directly optimize model behavior using evaluative signals that assess correctness. These signals may include preference feedback, reward functions, or verifiable criteria related to human intent, physical plausibility, or safety. Importantly, alignment in video generation exists on a **spectrum**: post-training methods differ in how directly, strongly, and reliably they influence aligned behavior, rather than forming a strict binary between aligned and non-aligned approaches. This distinction is orthogonal to the specific training or inference mechanisms employed and reflects *how* alignment is enforced rather than *when* or *where* optimization occurs.

Based on these definitions, we organize post-training and alignment methods for video generation into four broad categories according to the primary source and role of the signals used to shape model behavior. **(1) Supervised Fine-tuning Methods** primarily achieve implicit alignment by adapting pretrained models using labeled or structured supervision. **(2) Self-Training and Distillation Methods** also promote implicit alignment, leveraging model-generated data or teacher supervision to improve robustness, stability, or efficiency without explicitly evaluating correctness. **(3) Preference- and Reward-Based Methods** enable explicit alignment by optimizing model behavior with evaluative signals that assess correctness with respect to human intent, physical plausibility, or safety. **(4) Inference-Time Methods** influence alignment at deployment, either by enforcing explicit alignment through evaluative guidance or by supporting implicit alignment via iterative refinement and structured control. Together, these categories provide a unified and interpretable view of how post-training techniques shape alignment in video generation models across both training and inference. Figure 3 presents the overall taxonomy of post-training and alignment methods.

**Relationship to Existing Surveys.** Several recent surveys review video generation from perspectives complementary to ours. Xing et al. (2024c) provides a broad overview of video diffusion models, covering

generation, editing, and understanding tasks. Wang et al. (2025s) further expands this view by discussing architectural advances, evaluation protocols, and industrial developments. Ma et al. (2025d) focuses specifically on controllable video generation, organizing methods by the type of conditioning signal, such as depth, pose, camera trajectory, or audio. Lei et al. (2024) concentrates on human-centric video generation, including talking-head synthesis, portrait animation, and dance generation. These surveys offer valuable perspectives on model design, controllability, and domain-specific applications. Our survey differs in scope by focusing specifically on post-training and alignment in video generation. Rather than organizing methods by architecture or conditioning signal, we organize them by how alignment is enforced, including supervised fine-tuning, self-training and distillation, preference- and reward-based optimization, and inference-time methods. In this sense, our survey provides a dedicated and, to our knowledge, the first comprehensive review of post-training and alignment methods for video generation, complementing existing overviews of the broader landscape.

In short, the key contributions of this survey are as follows:

> **Contributions**
>
> - **Post-Training Methods for Video Generation**. We provide a comprehensive review of post-training and alignment methodologies for video generation models, including supervised fine-tuning, preference- and reward-based optimization, self-training and distillation, and inference-time alignment and control techniques.
>
> - **Taxonomy of Alignment Techniques**. We introduce a structured taxonomy that organizes post-training approaches according to their optimization mechanisms and alignment roles, highlighting adaptations to video-specific challenges.
>
> - **Datasets and Benchmarks for Video Alignment**. We systematically summarize commonly used datasets, benchmarks, and evaluation protocols for post-training and alignment in video generation, categorizing them by alignment objectives and temporal characteristics.

> **Survey Structure**
>
> - **Section 2: Preliminaries.** Problem formulation of video generation, dominant base models, and multi-dimensional alignment objectives.
>
> - **Section 3: Supervised Fine-tuning Methods.** Implicit alignment through supervised adaptation, including instruction tuning, domain specialization, controllability, personalization, and structured data pipelines.
>
> - **Section 4: Self-training and Knowledge Distillation.** Implicit alignment via self-generated supervision and teacher-student distillation.
>
> - **Section 5: Preference- and Reward-Based Methods.** Explicit alignment using reinforcement learning, preference optimization, and video reward modeling.
>
> - **Section 6: Inference-Time Methods.** Hybrid alignment at deployment through guidance-based control and iterative refinement.
>
> - **Section 7: Cross-Family Comparison and Multi-stage Pipelines.** Comparison of different post-training families, the role of backbone architecture in shaping post-training interfaces, and the multi-stage composition patterns used in modern video generation systems.
>
> - **Section 8: Datasets, Benchmarks, and Evaluation Protocols.** Post-training datasets, evaluation benchmarks, and assessment protocols for alignment in video generation.
>
> - **Section 9: Challenges and Future Directions.** Key open challenges and future directions for post-training and alignment in video generation models.

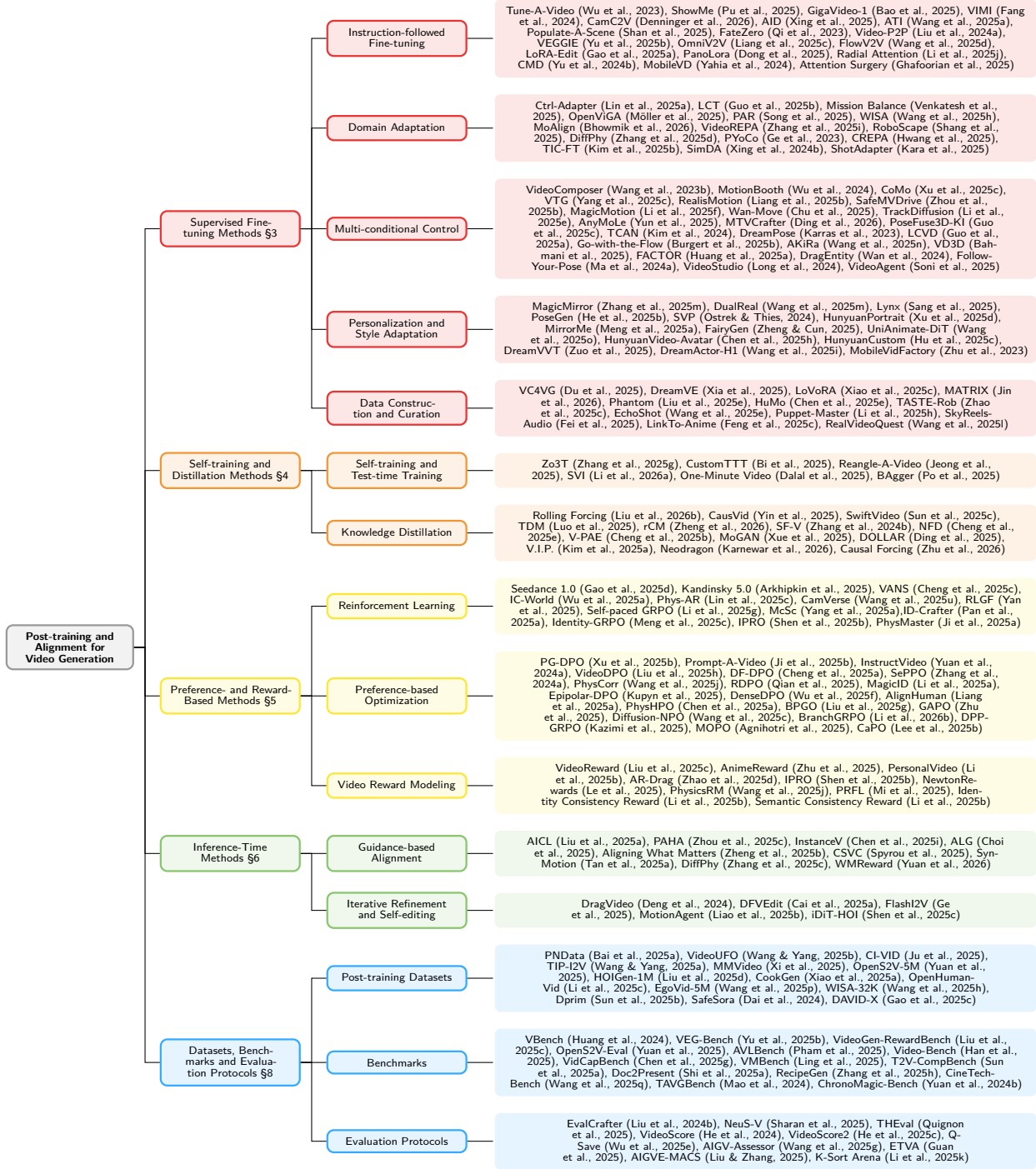

Figure 3: **Taxonomy of post-training and alignment in video generation models.** Methods are grouped by alignment type: Supervised fine-tuning and self-training and distillation methods provide *implicit alignment*. Preference- and reward-based methods achieve *explicit alignment*. Inference-time methods function as *hybrid mechanisms*, supporting both implicit and explicit alignment. The bottom node lists representative post-training datasets, benchmarks, and evaluation protocols.

## 2 Preliminaries: Video Generation Models and Alignment Dimensions

**Takeaways**

- **Video Generation Paradigms**: Video generation is formalized as learning a conditional distribution under three primary settings: Text-to-Video (T2V), Image-to-Video (I2V), and Video-to-Video (V2V).

- **Dominant Base Models**: Latent Diffusion Transformers (DiTs) are identified as the foundation of modern video generation and post-training, characterized by spatio-temporal latent representations, attention-based conditioning, and diffusion- or flow-based objectives.

- **Alignment Objectives**: Alignment is characterized as a multi-objective problem beyond likelihood-based training, requiring satisfaction of instruction adherence, temporal coherence, motion realism, perceptual quality, and safety constraints.

This section introduces the foundational formulation of video generation models and the alignment objectives that guide post-training. We first formalize common video generation settings, including text-to-video, image-to-video, and video-to-video, and outline the dominant architectural paradigms behind modern systems. We then characterize alignment in video generation as a multi-objective problem spanning instruction adherence, temporal coherence, motion realism, and safety. These preliminaries provide the conceptual and technical basis for understanding how subsequent post-training methods shape model behavior.

### 2.1 Video Generation Problem Setting

Formally, video generation is modeled as learning a conditional probability distribution $p(\mathbf{v}|\mathbf{c})$, where $\mathbf{v}$ represents a video sequence, and $\mathbf{c}$ denotes conditioning signals such as text, images, or edit instructions (Ho et al., 2022; Rombach et al., 2022). As illustrated in Figure 4, video generation tasks can be categorized into three paradigms based on input modalities and generation mechanisms.

**(1) Text-to-Video (T2V).** The goal of T2V is to synthesize a video $\mathbf{v}$ from a textual prompt $\mathbf{c}_{\text{text}}$ by sampling from a learned conditional distribution:

$$\mathbf{v} \sim p_\theta(\mathbf{v} \mid \mathbf{c}_{\text{text}}), \tag{1}$$

where $\theta$ denotes the parameters of the video generation model. This process corresponds to generation from scratch, requiring the model to produce both spatial content and temporal dynamics solely from the learned prior and textual conditioning (Hong et al., 2022; Singer et al., 2022). In practice, sampling typically begins with random noise $\mathbf{z}_T \sim \mathcal{N}(\mathbf{0}, \mathbf{I})$ in latent space, which is iteratively denoised using a spatio-temporal backbone, such as a 3D U-Net (Çiçek et al., 2016) or a Diffusion Transformer (DiT) (Peebles & Xie, 2023). The primary alignment challenges in T2V include semantic adherence to complex instructions and maintaining physical plausibility in open-domain, long-horizon video generation (Lin et al., 2025c).

**(2) Image-to-Video (I2V).** I2V, conditions generation on both a text prompt $\mathbf{c}_{\text{text}}$ and a reference image $\mathbf{I}_{\text{ref}}$ (typically the first frame) (Guo et al., 2024b; Xing et al., 2024a). The objective is to generate temporal dynamics that extend the context of $\mathbf{I}_{\text{ref}}$ while preserving its identity and visual details.

$$\mathbf{v} \sim p_\theta(\mathbf{v}|\mathbf{c}_{\text{text}}, \mathbf{I}_{\text{ref}}) \quad \text{s.t.} \quad \mathbf{v}_0 \approx \mathbf{I}_{\text{ref}}. \tag{2}$$

This is achieved by conditioning spatio-temporal diffusion backbones on image representations (e.g., via cross-attention), expanding the static image into a coherent temporal sequence. The alignment focus is on motion fidelity and preventing identity degradation over time (Xu et al., 2025d; Li et al., 2025f).

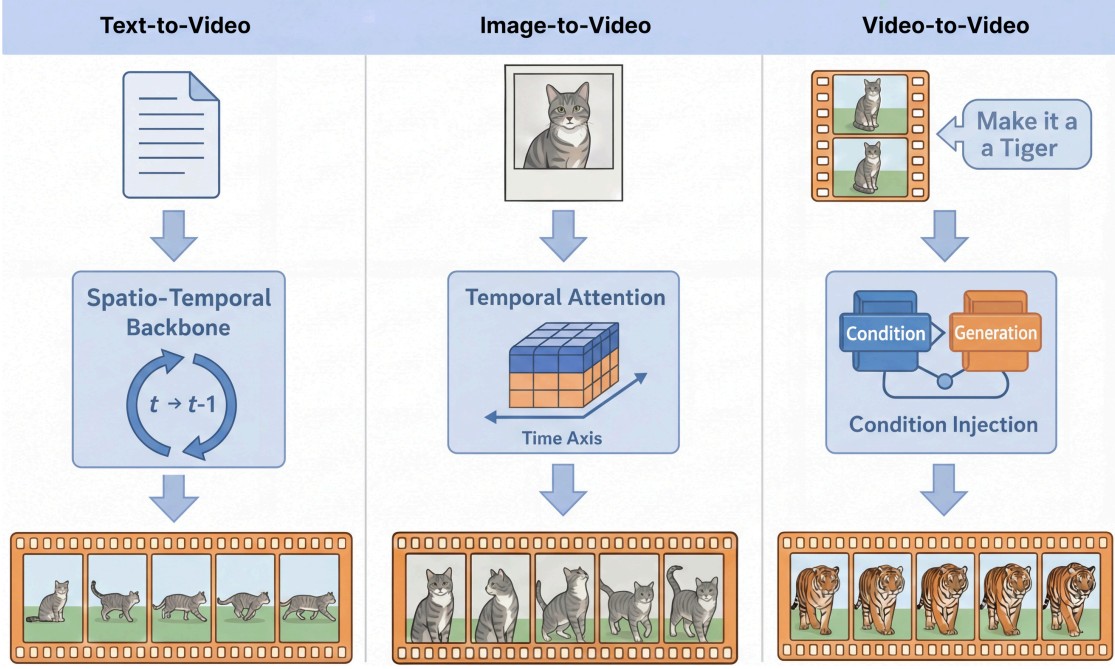

Figure 4: **Overview of video generation tasks**. **Left**: Text-to-Video (T2V) models learn a video prior to map noise to pixels guided by text prompts. **Middle**: Image-to-Video (I2V) injects dynamics into a static image, typically freezing spatial layers and training temporal attention modules. **Right**: Video-to-Video (V2V) focuses on structure-preserving editing, injecting spatial guidance to align with the source layout.

**(3) Video-to-Video (V2V) and Editing.** V2V aims to transform a source video $\mathbf{v}_{\text{src}}$ into a target video $\mathbf{v}_{\text{tgt}}$ according to an editing instruction $\mathbf{c}_{\text{edit}}$, while preserving its spatial-temporal layout (e.g., object motion, depth) (Qi et al., 2023; Liu et al., 2024a).

$$\mathbf{v}_{\text{tgt}} \sim p_\theta(\mathbf{v}|\mathbf{c}_{\text{edit}}, \mathbf{v}_{\text{src}}) \quad \text{s.t.} \quad \mathcal{S}(\mathbf{v}_{\text{tgt}}) \approx \mathcal{S}(\mathbf{v}_{\text{src}}), \tag{3}$$

where $\mathcal{S}(\cdot)$ represents structural features. Compared to text-to-video generation, V2V editing introduces an explicit structural consistency constraint that couples semantic modification with temporal coherence. The core challenge lies in balancing edit strength with structural preservation, as aggressive edits may disrupt motion dynamics, whereas conservative edits may fail to realize the intended transformation. To address this, existing methods typically employ **conditioning injection mechanisms** (e.g., ControlNet (Zhang et al., 2023) or lightweight adapters (Yu et al., 2025b)) or inversion-based guidance to anchor spatial layouts while modifying high-level semantics, prioritizing structural consistency and localized editability.

## 2.2 Base Models

While early work on video generation explores GAN-based architectures (Goodfellow et al., 2020) and 3D U-Nets (Çiçek et al., 2016), these approaches have increasingly been replaced in large-scale settings by Transformer-based backbones (Peebles & Xie, 2023). This shift is driven by the superior scalability of Transformers and their ability to model complex, long-horizon spatio-temporal dependencies. As a result, contemporary post-training methods primarily focus on two modern paradigms: latent diffusion models with Transformer backbones and autoregressive video generation models.

**Latent Diffusion Transformers (DiT) with Flow Matching.** Figure 5 illustrates the dominant latent diffusion pipeline adopted by modern video generation models in 2024–2025 (e.g., Wan (Wan et al., 2025), HunyuanVideo (Kong et al., 2024), OpenSora (Lin et al., 2024a)), which combines a spatio-temporal latent encoder, typically implemented as a 3D VAE, with a Diffusion Transformer backbone. Understanding its internal components is vital for effective alignment:

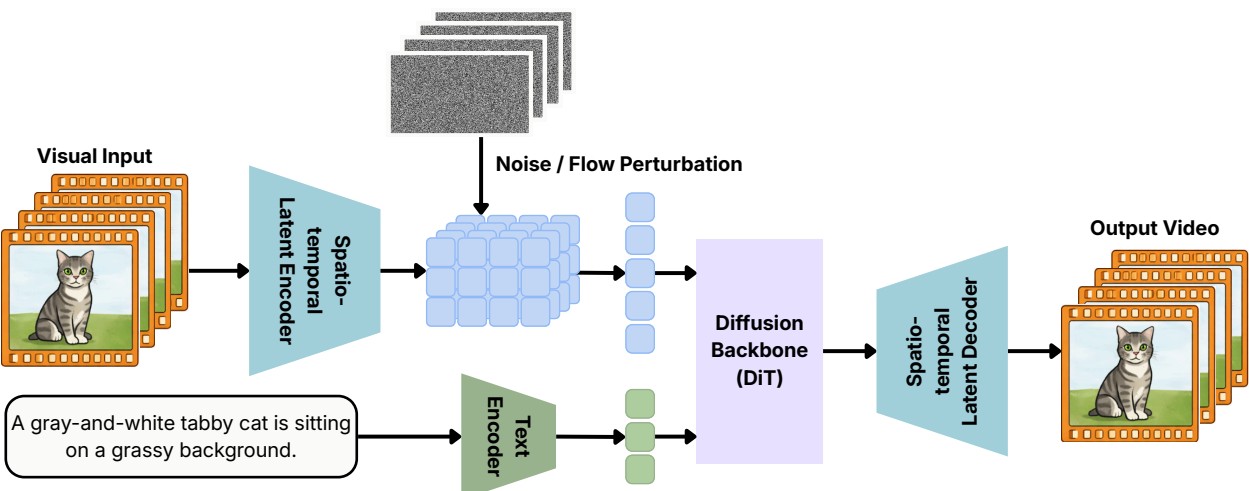

Figure 5: **Architecture of a modern Latent Diffusion Transformer (DiT) for video generation.** The pipeline has three stages: (1) **Compression:** A spatio-temporal latent encoder compresses the input video into latent representations. (2) **Diffusion Modeling:** Stochastic perturbations are applied in latent space, and the resulting spatio-temporal tokens (**blue tokens**) are processed by a DiT backbone together with text embeddings (**green tokens**) to learn a denoising objective. (3) **Decoding:** The predicted clean latents are mapped back to pixel space by a spatio-temporal latent decoder.

- **Spatio-temporal Latent Compression:** Videos are compressed into a latent space $\mathbf{z} = \mathcal{E}(\mathbf{v})$. Unlike frame-wise image VAEs, modern video encoders perform both spatial and temporal compression (Yu et al., 2024a), significantly reducing sequence length and making large-scale video generation computationally feasible. However, this compression also makes fine-grained temporal details less directly accessible, increasing the difficulty of precise motion control during post-training.

- **Spatio-temporal Patchification and Positional Encoding:** Latent tensors are flattened into spatio-temporal tokens and augmented with factorized or 3D Rotary Positional Embeddings (RoPE) (Su et al., 2024), enabling variable temporal lengths and robust spatio-temporal generalization. These components determine how the model represents spatial layout and temporal order. Long-video post-training methods therefore often require explicit handling of token positions and positional encodings to avoid temporal drift or positional mismatch.

- **Diffusion Transformer Backbone and Conditioning:** The denoising network is implemented as a ViT-style backbone, where conditioning signals (e.g., text prompts) are injected via cross-attention or Adaptive Layer Normalization (AdaLN). This module serves as the primary interface for post-training and alignment, since it directly controls how semantic instructions are translated into latent video dynamics. Adapter-based methods such as LoRA and ControlNet typically attach to attention or modulation blocks within this backbone.

- **Pre-training Objective (Flow Matching):** Many modern models adopt Flow Matching, often instantiated as Rectified Flow (Liu et al., 2023b), which learns a velocity field in latent space to map noise to data (Lipman et al., 2023; Liu et al., 2023c). This objective supports efficient sampling and provides the foundation for subsequent post-training and preference-based alignment methods. During alignment, the learned flow must be adjusted to improve human preference satisfaction while preserving generation stability.

**Autoregressive Video Generation.** As a complementary paradigm to diffusion-based models, autoregressive approaches (e.g., VideoPoet (Kondratyuk et al., 2024), VideoMAR (Yu et al., 2025a)) formulate video generation as a sequence modeling problem over discretized spatio-temporal tokens. Videos are first mapped into a token sequence, and generation proceeds in a causal manner via next-token prediction, analogous to

large language models (LLM):

$$p_\theta(\mathbf{v}) = \prod_i p_\theta(z_i \mid z_{<i}, \mathbf{c}).  \tag{4}$$

A key advantage of this formulation is that it enables the *direct application* of established LLM alignment algorithms (e.g., standard PPO (Schulman et al., 2017) or DPO (Rafailov et al., 2023) on token logits) without the adaptations required for continuous diffusion processes (Gupta et al., 2024). However, given that the current open-source landscape and recent post-training advancements are predominantly centered on diffusion architectures, most alignment methods discussed in this survey focus on optimizing continuous diffusion trajectories rather than discrete token sequences.

### 2.3 Alignment Dimensions for Video Generation

Despite powerful architectures, pre-trained models optimize for *data likelihood* rather than *human utility*. They tend to reproduce the "average" web video, often containing motion blur, static scenes, or uncurated compositions. Post-training and alignment aim to bridge this gap. Unlike image generation, alignment in video generation must simultaneously ensure per-frame visual quality and coherent dynamics across time. Based on these challenges, we categorize the primary alignment objectives into four key dimensions.

**(1) Instruction Following and Fine-grained Controllability.** A central objective of alignment is ensuring that generated videos accurately reflect user intent across multiple modalities. Beyond basic text-semantic matching, this requires models to correctly interpret and execute complex instructions, including multi-step logic, compositional descriptions, and explicit constraints such as edits or exclusions (e.g., "remove the object" or "keep the background unchanged") (Yuan et al., 2024a; Ma et al., 2024a). In practical scenarios, alignment must also support fine-grained controllability, where generation is conditioned on structured signals such as camera trajectories, depth maps, skeletal poses, or spatial layouts. These controls allow users to specify how scenes evolve and how actions are performed, rather than only describing what content should appear.

**(2) Temporal Consistency and Identity Preservation.** Beyond correctly interpreting user intent, a core challenge in video generation is maintaining coherence over time. Alignment methods in this dimension aim to reduce temporal artifacts caused by frame-level inconsistencies, such as flickering textures, unstable backgrounds, or unintended shape changes (Ma et al., 2024b). Beyond short-term stability, alignment must ensure that the identity of subjects remains consistent throughout a video (Molad et al., 2023; Hu, 2024). In long-form or personalized generation, characters or objects are expected to maintain the same appearance, including clothing, facial features, and overall visual style, even as they move, change viewpoint, or become partially occluded. When these requirements are not met, identity drift gradually accumulates, resulting in videos that appear unrealistic or inconsistent.

**(3) Motion Quality and Physical Plausibility.** While temporal consistency emphasizes stability over time, overly conservative generation can lead to static or lifeless videos. Pre-trained video generation models often favor static scenes or very small movements, since limited motion reduces the risk of visible errors during generation. Alignment in this dimension aims to encourage more expressive and dynamic motion that better reflects realistic actions and interactions (Wang et al., 2024; Yin et al., 2023). At the same time, the generated motion must obey basic physical rules (Meng et al., 2025b; Huang et al., 2024). This includes respecting constraints such as gravity, collisions between objects, object permanence, and simple cause-and-effect relationships. Effective alignment helps prevent visible artifacts, such as objects unrealistically disappearing, intersecting, or behaving in ways that contradict the physical structure of the scene.

**(4) Aesthetic Fidelity and Safety.** Beyond motion and physical correctness, alignment must also address overall perceptual quality and responsible generation behavior. From an aesthetic perspective, alignment aims to produce videos with high visual clarity, stable composition, and minimal perceptual artifacts, such as motion blur or distorted body parts. It also enables models to match human aesthetic preferences, including consistent lighting, color tone, and recognizable artistic styles (Kong et al., 2024; Liu et al., 2025h; Hu et al., 2025c). At the same time, alignment must ensure safe and reliable generation behavior. This includes reducing the production of harmful, biased, or NSFW content, as well as ensuring that models appropriately refuse unsafe requests or suppress undesirable concepts during generation (Dai et al., 2024).

# 3 Supervised Fine-tuning Methods

**Takeaways**

- Supervised fine-tuning is the primary mechanism for aligning pretrained video generation models with user intent, controllability requirements, and domain-specific constraints, without retraining or altering core model architectures from scratch, typically through targeted adaptation of pretrained models.

- By integrating instruction tuning, domain adaptation, and multi-conditional supervision, supervised fine-tuning enables control over semantics, motion, camera behavior, and spatial layout beyond text-only guidance.

- Lightweight adaptation, combined with structured and synthetic data pipelines, supports identity preservation, personalization, and robust alignment under limited supervision.

Supervised fine-tuning adapts pretrained video generation models through targeted supervision signals after large-scale pretraining. This family is most effective when aligned behavior can be improved by adding structured task-specific supervision, especially for instruction following, controllability, domain-specific adaptation, and personalization. Rather than directly optimizing preferences or rewards, supervised fine-tuning refines the conditional mapping from user inputs and control signals to target videos. Supervised fine-tuning methods are typically categorized according to the form of supervision they employ and the alignment capabilities they provide.

## 3.1 Preliminaries: A Unified View of Supervised Objectives

Although supervised fine-tuning methods differ in conditioning modality and adaptation scope, many can be viewed under a common conditional learning framework. Given conditioning inputs $x$ (e.g., text prompts, reference images, edit instructions, source videos, or structured control signals) and target video $v$, we consider a conditional generator $f_\theta$ trained with a supervised objective of the form

$$\mathcal{L}_{\mathrm{sup}} = \mathbb{E}_{(x,v)\sim\mathcal{D}}\big[\ell_\theta(x,v)\big], \tag{5}$$

Here, $\ell_\theta(x,v)$ is a model-specific loss defined over representations induced by $f_\theta$ under conditioning input $x$ and target $v$. Its exact form depends on the underlying generator. For diffusion- and flow-based video models, $\ell_\theta$ typically corresponds to denoising, noise-prediction, latent reconstruction, or velocity (vector-field) matching objectives defined in latent space, depending on the parameterization. Parameter-efficient tuning methods operate under the same objective while restricting optimization to a subset of parameters, such as adapters or LoRA modules.

A broad class of specialization methods further augments this objective with regularization terms:

$$\mathcal{L} = \mathcal{L}_{\mathrm{sup}} + \lambda\mathcal{L}_{\mathrm{reg}}, \tag{6}$$

where $\mathcal{L}_{\mathrm{reg}}$ enforces desirable properties such as preservation of pretrained priors, temporal consistency, or robustness under domain shift.

This formulation provides a unified optimization view of supervised post-training despite substantial variation in supervision signals, architectural interfaces, and downstream alignment goals. In the remainder of this section, instruction tuning can be viewed as refining the mapping from natural-language directives to target videos, multi-conditioning extends $x$ to include structured control signals, and domain adaptation modifies the training distribution and regularization strategy to specialize pretrained models while retaining useful general priors.

## 3.2 Instruction and Prompt-following Fine-tuning

A primary goal of supervised fine-tuning is to improve a model's ability to follow user instructions. While large-scale pretraining provides video models with powerful generative priors, their responses to natural-language

prompts often remain coarse, ambiguous, or inconsistent over time. Instruction and prompt-following fine-tuning addresses this gap by explicitly aligning textual directives, such as editing commands, compositional constraints, and multi-step instructions, with the corresponding video outputs, typically using relatively small, curated instruction datasets.

**Direct Instruction-to-Video Supervision.** Early efforts in instruction-following video generation focus on directly aligning user instructions with video outputs through explicit fine-tuning of pretrained generative models (Hong et al., 2022; Wang et al., 2023a; 2025k). Tune-A-Video (Wu et al., 2023) provides a representative example by studying one-shot text-to-video generation from only a single text-video pair. Its core idea is to adapt a pretrained text-to-image diffusion model with a sparse causal spatio-temporal attention mechanism so that the strong image prior can be reused for temporally coherent video synthesis. At inference time, DDIM inversion provides structure guidance, allowing the tuned model to preserve the input video layout while learning continuous motion from extremely limited supervision. ShowMe (Pu et al., 2025) extends direct instruction supervision in a different direction by unifying instructional image editing and video prediction within a single video diffusion model. Instead of focusing on one-shot adaptation, it treats both tasks as action-object state transformation and uses a two-stage tuning strategy with task-specific adapters to selectively activate spatial and temporal components.

Beyond strictly paired instruction-video supervision, more recent work explores scalable optimization strategies, ranging from joint image-video fine-tuning to inference-time adaptation that relaxes reliance on exhaustive video data while improving instruction adherence (Khachatryan et al., 2023; Chen et al., 2024a; Wang et al., 2025r). GigaVideo-1 (Bao et al., 2025) proposes an automatic dataset synthesis pipeline for video diffusion model fine-tuning that emphasizes physical and temporal consistency without relying on large-scale curated external datasets. The method leverages LLM-augmented prompt generation and a reward-guided optimization strategy, where feedback from a frozen multimodal large language model (MLLM) is used to adaptively reweight synthesized training samples during fine-tuning. VIMI (Fang et al., 2024) further extends instruction supervision to multimodal settings by introducing a multimodal instruction pretraining framework for grounded video generation. The framework constructs a large-scale multimodal prompt-video dataset via retrieval-augmented in-context examples and employs a two-stage pipeline of multimodal conditional video pretraining and multimodal instruction tuning, leveraging MLLMs to unify text-to-video, subject-driven video generation, and video prediction within a single model.

**Image Guided Video Generation.** While direct instruction-video supervision aligns generation with user intent, it often suffers from ambiguity and instability when synthesizing long or complex dynamics. A related line of work mitigates this issue by introducing visual context as an additional grounding signal (Blattmann et al., 2023b; Denninger et al., 2026). AID (Xing et al., 2025) provides a representative example by adapting a pretrained image-to-video diffusion model for instruction-guided video prediction. Its key idea is to reuse the dynamics priors of Stable Video Diffusion while injecting textual control through an MLLM and a Dual Query Transformer, which fuse the input image and instruction into conditional embeddings for future-frame prediction. The method further introduces temporal and spatial adapters, allowing the pretrained video prior to be transferred to instruction-guided prediction with relatively low adaptation cost. ATI (Wang et al., 2025a) emphasizes a different role of image guidance: instead of using the image mainly to ground future prediction under instructions, it uses the input image as the reference canvas for fine-grained trajectory control, with a Gaussian-based motion injector encoding local, object-level, and camera motion into the latent space. Populate-A-Scene (Shan et al., 2025) extends image guidance toward scene-aware semantic interaction, using a scene image together with prompts about human appearance and action to generate affordance-aware human-world interactions.

**Instruction-guided Video Editing.** Instruction-guided video editing aims to modify existing video content according to user instructions while preserving temporal coherence and physical consistency, posing challenges beyond those in unconditional or image-based editing (Qi et al., 2023; Liu et al., 2024a). VEGGIE (Yu et al., 2025b) addresses this with an end-to-end framework that integrates video concept editing, grounding, and reasoning based on user instructions. The system employs an MLLM to interpret user intents into frame-specific queries and uses a curriculum learning strategy, along with a pipeline that transforms static

image data into dynamic video-editing samples. Similarly, OmniV2V (Liang et al., 2025c) explores a unified dynamic content manipulation module to integrate various scenario-based operations. It incorporates a LLaVA-based visual-text instruction module (Liu et al., 2023a) to understand content correspondence and utilizes a multi-task data processing system to efficiently handle data overlap and augmentation. Beyond semantic and structural edits, maintaining physical plausibility during instruction-guided motion transfer remains a key challenge. FlowV2V (Wang et al., 2025d) explicitly targets this issue by employing optical flow to model complex motion dynamics and mitigate failures caused by shape deformation. The approach combines first-frame editing with conditional generation by simulating a pseudo-flow sequence aligned with the deformed shape, enabling physically consistent video editing under user instructions.

**Parameter-efficient and Efficiency-aware Instruction Fine-tuning.** Beyond expanding the scope of instruction-aligned behaviors, a complementary line of supervised fine-tuning work addresses efficiency at both the adaptation level and the model-computation level to enable instruction- or task-specific specialization under the high computational costs of video generation models. On the adaptation side, parameter-efficient fine-tuning strategies update only a small subset of model parameters while keeping the pretrained backbone frozen (Acuaviva et al., 2025; Kim et al., 2025b; Abdal et al., 2025; Gao et al., 2025a). PanoLora (Dong et al., 2025) exemplifies this direction by framing panoramic video generation as a specialization problem and proposing a LoRA-based fine-tuning strategy, supported by analysis showing that low-rank updates suffice to model the transformation. Beyond reducing the number of trainable parameters, several works further address the prohibitive computational overhead of video generation through architectural and attention-level optimizations (Li et al., 2025j; Yu et al., 2024b). MobileVD (Yahia et al., 2024) reduces memory usage by lowering frame resolution and applying channel-wise and temporal block pruning, and further compresses the denoising process into a single step via adversarial training. At the transformer level, Attention Surgery (Ghafoorian et al., 2025) introduces hybrid attention mechanisms guided by a cost-aware block-rate strategy that balances expressiveness and efficiency across layers based on the observation that different blocks exhibit varying reconstruction errors under different token sample ratios. From a representation perspective, CMD (Yu et al., 2024b) proposes an autoencoder that decouples video content and motion into a content frame and a low-dimensional motion latent. The content frames are generated by a fine-tuned image diffusion model, while motion latents are produced by a lightweight diffusion model, enabling the reuse of pretrained image models within a compact latent space.

## 3.3 Domain Adaptation and Specialization

General-purpose video generation models, while powerful in open-domain settings, often encounter significant performance degradation when applied to specialized fields such as healthcare, industrial physics, or long-form storytelling. These failures arise from domain shifts or capability gaps relative to the pretrained setting, where the target data distribution differs significantly from the web-scale data used during pretraining. Unlike instruction and prompt-following fine-tuning, which primarily aligns models with user intent, domain adaptation focuses on aligning a pretrained model's internal representations and spatiotemporal priors with a new target distribution. Consequently, supervised fine-tuning for domain adaptation has shifted from simple fine-tuning toward approaches that emphasize domain-specific supervision, robustness to distribution shift, and efficient specialization.

**Navigating Domain Shifts and Data Scarcity.** We begin by characterizing the types of domain shifts that motivate specialization in video generation models, which often arise when foundation models fail to generalize to specialized visual distributions or operate reliably under data scarcity. Beyond appearance-level shifts, many specialization scenarios impose structural requirements that deviate substantially from web-scale training data (Lin et al., 2025a). A representative example is long-form storytelling, in which the target distribution requires scene-level coherence and long-range temporal consistency rather than short, loosely connected clips. LCT (Guo et al., 2025b) addresses this shift by expanding the context window of a pretrained single-shot video diffusion model from individual shots to entire scenes through supervised long-context tuning. Its design introduces interleaved 3D positional embeddings and an asynchronous noise strategy, enabling the adapted model to learn cross-shot consistency directly from scene-level data while supporting both joint and autoregressive shot generation.

A different type of shift arises in high-stakes domains such as healthcare, where the primary challenge is data scarcity rather than long-context structure. Mission Balance (Venkatesh et al., 2025) tackles the "long-tail" problem in medical imaging where rare pathological events are underrepresented. It introduces a two-stage fine-tuning approach that decouples spatial fidelity from temporal dynamics, allowing the model to synthesize high-fidelity surgical videos even with limited training examples. More broadly, domain shifts also arise in specialized content distributions such as automotive driving scenes (Möller et al., 2025) and panoramic video generation (Dong et al., 2025), as well as settings where models must better adhere to physical commonsense under distribution shift (Thozhiyoor et al., 2025; Song et al., 2025).

**Domain-Specific Supervision Signals.** To effectively transfer models to specialized domains, researchers increasingly employ domain-specific supervision signals that go beyond generic text instructions. Such signals often encode task-relevant structure, physical constraints, or relational cues that are underrepresented in web-scale video-text data (Wang et al., 2025h; Bhowmik et al., 2026). In domains governed by physical laws, VideoREPA (Zhang et al., 2025i) improves the physical commonsense of text-to-video generation by aligning video models with relational and physics-relevant cues distilled from foundation models, providing an implicit yet domain-aligned supervision signal for physically plausible dynamics. Pushing beyond generic plausibility toward actionable interactions, RoboScape (Shang et al., 2025) introduces a physics-informed world model for embodied AI. Instead of relying solely on RGB pixel loss, it jointly learns video generation and auxiliary physics prediction tasks. This form of implicit physical supervision encourages the model to respect 3D geometry and object interactions, producing video simulations suitable for robotic policy training.

**Robust Fine-Tuning and Catastrophic Forgetting.** A central challenge in domain adaptation is catastrophic forgetting: naïvely fine-tuning a pretrained video generator on a narrow in-domain dataset can improve domain-specific fidelity while degrading general prompt-following behavior or disrupting learned spatiotemporal priors outside the adapted distribution. This phenomenon reflects a fundamental tension between specialization and retention in video generation models. Recent supervised fine-tuning methods, therefore, focus on retention-aware objectives that stabilize adaptation under distribution shift and data scarcity (Zhang et al., 2025d). PYoCo (Ge et al., 2023) provides a representative example by identifying the noise prior itself as a source of temporal degradation when adapting image diffusion models to video. Its key idea is to replace the frame-independent image noise prior with a video noise prior that preserves temporal correlations across frames, thereby reducing motion-structure collapse during fine-tuning. CREPA (Hwang et al., 2025) addresses a failure mode at the representation level. Instead of aligning each frame only to its own external feature target, it aligns hidden states with features from neighboring frames, explicitly encouraging cross-frame semantic consistency and improving robustness during parameter-efficient adaptation.

TIC-FT (Kim et al., 2025b) further stabilizes adaptation through temporally structured conditioning. It concatenates the condition and target frames along the temporal axis, with buffer frames of gradually increasing noise inserted in between. This helps fine-tune better to follow the pretrained model's temporal dynamics and improves generalization under limited data. Together, these works highlight that effective domain adaptation requires not only stronger in-domain supervision but also objectives that preserve the pretrained model's general spatiotemporal priors.

**Efficient Adaptation for Specialization.** Beyond robustness, practical deployment introduces additional constraints on scalability and efficiency. Domain adaptation often requires reusing a single pretrained model across multiple specialized domains, where full fine-tuning is both prohibitively expensive and prone to overfitting under limited in-domain data. In this context, parameter-efficient post-training methods become a practical necessity rather than a mere optimization choice. Adapter-based approaches such as SimDA (Xing et al., 2024b) enable specialization by updating only a small fraction of parameters while preserving shared spatiotemporal priors. It uses lightweight spatial and temporal adapters to transfer a pretrained image diffusion backbone to video generation. Similarly, LoRA-style adaptation has proven effective for small-data specialization, including in the work of Çatay et al. (2025), LoRA modules are inserted into the cross-attention layers of a pretrained image-to-video model to adapt its visual representations to a cinematic domain. Its core mechanism is a two-stage pipeline that first learns domain-specific visual style from a small dataset and

then expands the resulting stylized keyframes into temporally coherent videos, making it more specialized to small-data style transfer than SimDA's general image-to-video adaptation.

For long-form or multi-shot specialization, ShotAdapter (Kara et al., 2025) addresses a different problem by extending a pretrained single-shot generator to multi-shot generation. It introduces a transition token to control where new shots begin and a local attention masking strategy that enables shot-specific prompting, allowing the adapted model to generate multi-shot videos with controllable shot number, duration, and content without full retraining.

### 3.4 Multi-conditioning and Controllability

As video generation models are increasingly deployed in interactive and task-driven settings, aligning models with user intent through instruction fine-tuning alone often proves insufficient for precise and reliable control. Multi-conditioning and controllability methods address this limitation by training pretrained models to accept structured control inputs, including explicit signals and intermediate planning representations. These signals expose controllable interfaces over motion, spatial layout, and temporal evolution during generation, enabling fine-grained and reliable manipulation beyond implicit instruction execution.

**Motion and Camera Control.**  A prominent direction in controllable video generation focuses on explicit motion and camera control, where pretrained models are guided by structured temporal signals to regulate dynamics beyond their learned motion priors. Early efforts in this direction emphasize decoupling motion dynamics from visual appearance, enabling controllable motion manipulation while preserving content fidelity. For example, MotionBooth (Wu et al., 2024) provides a representative example by fine-tuning a text-to-video model on a few images of a customized subject while introducing subject-aware objectives to preserve appearance during motion-controlled generation. It further combines subject-motion control and camera-motion control through training-free inference mechanisms, showing how motion customization can be added without retraining a separate control model for each target subject. Complementarily, CoMo (Xu et al., 2025c) extends this line of work by treating motion control as a compositional problem, decomposing complex motions into reusable primitives that can be flexibly recombined under textual guidance. These approaches establish a foundational principle for motion control: separating temporal dynamics from appearance facilitates fine-grained manipulation while maintaining identity stability.

Building upon this principle, a large body of work introduces explicit motion-related conditions to more directly regulate temporal evolution. One common strategy is to guide video generation using structured motion signals such as trajectories (Yang et al., 2025c; Liang et al., 2025b; Zhou et al., 2025b; Li et al., 2025f; Chu et al., 2025; Xiao et al., 2025b; Li et al., 2025e; Yun et al., 2025; Lee et al., 2025c; Ding et al., 2026; Burgert et al., 2025a), poses (Guo et al., 2025c; Chen et al., 2025f; Liu et al., 2025f; Pallotta et al., 2025; Mahdi et al., 2025; Shao et al., 2025; Tu et al., 2025; Kong et al., 2025; Guo et al., 2025a; Liu et al., 2025i), or other temporally aligned control cues. By injecting these conditions into the temporal modeling components, such methods enable precise control over subject movement while preserving appearance-related content. VideoComposer (Wang et al., 2023b), for instance, is representative of this line because it treats controllable video generation as compositional synthesis over textual, spatial, and temporal conditions. In particular, it introduces motion vectors from compressed videos as explicit temporal control signals and uses a Spatio-Temporal Condition encoder to fuse sequential spatial and temporal cues through a unified interface. This design makes it possible to support motion transfer, user-specified trajectories, and other forms of temporal control without retraining the base model.

In addition to subject motion, camera controllability has emerged as a critical aspect of video generation (Burgert et al., 2025b; Bai et al., 2025b; Wang et al., 2025n; Pan et al., 2025b; He et al., 2025a; Cheong et al., 2025; Danier et al., 2025; Li et al., 2025i; Jeong et al., 2025; Ma et al., 2025e; Cheng et al., 2025d; Wu et al., 2026a; Wang et al., 2025v; Denninger et al., 2026; Guhan et al., 2025). Camera-aware methods explicitly model viewpoint evolution by conditioning diffusion transformers on camera paths or 3D camera parameters. VD3D (Bahmani et al., 2025) serves as a representative camera-side example by introducing a ControlNet-like conditioning mechanism with spatiotemporal camera embeddings. These embeddings encode per-frame 3D camera motion directly into the transformer-based video diffusion model, enabling precise viewpoint control while maintaining visual fidelity.

**Object-, Part-, and Spatial-level Control.** Complementary to global motion control, another line of work focuses on achieving object- and spatial-level controllability by explicitly binding generation to specific entities, regions, or parts within a scene (Huang et al., 2025b; Hu et al., 2025a; Shen et al., 2025c; Zheng et al., 2025b; Deng et al., 2024; Chen et al., 2024b; Mao et al., 2026; Pham et al., 2025; Chang et al., 2025; Xie et al., 2025; Zhou et al., 2025a; Wang et al., 2026c; Huang et al., 2025e; Deng et al., 2026b; Karnewar et al., 2026; lei Li et al., 2025). These approaches typically rely on structured spatial conditions such as masks, bounding boxes, instance tokens, or 3D proxies to preserve object identity and spatial consistency across frames while enabling localized manipulation. At the object level, FACTOR (Huang et al., 2025a) introduces fine-grained control by conditioning video generation on entity-specific appearance and spatial context. By jointly encoding object descriptions, sparse bounding-box trajectories, and reference images, FACTOR enables localized manipulation of multiple objects while maintaining consistent identities and spatial layouts over time.

Beyond individual object binding, relational multi-entity control further models spatial dependencies among interacting entities. DragEntity (Wan et al., 2024) represents each object as a latent entity and explicitly incorporates relative spatial relationships when applying trajectory guidance. This entity-centric formulation enables simultaneous control of multiple objects while preserving structural integrity and reducing the distortions commonly observed in pixel-level dragging approaches. At an even finer granularity, part-level control targets the internal structure and articulation of objects. Puppet-Master (Li et al., 2025h) binds sparse drag signals to specific object parts through dedicated drag tokens, enabling fine-grained internal dynamics such as articulation and deformation while maintaining overall object identity and spatial coherence across frames.

**Programmatic and Latent Control.** Beyond direct conditioning, some approaches treat controllable video generation as the execution of an explicit plan or program derived from high-level instructions. In these methods, natural language prompts are first translated into structured intermediate representations, such as scripts, trajectories, or action graphs, which are then executed or iteratively refined by video generation models to support long-horizon consistency and interpretable control (Zhang et al., 2025j;f; Zhao et al., 2025a; Xiang et al., 2025). Within this paradigm, VideoStudio (Long et al., 2024) casts video generation as a script-driven process by leveraging a large language model to convert an input prompt into a structured multi-scene program. The resulting script explicitly specifies scene-level events, entities, and camera movements, which are then executed by a diffusion model to generate each scene sequentially, enabling consistent content and coherent long-horizon video generation. While VideoStudio focuses on executing a fixed, LLM-generated program, VideoAgent (Soni et al., 2025) further extends this execution-centric perspective by treating generated videos as intermediate plans rather than final outputs. By iteratively refining and selecting video plans prior to execution, VideoAgent introduces an explicit plan selection and execution interface that separates high-level programmatic control from direct conditioning, thereby enabling controllability at the level of long-horizon behavior rather than frame-wise appearance.

**Joint Audio-Video Generation and Synchronization.** Audio has emerged as a powerful control signal for temporally precise video generation, particularly in human-centric scenarios requiring tight synchronization between visual dynamics and sound (Peng et al., 2025; Fei et al., 2025; Zhang et al., 2025k). This line of work is now extending to joint audio-video generation, where models must synthesize both modalities simultaneously while maintaining semantic consistency and fine-grained temporal alignment (Zhang et al., 2025b; Wang et al., 2025b; Liu et al., 2026a). This setting introduces a cross-modal alignment challenge: post-training must coordinate two coupled generative processes while avoiding temporal misalignment, lip-speech inconsistency, and degradation in unimodal quality. Apollo (Wang et al., 2026b) exemplifies this direction through a progressive pretrain-post-train curriculum. It first establishes single-modal and joint generation capabilities on large-scale multi-scene data, and then improves synchronization and cross-modal understanding through aligned audio-video training and increasingly constrained multi-task optimization. In this framework, post-training serves not only to improve synchronization, but also to preserve unimodal quality while scaling joint audiovisual generation. Together, these studies highlight the growing role of post-training in coordinating audio and visual generation, where synchronization, semantic consistency, and modality-specific quality must be optimized jointly.

### 3.5 Personalization and Style Adaptation

Personalization and style adaptation aim to customize video generation models to produce subject-consistent outputs that reflect specific identities, appearances, or stylistic preferences. Unlike generic controllability mechanisms that regulate motion or camera dynamics, personalization focuses on preserving identity fidelity across diverse motions, viewpoints, and conditioning signals, often under limited supervision or reference data. Recent advances explore lightweight post-training strategies that adapt pretrained video diffusion models to individual subjects, characters, or application-specific requirements, while maintaining the original model's generative capacity and temporal coherence. In this subsection, we review representative approaches from three complementary perspectives: identity-preserving personalization mechanisms, modality- and character-centric scenarios, and application-driven customization for humans and products.

**Identity-Preserving Personalization via Lightweight Conditioning and Adapters.** Several works explore supervised fine-tuning strategies to enhance identity fidelity in personalized video generation while minimizing disruption to motion dynamics and semantic alignment. Broadly, these approaches focus on either conditioning-based personalization or explicit disentanglement of identity and motion representations. MagicMirror (Zhang et al., 2025m) provides a representative example of conditioning-based personalization. Built on video diffusion transformers, it introduces a dual-branch facial feature extractor and a lightweight cross-modal adapter with conditioned normalization to inject identity information while preserving natural motion. A two-stage training strategy with synthetic identity pairs and video data further stabilizes identity-consistent generation without requiring person-specific full-model retraining. DualReal (Wang et al., 2025m) addresses a different aspect of personalization by explicitly modeling the identity–motion trade-off through adaptive joint training. Its framework alternates between identity-aware and motion-aware optimization phases and uses a stage-aware controller to regulate how the two dimensions are fused across denoising steps and transformer depths, improving the integration of appearance and dynamics under customized generation. Related efforts further investigate identity preservation under sparse conditioning signals or in multi-character interaction scenarios (Liao et al., 2025a; Sang et al., 2025; He et al., 2025b).

**Portrait-, Character-, and Multimodal-Centric Personalization.** A related line of work focuses on portrait-, character-, and multimodal-centric video personalization, where preserving subject identity across pose variation, expression changes, and modality shifts is particularly critical. In the portrait domain, HunyuanPortrait (Xu et al., 2025d) provides a representative example by combining implicit motion control with lightweight adapter-based personalization. It decouples portrait motion from identity using pretrained encoders, represents motion through implicit control signals, and injects these controls into Stable Video Diffusion through attention-based adapters, improving both temporal consistency and controllability in portrait animation. SVP (Ostrek & Thies, 2024) places greater emphasis on long-range facial consistency over extended sequences, while facelet-based compensation (Deng et al., 2025) targets robustness under partial occlusion and large head motion through localized facial correction. MirrorMe (Meng et al., 2025a) further extends portrait personalization to audio-driven animation, enabling identity-preserving facial motion synchronized with speech.

Beyond face-centric settings, personalization has been extended to character animation and multimodal generation. FairyGen (Zheng & Cun, 2025) adapts the same personalization goal to drawn characters by preserving visual style and character consistency from a single illustrated reference, while UniAnimate-DiT (Wang et al., 2025o) focuses on coherent human motion generation from reference images. It employs a large-scale video diffusion transformer to generate coherent human motion conditioned on reference images. In multimodal scenarios, HunyuanVideo-Avatar (Chen et al., 2025h) and HunyuanCustom (Hu et al., 2025c) extend subject-consistent generation to richer input conditions, including audio, images, video, and text, enabling more expressive and controllable character animation across modalities.

**Application-Driven Personalization for Humans and Products.** Beyond generic identity customization, personalization in video generation is often driven by application-specific requirements involving humans and products. A prominent line of work focuses on human-product interaction scenarios. DreamVVT (Zuo et al., 2025) targets realistic virtual try-on by introducing a stage-wise diffusion transformer framework that progressively aligns garment appearance with human motion and body structure under in-the-wild conditions.

Similarly, DreamActor-H1 (Wang et al., 2025i) addresses human–product demonstration videos by designing motion-aware diffusion transformers that generate high-fidelity interactions while preserving both human dynamics and product details. In addition to interaction-centric applications, personalization has also been explored under deployment- and efficiency-driven constraints. MobileVidFactory (Zhu et al., 2023) adapts diffusion-based video generation to mobile social media applications through a supervised pipeline optimized for computational efficiency and stylistic consistency, enabling automated personalized video creation from text prompts under resource-constrained settings. Together, these works suggest that personalization is increasingly shaped by downstream application needs, where identity preservation must be balanced with interaction realism, product fidelity, stylistic consistency, and deployment efficiency.

### 3.6 Data Construction and Curation Pipelines

Recent progress in video generation is strongly driven by advanced data construction pipelines that actively shape model training. Moving beyond raw video-text pairs, modern approaches design structured supervision and scalable labeling mechanisms to introduce intermediate semantic representations and automatically generated signals. These pipelines bridge user intent, object dynamics, and temporal coherence, thereby facilitating controllable and robust video generation.

**Structured Semantic Supervision.** Several pipelines enrich training data with intermediate semantic representations that explicitly guide temporal modeling and compositional generation. DreamVE (Xia et al., 2025) is a representative example of constructing structured supervision directly at the data level for unified image and video editing. It builds large-scale training pairs using two forms of synthetic data construction. One creates edited examples by composing visual elements, and the other uses generative models to produce edited outputs. This makes editing intent and compositional transformations explicit in the supervision, instead of relying only on raw captions. VC4VG (Du et al., 2025) follows a similar instruction-centric direction but places greater emphasis on optimized textual supervision for controllable generation. Beyond textual supervision, several data construction pipelines automatically derive object- and interaction-aware signals that serve as structured supervision during training. LoVoRA (Xiao et al., 2025c) and MATRIX (Jin et al., 2026) construct temporally aligned object localization and mask tracks to provide cross-frame object-level supervision without manual annotation.

Phantom (Liu et al., 2025e) and Puppet-Master (Li et al., 2025h) further extend this idea to part-level motion and cross-modal alignment representations, enabling disentangled supervision over appearance, structure, and dynamics. Human-centric pipelines such as HuMo (Chen et al., 2025e) and TASTE-Rob (Zhao et al., 2025c) instead emphasize pose and hand–object interaction cues as intermediate supervision signals to support structured motion learning. In audio-driven scenarios, recent pipelines proposed by Zhang et al. (2025e), EchoShot (Wang et al., 2025e), and SkyReels-Audio (Fei et al., 2025) incorporate fine-grained audio–visual alignment signals to enable temporally synchronized motion generation.

**Synthetic Data and Scalable Labeling.** To alleviate the high cost and limited scalability of dense video annotation, many pipelines adopt automated data generation and labeling strategies that function as implicit supervision mechanisms. In practice, these strategies differ in where the scalable supervision comes from. Some synthesize structured labels directly from controllable generation pipelines, some use learned evaluators as automatic feedback signals, and others organize supervision around realistic user intents instead of exhaustive frame-level annotation. LinkTo-Anime (Feng et al., 2025c) provides a representative example of synthetic supervision generation by rendering optical flow from controllable intermediate animation states, which yields accurate motion labels for training without manual annotation. This makes temporal supervision explicit at the data-construction stage and is particularly effective when motion signals can be generated more reliably than they can be annotated. VideoScore (He et al., 2024) illustrates a different form of scalable labeling. It learns an automatic feedback model that approximates fine-grained human judgments and can be reused for supervision and model selection at scale. Related efforts also explore organizing supervision around realistic user intents rather than exhaustive frame-level annotations, reducing annotation overhead while preserving semantic alignment (Wang et al., 2025l). Together, these approaches show how scalable supervision can be obtained from synthetic generation, learned feedback, and user-intent-driven data organization, offering practical alternatives to costly dense video annotation.

Table 1: Summary of supervised fine-tuning methods for video generation. Entries are sorted chronologically by publication date. "-" indicates that the corresponding information is not reported or not clearly specified in the original paper.

| Model | Sub-category | Stages | Base Model | GPU | Venue | Year | Link |
|---|---|---|---|---|---|---|---|
| Tune-A-Video (Wu et al., 2023) | Instruction-followed Fine-tuning | 1 | Stable Diffusion | A100 | ICCV | 2023 | ⌂ ✗ |
| VideoComposer (Wang et al., 2023b) | Multi-conditional Control | 2 | Stable Diffusion | - | NeurIPS | 2023 | ⌂ ✗ |
| DreamPose (Karras et al., 2023) | Multi-conditional Control | 2 | Stable Diffusion | 2xA100 | ICCV | 2023 | ⌂ ✗ |
| PYoCo (Ge et al., 2023) | Domain Adaptation | 4 | eDiff-I | - | ICCV | 2023 | – ✗ |
| SparseCtrl (Guo et al., 2024a) | Multi-conditional Control | 1 | Stable Diffusion | - | ECCV | 2024 | ⌂ ✗ |
| VideoDirectorGPT (Lin et al., 2024b) | Instruction-followed Fine-tuning | 1 | ModelScopeT2V | 8xA6000 | COLM | 2024 | ⌂ ✗ |
| SimDA (Xing et al., 2024b) | Domain Adaptation | 1 | Stable Diffusion | 8xA100 | CVPR | 2024 | ⌂ ✗ |
| MotionBooth (Wu et al., 2024) | Multi-conditional Control | 1 | Zeroscope LaVie | A100 | NeurIPS | 2024 | ⌂ ✗ |
| CMD (Yu et al., 2024b) | Instruction-followed Fine-tuning | 2 | Stable Diffusion | A100 | ICLR | 2024 | – ✗ |
| Follow-Your-Pose (Ma et al., 2024a) | Multi-conditional Control | 2 | Stable Diffusion | 8xA100 | AAAI | 2024 | ⌂ ✗ |
| VD3D (Bahmani et al., 2025) | Multi-conditional Control | 1 | SnapVideo | 64xA100 (40G) | ICLR | 2024 | ⌂ ✗ |
| VideoStudio (Long et al., 2024) | Multi-conditional Control | 2 | Stable Diffusion | 64xA100 | ECCV | 2024 | ⌂ ✗ |
| DriveDreamer-2 (Zhao et al., 2025b) | Multi-conditional Control | 2 | Stable Diffusion | 8xA800 | AAAI | 2025 | ⌂ ✗ |
| FlipSketch (Bandyopadhyay & Song, 2025) | Multi-conditional Control | 1 | ModelScope | - | CVPR | 2025 | ⌂ ✗ |
| MinT (Wu et al., 2025g) | Instruction-followed Fine-tuning | 1 | OpenSora | A100 | CVPR | 2025 | – ✗ |
| CTRL-Adapter (Lin et al., 2025a) | Domain Adaptation | 1 | I2VGen-XL Stable Video Diffusion Latte Hotshot-XL | A100 | ICLR | 2025 | ⌂ ✗ |
| I2VControl (Feng et al., 2025a) | Multi-conditional Control | 1 | MagicVideo-V2 | - | ICCV | 2025 | – ✗ |
| CustomCrafter (Wu et al., 2025d) | Personalization and Style Adaptation | 1 | VideoCrafter2 | 4xA100 | AAAI | 2025 | ⌂ ✗ |
| TrackGo (Zhou et al., 2025a) | Multi-conditional Control | 1 | Stable Video Diffusion | 8xA100 | AAAI | 2025 | – ✗ |
| Puppet-Master (Li et al., 2025h) | Data Construction and Curation | 1 | Stable Video Diffusion | A6000 | ICCV | 2025 | ⌂ ✗ |
| ReCapture (Zhang et al., 2025a) | Multi-conditional Control | 2 | Stable Video Diffusion | A100 | CVPR | 2025 | – ✗ |
| MagicStick (Ma et al., 2025c) | Multi-conditional Control | 1 | Stable Diffusion | RTX 3090Ti | WACV | 2025 | ⌂ ✗ |
| FACTOR (Huang et al., 2025a) | Multi-conditional Control | 2 | Phenaki | - | WACV | 2025 | – ✗ |
| EDG (Tian et al., 2025a) | Multi-conditional Control | 3 | DynamiCrafter | 8xA100 | CVPR | 2025 | ⌂ ✗ |
| Go-with-the-Flow (Burgert et al., 2025b) | Multi-conditional Control | 1 | Stable Diffusion CogVideoX | 8xA100 | CVPR | 2025 | ⌂ ✗ |
| GS-DiT (Bian et al., 2025) | Multi-conditional Control | 2 | CogVideoX | 8xA100 | CVPR | 2025 | ⌂ ✗ |
| FramePack (Zhang et al., 2025f) | Multi-conditional Control | 1 | HunyuanVideo | 8xA100 | NeurIPS | 2025 | ⌂ ✗ |
| HunyuanPortrait (Xu et al., 2025d) | Personalization and Style Adaptation | 1 | Stable Video Diffusion | 128xA100 | CVPR | 2025 | ⌂ ✗ |
| LCT (Guo et al., 2025b) | Domain Adaptation | 2 | MMDiT | 128xH800 | ICCV | 2025 | – ✗ |
| TASTE-Rob (Zhao et al., 2025c) | Data Construction and Curation | 3 | DynamiCrafter | A6000 | CVPR | 2025 | ⌂ ✗ |
| Phantom (Liu et al., 2025e) | Data Construction and Curation | 2 | MMDiT | A100 | ICCV | 2025 | ⌂ ✗ |
| RealCam-I2V (Li et al., 2025i) | Multi-conditional Control | 1 | DynamiCrafter | - | ICCV | 2025 | ⌂ ✗ |
| VideoREPA (Zhang et al., 2025i) | Domain Adaptation | 1 | CogVideoX | 8xA100 | NeurIPS | 2025 | ⌂ ✗ |
| WISA (Wang et al., 2025h) | Domain Adaptation | 1 | CogVideoX | 8xA100 | NeurIPS | 2025 | ⌂ ✗ |
| DiffPhy (Zhang et al., 2025d) | Domain Adaptation | 1 | Wan2.1 | 4xH100 | ICLR | 2025 | ⌂ ✗ |
| MoAlign (Bhowmik et al., 2026) | Domain Adaptation | 2 | CogVideoX | 4xH100 | ICLR | 2025 | – ✗ |
| TIC-FT (Kim et al., 2025b) | Domain Adaptation | 1 | CogVideoX Wan2.1 | H100 | NeurIPS | 2025 | ⌂ ✗ |
| RoboScape (Shang et al., 2025) | Domain Adaptation | 1 | - | 32xA800 | NeurIPS | 2025 | ⌂ ✗ |
| EchoShot (Wang et al., 2025e) | Data Construction and Curation | 1 | Wan2.1 | A100 | NeurIPS | 2025 | ⌂ ✗ |
| Follow-Your-Creation (Ma et al., 2025e) | Multi-conditional Control | 2 | Wan2.1 | A800 | arXiv | 2025 | – ✗ |
| HunyuanVideo-Avatar (Chen et al., 2025h) | Personalization and Style Adaptation | 2 | HunyuanVideo | 160xA100 (96G) | arXiv | 2025 | ⌂ ✗ |
| LTD (Wu et al., 2026b) | Multi-conditional Control | 1 | Wan2.1 | 8xH20 | ICASSP | 2026 | – ✗ |
| ALIVE (Guo et al., 2026) | Multi-conditional Control | 6 | Waver1.0 | - | arXiv | 2026 | ⌂ ✗ |

# 4 Self-training and Knowledge Distillation Methods

> **Takeaways**
>
> - Self-training and test-time training reuse self-generated outputs or intermediate representations as supervision, enabling iterative refinement under limited or no external annotations, with particular benefits for temporal consistency, controllability, and long-context video generation.
>
> - Knowledge distillation transfers capabilities from large teacher models to more efficient students by encouraging consistency between teacher and student generation behaviors, thereby reducing inference cost while maintaining video quality and temporal coherence.

Self-training and distillation methods refine video generation models by transferring supervision from generated targets, stronger teachers, or auxiliary optimization signals, rather than relying entirely on newly curated human-labeled video data. These methods are particularly valuable when high-quality aligned supervision is limited, costly, or noisy, and when robustness or inference efficiency must be improved without full retraining. Their central advantage is that they provide a scalable path to behavioral refinement by reusing existing models, generated data, or compressed training signals. Accordingly, we organize this family around two broad directions: self-training and test-time adaptation, and knowledge distillation.

## 4.1 Preliminaries: A Unified View of Self-Training and Distillation

Self-training and distillation methods replace direct human-labeled supervision with model-generated targets or teacher guidance. A generic self-training objective can be written as

$$\mathcal{L}_{\text{self}} = \mathbb{E}_{x \sim \mathcal{D}_{\text{src}}} \big[ \ell \big( f_\theta(x), \hat{t}(x) \big) \big], \tag{7}$$

where $\mathcal{D}_{\text{src}}$ denotes unlabeled or weakly labeled inputs used to construct pseudo-supervision, and $\hat{t}(x)$ represents pseudo-targets generated by a teacher model, a stronger generator, or by the model itself (e.g., filtered self-generated samples). In video generation, these pseudo-targets may take the form of generated videos, latent trajectories, denoising targets, or other intermediate supervisory signals.

Knowledge distillation instead trains a student model to match a teacher under

$$\mathcal{L}_{\text{distill}} = \mathbb{E}_{x \sim \mathcal{D}} \big[ d \big( f_\theta(x), f_T(x) \big) \big], \tag{8}$$

where $f_T$ denotes the teacher model, and $d(\cdot, \cdot)$ measures discrepancy between student and teacher outputs, such as distributional divergence, regression losses, or trajectory matching, depending on the underlying generator and distillation strategy.

These formulations provide a unified perspective on self-training and distillation. Self-training improves model behavior by leveraging pseudo-supervision derived from generated targets, while distillation transfers behavior from stronger or more expensive teachers to more efficient students. In both cases, alignment is shaped indirectly through transferred supervision rather than direct evaluative feedback.

## 4.2 Self-training and Test-time Training

Self-training and test-time training improve video generation models by using the model's own outputs, errors, or intermediate representations as supervision. Some methods adapt lightweight parameters during inference, some embed test-time learning directly into temporal modeling modules, and others perform offline self-training on self-generated trajectories or error signals. By turning generation-time feedback into learning signals, this paradigm supports iterative refinement under limited external supervision.

**Inference-time Parameter Adaptation.** A prominent line of work applies test-time training by adapting a small set of model parameters during inference to improve temporal consistency or task-specific performance. Zhang et al. (2025g) propose Zo3T, where a lightweight LoRA adapter is optimized at inference time

together with the manipulated latent state for trajectory-guided image-to-video generation. The adaptation is driven by a regional feature consistency loss that aligns intermediate features across frames while keeping the generation close to the pretrained model's manifold. Zo3T further refines the conditional guidance field through a one-step lookahead strategy, making test-time adaptation part of the denoising process itself rather than a separate post-hoc correction step. CustomTTT (Bi et al., 2025) extends this paradigm to customized video generation by decoupling appearance and action modeling. Separate LoRA adapters are trained for figure and action, and merged at inference time via self-supervised distillation to mitigate conflicts introduced by joint optimization. Similarly, Jeong et al. (2025) adapt LoRA parameters on the input video using pseudo-labels derived from a self-supervised formulation, enabling test-time fine-tuning for video viewpoint transformation.

**Inference-time Temporal Modeling via Test-Time Training.** Beyond parameter-efficient adaptation, test-time training has also been integrated directly into temporal modeling architectures to address long-context video generation. Dalal et al. (2025) propose a hybrid architecture in which test-time training (TTT) layers are embedded as recurrent modules within the model's temporal computation. At inference time, these layers update their internal state by reconstructing low-rank token representations from the preceding layer in a self-supervised manner, so the model can continually compress and carry forward long-range temporal information as generation unfolds. In this sense, the TTT layers function as an adaptive memory mechanism inside the temporal backbone, capturing long-range dependencies without relying on global self-attention or extended attention windows. This formulation reframes test-time training as an integral component of temporal modeling rather than a post-hoc adaptation strategy, enabling efficient long-context video generation without offline fine-tuning.

**Offline Self-training with Self-generated Supervision.** Complementary to test-time adaptation, self-training methods improve video generation models through offline optimization on self-generated signals. VideoAgent (Soni et al., 2025) adopts a rejection-sampling-based self-training framework for embodied control, where successful trajectories collected during real-world robot execution are reused as training data to iteratively refine the video generation model. Before execution, the generated video plans are first refined through self-conditioning consistency using feedback from a pretrained vision-language model, so that self-generated plans can be improved before they are converted into robot actions. This makes self-training operate on both execution outcomes and model-refined video plans, turning successful interaction histories into progressively better supervision for embodied video prediction. SVI (Li et al., 2026a) addresses error accumulation in long video generation through iterative error recycling. The method estimates generation errors by approximating diffusion trajectories with one-step integration, injects these errors into subsequent training inputs, and explicitly trains the model to correct its accumulated deviations. By recycling self-generated errors into supervisory prompts and replayable training signals, SVI turns autoregressive drift into a direct source of supervision for long-horizon video generation.

### 4.3 Knowledge Distillation

Knowledge distillation has been widely adopted in video generation as an effective approach for accelerating inference, transferring or consolidating model capabilities, and improving overall generation quality. Existing work can be broadly categorized by the form of supervision and the role of the teacher model, with diffusion-based video generation as the primary focus.

**Diffusion-to-Autoregressive Distillation.** A line of research distills slow but expressive diffusion-based teacher models into fast autoregressive student models capable of long-horizon video generation. Representative works typically adopt Distribution Matching Distillation (DMD), which aligns the output distributions of teacher and student models to enable efficient generation while preserving visual fidelity. CausVid (Yin et al., 2025) exemplifies this paradigm by distilling a pretrained bidirectional video diffusion model into a causal autoregressive diffusion transformer with key-value caching for streaming inference. To make this asymmetric teacher-student transfer stable, it combines DMD with ODE-based student initialization and uses the stronger bidirectional teacher to supervise the causal student, which helps reduce error accumulation during long-horizon autoregressive generation. Rolling Forcing (Liu et al., 2026b) extends this line of work

Table 2: Summary of self-training and knowledge distillation methods for video generation. Entries are sorted chronologically by publication date. "-" indicates that the corresponding information is not reported or not clearly specified in the original paper.

| Model | Sub-category | Stages | Base Model | GPU | Venue | Year | Link |
|---|---|---|---|---|---|---|---|
| **SF-V (Zhang et al., 2024b)** | Knowledge Distillation | 1 | Stable Video Diffusion | 8xA100 | NeurIPS | 2024 | ⌂ ✗ |
| **CausVid (Yin et al., 2025)** | Knowledge Distillation | 2 | Wan2.1 | - | CVPR | 2025 | ⌂ ✗ |
| **CustomTTT (Bi et al., 2025)** | Self-training and Test-time Training | 3 | CogVideoX | A6000 | AAAI | 2025 | ⌂ ✗ |
| **DOLLAR (Ding et al., 2025)** | Knowledge Distillation | 3 | DiT OpenSora LDM | 8xA100 | ICCV | 2025 | ⌂ ✗ |
| **TDM (Luo et al., 2025)** | Knowledge Distillation | 1 | Stable Diffusion | - | arXiv | 2025 | ⌂ ✗ |
| **Reangle-A-Video (Jeong et al., 2025)** | Self-training and Test-time Training | 2 | CogVideoX | - | ICCV | 2025 | ⌂ ✗ |
| **One-Minute Video (Dalal et al., 2025)** | Self-training and Test-time Training | 1 | CogVideoX | 256xH100 | CVPR | 2025 | ⌂ ✗ |
| **Self Forcing (Huang et al., 2025c)** | Knowledge Distillation | 1 | Wan2.1 | 64xH100 | NeurIPS | 2025 | – ✗ |
| **NFD (Cheng et al., 2025e)** | Knowledge Distillation | 3 | - | A100 | arXiv | 2025 | – ✗ |
| **ADM (Lu et al., 2025)** | Knowledge Distillation | 1 | CogVideoX Stable Diffusion XL Stable Diffusion3 | - | ICCV | 2025 | – ✗ |
| **V.I.P. (Kim et al., 2025a)** | Knowledge Distillation | 1 | VideoCrafter2 AnimateDiff | 4xA100 | ICCV | 2025 | – ✗ |
| **SwiftVideo (Sun et al., 2025c)** | Knowledge Distillation | 3 | Wan2.1 | 8xA100 | arXiv | 2025 | – ✗ |
| **V-PAE (Cheng et al., 2025b)** | Knowledge Distillation | 2 | Wan2.1 | 32xH20 | arXiv | 2025 | – ✗ |
| **Zo3T (Zhang et al., 2025g)** | Self-training and Test-time Training | – | Stable Video Diffusion | A100 | arXiv | 2025 | – ✗ |
| **SVI (Li et al., 2026a)** | Self-training and Test-time Training | 1 | Wan2.1 | - | arXiv | 2025 | ⌂ ✗ |
| **Neodragon (Karnewar et al., 2026)** | Knowledge Distillation | 4 | Pyramidal Flow DiT | H100 | arXiv | 2025 | ⌂ ✗ |
| **VideoTPO (Chen et al., 2025b)** | Self-training and Test-time Training | – | Wan2.1 Kling | - | arXiv | 2025 | ⌂ ✗ |
| **MoGAN (Xue et al., 2025)** | Knowledge Distillation | 2 | Wan2.1 | 16xH200 | arXiv | 2025 | – ✗ |
| **Causal Forcing (Zhu et al., 2026)** | Knowledge Distillation | 3 | Wan2.1 | H100 | arXiv | 2026 | ⌂ ✗ |
| **EchoTorrent (Meng et al., 2026)** | Knowledge Distillation | 4 | InfiniteTalk | 64xA100 | arXiv | 2026 | – ✗ |
| **AMD (Bai et al., 2026)** | Knowledge Distillation | 2 | Wan2.1 | 8xH800 | arXiv | 2026 | – ✗ |

toward real-time long-video streaming by jointly denoising multiple consecutive frames with progressively increasing noise levels, instead of sampling one frame at a time. It further introduces an attention sink mechanism for long-range consistency and an extended-window few-step distillation algorithm over non-overlapping windows, reducing exposure bias under self-generated histories. By compressing iterative diffusion sampling into a small number of autoregressive steps, these methods substantially reduce inference latency without retraining models from scratch.

**Trajectory-Level and Continuous-Time Distillation.** Unlike distillation methods that match distributions only at the final state, a line of work provides denser supervision by aligning diffusion trajectories over time. SwiftVideo (Sun et al., 2025c) introduces Continuous-Time Consistency Distillation (CCD) based on a flow-matching formulation, directly aligning the velocity fields predicted by the teacher and student models at each timestep. In this formulation, the teacher velocity serves as the primary supervision signal, enabling strong temporal consistency under few-step sampling. In parallel, Luo et al. (2025) extend distribution matching from single-step alignment to trajectory-level consistency by enforcing alignment at multiple intermediate diffusion states, which reduces error accumulation and stabilizes long-horizon generation. rCM (Zheng et al., 2026) also targets continuous-time modeling but adopts a different supervision role assignment, treating student self-consistency as the primary objective and introducing teacher score (velocity) distillation as an auxiliary regularizer rather than a timestep-wise regression target, thereby preserving generation diversity while mitigating error accumulation and detail degradation in few-step regimes.

**Adversarial and Hybrid Distillation Objectives.** Beyond distribution matching and consistency-based objectives, another line of work augments diffusion model distillation with adversarial learning to improve generation quality and training stability. In this paradigm, a discriminator provides additional supervision by

distinguishing between teacher- and student-generated predictions or representations. SF-V (Zhang et al., 2024b) fine-tunes a student initialized from the teacher model and employs a discriminator built upon a frozen teacher encoder with trainable spatial and temporal heads to assess generation quality. Similarly, NFD (Cheng et al., 2025e) introduces adversarial supervision after a score-consistency-based warm-up phase, using a discriminator initialized from the teacher model. More recent approaches further combine adversarial learning with consistency objectives to mitigate error accumulation and distribution mismatch. For example, works by Cheng et al. (2025b) and Xue et al. (2025) integrate GAN-based supervision into DMD, with different emphases on overall distributional quality and motion dynamics. DOLLAR (Ding et al., 2025) adopts a hybrid formulation that combines distribution matching, consistency-based distillation, and latent reward optimization to alleviate mode collapse and fidelity degradation in few-step video generation.

**Training Strategies for Distillation.** Beyond architectural and objective-level innovations, training data organization and distillation-related strategies also play an important role in scalable video generation. Seedance 1.0 (Gao et al., 2025d) adopts a progressive training scheme that gradually increases video resolution and temporal complexity, facilitating more stable optimization. Other works modify the distillation process itself; for example,self-forcing methods mitigate exposure bias by conditioning training on self-generated histories (Liu et al., 2026b; Kim et al., 2025a). In addition, text encoder distillation has emerged as a complementary technique that significantly reduces model size and inference cost while preserving semantic alignment (Karnewar et al., 2026). Collectively, these studies highlight the importance of coordinated distillation objectives, data curricula, and auxiliary supervision in efficient video generation.

## 5 Preference- and Reward-based Methods

**Takeaways**

- Reinforcement learning aligns video generation by explicitly modeling generation as a long-horizon decision process, enabling the enforcement of temporal consistency, physical plausibility, and structured constraints through trajectory-level optimization.

- Preference-based optimization directly optimizes relative comparisons between generated videos, with recent advances introducing temporally structured, physically grounded, and stability-aware preference objectives tailored to diffusion-based video models.

- Video reward modeling underpins both reinforcement learning and preference-based alignment by decomposing human preferences into multi-dimensional, identity-aware, and physics- or reasoning-aware signals that capture video-specific quality beyond frame-level appearance.

Preference- and reward-based methods align video generation models by optimizing evaluative signals that reflect behavioral correctness, rather than relying solely on fixed supervised targets such as supervised reconstruction or transferred teacher outputs. They are especially useful when alignment involves competing objectives, such as semantic fidelity, temporal coherence, physical plausibility, identity consistency, and safety, that are difficult to balance through direct supervision alone. Their key advantage is that they express alignment in terms of comparative or outcome-level feedback, providing a more direct mechanism for steering pretrained generators toward desired behaviors. We therefore organize this section around three main paradigms: reinforcement learning, preference-based optimization, and video reward modeling.

### 5.1 Preliminaries: Optimization Paradigms

Three optimization paradigms are representative in preference-based and reinforcement learning for video generation models: Proximal Policy Optimization (PPO), Direct Preference Optimization (DPO), and Group Relative Policy Optimization (GRPO). These paradigms form the methodological basis for the methods reviewed in this section. Although they originate from language modeling and image generation, we present them in a unified formulation applicable to both autoregressive and diffusion-based video generation models.

We use $x$ to denote the multimodal conditioning signal (e.g., text, images, or control inputs), $y$ to denote a generated video or its latent representation, and $\tau$ to denote a generation trajectory. For autoregressive models, $\tau$ denotes a sequence of generated tokens or video tokens. For diffusion-based generators, $\tau$ denotes the denoising trajectory over latent states. In this case, we interpret each denoising step as an action and define the trajectory likelihood $\log \pi_\theta(\tau \mid x)$ as the sum of step-wise conditional log-densities (or a training-time surrogate), since the marginal likelihood of the final generated video is generally intractable.

**PPO-style Reinforcement Learning (RLHF and RLAIF).** Reinforcement Learning with Human Feedback (RLHF) (Ouyang et al., 2022) aligns a generative policy by first training a reward model (RM) and then optimizing the policy using PPO (Schulman et al., 2017) under a constraint that limits deviation from a reference model $\pi_{\mathrm{ref}}$ (e.g., an SFT or pretrained model). The reward model is typically trained on preference pairs $(x, y^+, y^-)$ using a Bradley–Terry objective (Bradley & Terry, 1952),

$$\mathcal{L}_{\mathrm{RM}}(\phi) = -\mathbb{E}_{(x, y^+, y^-)} \log \sigma\big(r_\phi(x, y^+) - r_\phi(x, y^-)\big),\tag{9}$$

where $r_\phi(x, y)$ denotes a scalar reward and $\sigma(\cdot)$ is the logistic function. Given a fixed reward model, PPO optimizes the policy by maximizing a clipped policy-gradient objective augmented with a KL regularization term relative to the reference policy. Let $r_t(\theta) = \frac{\pi_\theta(a_t \mid x, \tau_{<t})}{\pi_{\theta_{\mathrm{old}}}(a_t \mid x, \tau_{<t})}$ denote the probability ratio, where $a_t$ denotes the step-wise action under the current generation parameterization. In autoregressive models, it corresponds to a token-level generation decision, whereas in diffusion-based models it corresponds to a denoising or latent-transition decision. Let $\hat{A}_t$ denote an advantage estimator, commonly implemented by broadcasting a sequence- or trajectory-level reward across individual steps. The PPO objective is

$$\mathcal{L}_{\mathrm{PPO}}(\theta) = -\mathbb{E}\left[\sum_t \min\big(r_t(\theta)\,\hat{A}_t,\ \mathrm{clip}\big(r_t(\theta),\, 1-\epsilon,\, 1+\epsilon\big)\,\hat{A}_t\big)\right] + \beta\,\mathrm{KL}(\pi_\theta(\cdot|x)\,\|\,\pi_{\mathrm{ref}}(\cdot|x)).\tag{10}$$

Reinforcement Learning with AI Feedback (RLAIF) (Bai et al., 2022) follows the same optimization procedure but replaces human annotations with AI-generated rewards or preferences. Although PPO-style reinforcement learning provides a principled framework for trajectory-level credit assignment, explicit RLHF or RLAIF is relatively uncommon in video generation due to the difficulty of designing stable and dense reward signals over high-dimensional, long-horizon video trajectories. In video generation, rewards are often computed at the clip or trajectory level, while optimization is performed over many intermediate generation steps, making credit assignment substantially harder than in short-form text generation.

**Direct Preference Optimization (DPO).** Direct Preference Optimization (DPO) (Rafailov et al., 2023) eliminates the need for an explicit reward model by directly optimizing the policy to match observed preferences relative to a fixed reference policy. Given preference pairs $(x, y^+, y^-)$ and a temperature parameter $\beta > 0$, the DPO objective is

$$\mathcal{L}_{\mathrm{DPO}}(\theta) = -\mathbb{E}\log\sigma\Big(\beta\big[\log \pi_\theta(y^+|x) - \log \pi_{\mathrm{ref}}(y^+|x) - \log \pi_\theta(y^-|x) + \log \pi_{\mathrm{ref}}(y^-|x)\big]\Big).\tag{11}$$

For autoregressive generators, the log-probability terms are standard sequence likelihoods. For diffusion-based video generators, the same preference optimization idea is usually applied through trajectory-level or denoising-based objectives, because the likelihood of the final generated video is not directly tractable.

This formulation can be interpreted as implicitly inducing a reward proportional to the log-probability ratio between the policy and the reference model, thereby combining preference alignment and KL regularization into a single contrastive objective. DPO-style optimization has proven particularly attractive for video generation models, where training high-quality reward models is challenging, and preference supervision can be applied at varying temporal granularities.

**Group Relative Policy Optimization (GRPO).** Group Relative Policy Optimization (GRPO) (Shao et al., 2024) provides an alternative alignment paradigm that replaces learned rewards or explicit preference pairs with verifiable outcome-level signals. For a given conditioning input $x$, GRPO samples a group of $K$

trajectories $\{\tau^{(k)}\}_{k=1}^{K}$ from the current policy $\pi_{\theta_{\mathrm{old}}}$ and evaluates each trajectory using a verifiable scoring function $r^{(k)} \in [0,1]$, such as prompt-following checks, temporal consistency tests, identity-preservation heuristics, motion smoothness criteria, or task-specific physical constraints. A group baseline $\bar{r} = \frac{1}{K} \sum_{j=1}^{K} r^{(j)}$ is computed, and group-relative advantages are defined as

$$A^{(k)} = r^{(k)} - \mathrm{stopgrad}(\bar{r}), \qquad \ell^{(k)}(\theta) = \sum_{t \in \tau^{(k)}} \log \pi_\theta(a_t \mid x, \tau_{<t}). \tag{12}$$

The GRPO objective is then

$$\mathcal{L}_{\mathrm{GRPO}}(\theta) = -\frac{1}{K} \sum_{k=1}^{K} A^{(k)} \, \ell^{(k)}(\theta) \; + \; \beta \, \mathrm{KL}(\pi_\theta(\cdot|x) \, \| \, \pi_{\mathrm{ref}}(\cdot|x)). \tag{13}$$

By relying on relative comparisons within sampled group, GRPO avoids explicit reward modeling and reduces sensitivity to absolute score calibration. This property is particularly appealing for video generation, where designing reliable scalar rewards is hard, but outcome-level verification or heuristic constraints are available.

## 5.2 Reinforcement Learning for Video Generation

Reinforcement learning (RL) aligns video generation models by treating generation as a sequential decision-making process optimized under long-horizon video-level objectives. Compared to supervised post-training and preference-based optimization, RL explicitly models the interaction between generation actions and delayed rewards, making it well-suited to enforcing temporal consistency, physical plausibility, and structured constraints. Existing approaches apply RL at different levels of the generation pipeline, ranging from system-level alignment to diffusion-level optimization and constraint-aware training strategies.

**Reinforcement Learning as an End-to-End Alignment Framework.** Several works apply RL as an end-to-end alignment framework at the system level, treating the entire video generation pipeline as a policy optimized with respect to long-horizon video-level objectives (Liu et al., 2025c; Arkhipkin et al., 2025; Zhong et al., 2026; Wang et al., 2026a). In this setting, RL serves as an end-to-end post-training mechanism that directly aligns model behavior beyond supervised fine-tuning. VANS (Cheng et al., 2025c) exemplifies this paradigm by jointly aligning a vision-language model and a video diffusion model for video next-event prediction. Its Joint-GRPO strategy optimizes both components under a shared reward, encouraging the VLM to produce semantically accurate captions that are well suited for downstream video generation, while guiding the video diffusion model to generate videos faithful to those captions and the input visual context. This formulation makes RL operate at the level of the full reasoning-to-generation pipeline. Similarly, Seedance 1.0 (Gao et al., 2025d) incorporates video-specific RL from human feedback as a system-level alignment component, directly maximizing multi-dimensional reward signals to jointly improve prompt adherence, motion plausibility, and visual fidelity in large-scale video generation. RLIR (Ye et al., 2025b) provides a more explicit sequential decision-making formulation by recovering verifiable reward signals from generated videos with an inverse dynamics model. By mapping high-dimensional video outputs into a lower-dimensional action space, it constructs objective rewards for optimization via GRPO and shows that end-to-end RL can be applied cleanly when suitable action-reward representations are available.

**Reinforcement Learning for Optimizing the Generation Process.** Beyond end-to-end alignment, another line of work applies RL directly to the video generation process itself, intervening at the level of diffusion sampling and generation trajectories (Zhao et al., 2025d; Yan et al., 2025). Instead of treating RL solely as a high-level post-training objective, these methods integrate RL signals into intermediate stages of generation, enabling fine-grained control over temporal dynamics and physically grounded motion. Phys-AR (Lin et al., 2025c) provides a representative example by reformulating diffusion-based video generation as a token-level sequential decision process by introducing diffusion timestep tokens that explicitly represent evolving physical states. Its core idea is to recover recursive visual tokens during diffusion and use them to support symbolic reasoning over physical conditions, so that reinforcement learning can optimize the resulting reasoning trajectories under rule-based physical rewards. This design allows the model to enforce motion

consistency and generalize to out-of-distribution physical settings such as unseen velocities or accelerations. CamVerse (Wang et al., 2025u) applies the same process-level RL perspective to camera-controlled video generation. It treats the video diffusion model as a stochastic policy and introduces a verifiable geometry reward that estimates 3D camera trajectories for generated and reference videos, computes segment-wise relative poses, and provides dense feedback on camera-trajectory alignment. This reward design makes online RL effective for optimizing geometrically consistent and controllable camera motion throughout sampling.

**Reinforcement Learning for Stability, Efficiency, and Structured Constraints.** Applying RL to video generation poses challenges in training stability, controllability, and enforcing task-specific constraints. Recent work extends RL beyond generic policy optimization through curriculum-style training and structured objectives that improve robustness (Pan et al., 2025a; Shen et al., 2025b). To improve optimization stability, Self-Paced GRPO (Li et al., 2025g) proposes a competence-aware RL framework in which reward supervision co-evolves with the generator. By progressively shifting the reward function's emphasis from coarse visual quality to temporal coherence and semantic alignment, self-paced GRPO mitigates reward saturation and stabilizes long-horizon policy optimization. Beyond training dynamics, RL has also been used to impose structured constraints. PhysMaster (Ji et al., 2025a) adopts a top-down strategy that learns a physics-aware representation from the input image as an explicit conditioning signal for image-to-video generation. It optimizes a dedicated PhysEncoder according to the physical plausibility of the final generated videos, using preference-based RL to improve how physical cues are extracted and injected into the generation process. Identity-GRPO (Meng et al., 2025c) targets a different structured objective, namely multi-human identity consistency in dynamic interaction videos. It combines a reward model trained on preference data focused on human consistency with a GRPO variant tailored to multi-human generation, allowing RL to optimize identity preservation under complex spatial-temporal interactions.

### 5.3 Preference-based Optimization for Video Alignment

Preference-based optimization aligns video generation models by directly optimizing relative-preference objectives over generated samples. Unlike reinforcement learning, which requires complex trajectory-level credit assignment, these methods rely on simpler pairwise or relative comparisons to shape model behavior, making them particularly suitable for high-dimensional video diffusion models. Recent work adapts DPO and its variants to the video domain by designing scalable mechanisms for constructing reliable preference signals under limited or fully automated supervision.

**Preference-based Optimization as a Direct Alignment Objective.** A growing line of work formulates video alignment as direct optimization over preference signals, avoiding explicit reward modeling and policy-based reinforcement learning (Xu et al., 2025b; Ji et al., 2025b; Yuan et al., 2024a). In practice, these methods differ mainly in how they construct reliable preference pairs for training. VideoDPO (Liu et al., 2025h) provides a representative example by adapting DPO to video diffusion models with an automatic preference-pair construction pipeline. It introduces OmniScore, a multi-dimensional scoring function that jointly evaluates visual quality and text-video semantic alignment, then ranks multiple generated videos for each prompt to form winner-loser pairs. VideoDPO further reweights these pairs according to score gaps, so that clearer preference distinctions contribute more strongly during optimization. DF-DPO (Cheng et al., 2025a) constructs preference pairs in a different way by using real videos as winning samples and edited counterparts with explicit temporal or spatial artifacts as losing samples, which removes the ambiguity of comparing multiple generated outputs and provides scalable supervision without an additional discriminator. SePPO (Zhang et al., 2024a) extends this line through a semi-policy framework that uses historical model checkpoints as reference policies for preference construction. Its anchor-based adaptive flipper stabilizes optimization by checking whether the reference sample is actually worse than the current model output before assigning the preference direction.

**Fine-Grained and Structured Preference Supervision.** Beyond video-level binary preferences, effective preference-based alignment for video generation requires structuring preference signals across finer temporal and semantic dimensions (Wang et al., 2025j; Qian et al., 2025; Li et al., 2025a; Kupyn et al., 2025; Liu et al., 2025b; Cai et al., 2025b). DenseDPO (Wu et al., 2025f) addresses the motion bias of vanilla DPO by

constructing structurally aligned video pairs from partially noised real videos and collecting segment-level preference labels, enabling dense temporal supervision that localizes artifacts while preserving global motion dynamics. Vanilla DPO pairs generated from independent noise seeds exhibit large motion differences, causing annotators to favor artifact-free slow-motion clips; DenseDPO neutralizes this bias by denoising corrupted copies of a single real reference video so that both videos share motion structure while differing only in local details. AlignHuman (Liang et al., 2025a) structures preference supervision along the denoising timeline by exploiting the observation that early timesteps mainly govern motion dynamics, while later timesteps more strongly affect fidelity and human structure. Based on this decomposition, it proposes timestep-segment preference optimization, partitions preference data across denoising intervals, and trains two specialized LoRA experts that are activated in their corresponding timestep ranges during inference. This divide-and-conquer design allows preference optimization to target motion naturalness and visual fidelity separately, reducing the tension between these objectives in human animation. Beyond temporal structuring, PhysHPO (Chen et al., 2025a) generalizes fine-grained preference optimization by organizing preferences across hierarchical semantic levels, including instance, state, and motion. Its hierarchical cross-modal DPO objective aligns each level with a corresponding aspect of physical plausibility, so that supervision is no longer concentrated only on surface appearance or global video-level judgments. PhysHPO further couples this hierarchical preference design with an automated data-selection pipeline that identifies high-quality real videos from large-scale text-video corpora, providing scalable supervision for physically plausible video generation.

**Stability, Efficiency, and Hybrid Preference Optimization.** While preference objectives effectively guide alignment, applying them to video diffusion models often introduces instability, high cost, and scalability issues; recent work therefore focuses on more robust and efficient optimization (Liu et al., 2025g; Zhu et al., 2025; Wang et al., 2025c). BranchGRPO (Li et al., 2026b) improves efficiency and stability by restructuring GRPO rollouts into a branching tree with shared prefixes, depth-wise reward fusion, and pruning. This design amortizes computation across trajectories that share early sampling paths, while tree-based advantage estimation provides denser process-level supervision under sparse rewards. Therefore, BranchGRPO reduces rollout cost and stabilizes optimization for preference alignment in diffusion-based generation. DPP-GRPO (Kazimi et al., 2025) extends preference optimization from individual samples to candidate sets by incorporating a Determinantal Point Process term into GRPO. The DPP term imposes diminishing returns on redundant samples, making diversity an explicit alignment objective over multiple generated videos for the same prompt. This allows the model to cover a broader range of plausible video outcomes while maintaining prompt fidelity and perceptual quality.

### 5.4   Video Reward Modeling

Effective alignment of video generation models critically depends on the availability of reliable reward signals that reflect human preferences. Unlike images or text, video reward modeling must account for high-dimensional factors such as temporal dynamics, motion consistency, and long-range coherence, which substantially increase the difficulty of reward design. Recent work on video reward modeling not only expands the aspects of video quality being evaluated, but also improves the robustness of reward design itself. Representative directions include mitigating reward misspecification, multi-dimensional quality assessment, identity and temporal consistency, and physics- and reasoning-aware evaluation.

**Reward Hacking and Reward Misspecification.** A fundamental challenge in video reward modeling is reward hacking, where optimizing a learned reward or proxy metric improves the target score without yielding proportional gains in actual alignment quality (Skalse et al., 2022; Karwowski et al., 2024). This issue is especially pronounced in video generation because reward signals must capture multiple competing objectives over long and high-dimensional trajectories Liang et al. (2026). As a result, overly coarse or imperfect reward functions can encourage narrow reward-aligned behaviors, such as exaggerated motion, over-smoothed dynamics, or local improvements that mask failures elsewhere in the video. Recent work addresses this problem by making reward design more robust and temporally informative (Li et al., 2025g; Deng et al., 2026a; Yin et al., 2026). DenseGRPO (Deng et al., 2026a) addresses a related misspecification, the sparse reward problem, where a single terminal reward is broadcast uniformly to all denoising steps despite each step's fine-grained contribution varying; it estimates step-wise reward gains via ODE-denoising of intermediate

latents, providing dense per-step feedback that closes this feedback contribution mismatch and avoids reward exploitation at intermediate timesteps. Furthermore, SoliReward Lian et al. (2025) improves reward-model training with lower-noise annotations and regularized preference learning to reduce susceptibility to reward hacking, while Diffusion-DRF Wang et al. (2026d) replaces single scalar feedback with aspect-structured, multi-dimensional reward signals derived from a frozen vision-language critic.

**Multi-Dimensional Video Quality Assessment.** A central direction in video reward modeling is to decompose human preference into multiple quality dimensions, recognizing that video alignment cannot be captured by a single scalar score (Gao et al., 2025d; Bao et al., 2025). VideoReward (Liu et al., 2025c) establishes a large-scale, human-annotated preference dataset over modern video generation models and trains a multi-dimensional reward model that separately evaluates visual quality, motion quality, and text-video alignment. By explicitly modeling these dimensions under a Bradley–Terry-with-ties formulation, VideoReward provides a robust reward backbone for preference-based and reinforcement learning alignment in video generation. While VideoReward targets open-domain video generation, AnimeReward (Zhu et al., 2025) shows that generic video reward models fail to capture domain-specific quality criteria in anime generation, particularly appearance stylization and character consistency. To address this gap, AnimeReward constructs the first anime-specific multi-dimensional reward dataset and employs specialized vision-language models for different evaluation dimensions, demonstrating that domain-aware reward decomposition is critical for aligning stylized video generation with human preferences.

**Identity, Consistency, and Temporal Coherence Rewards.** Beyond overall quality assessment, a central challenge in video generation is preserving subject identity and maintaining coherent appearance and motion over time, motivating reward designs that explicitly target video-specific consistency failures. One important direction uses identity-preserving rewards to maintain subject consistency under large pose, expression, and motion changes. PersonalVideo (Li et al., 2025b) follows this direction by combining an Identity Consistency Reward with a complementary Semantic Consistency Reward. The identity reward evaluates whether generated frames preserve the reference identity, while the semantic reward constrains the semantic distribution of generated videos to remain aligned with the original text-to-video model, helping preserve dynamic behavior and semantic faithfulness during identity injection. IPRO (Shen et al., 2025b) formulates identity preservation as direct optimization with a differentiable facial identity reward. It backpropagates the reward signal through the final denoising steps of the diffusion process and further stabilizes optimization with KL regularization against the base model, which helps suppress identity drift across frames while maintaining temporal coherence. A second direction focuses more directly on temporal controllability. Along this direction, AR-Drag (Zhao et al., 2025d) introduces a trajectory-based reward model that explicitly evaluates motion paths in autoregressive generation. This reward provides fine-grained supervision over temporal dynamics and controllability, enabling stable, coherent motion generation in long-horizon, few-step autoregressive-controlled diffusion models.

**Physics- and Reasoning-Aware Reward Modeling.** Beyond perceptual quality and temporal consistency, recent work explores reward designs that explicitly encode physical laws and reasoning structure, aiming to align video generation with objective physical plausibility rather than subjective visual cues. These methods differ mainly in the source of physical supervision, including verifiable physical proxies, learned physics reward models, process-aware latent evaluation, and geometry-based consistency signals. NewtonRewards (Le et al., 2025) represents the first direction by introducing a physics-grounded post-training framework based on verifiable rewards. It extracts measurable proxies from generated videos using frozen utility models, with optical flow serving as a proxy for velocity and high-level appearance features serving as a proxy for mass, and uses them to enforce Newtonian kinematic constraints and mass conservation. Similarly, PhysCorr (Wang et al., 2025j) follows a learned-reward approach through PhysicsRM, a dual-dimensional physics reward model that jointly evaluates intra-object stability and inter-object interactions, providing structured assessment of physical consistency beyond frame-level aesthetics. VIGOR (Yin et al., 2026) introduces a geometry-based reward that evaluates multi-view consistency through cross-frame pointwise reprojection error computed with a pretrained geometric foundation model. By focusing on geometrically meaningful correspondences, it targets artifacts such as object deformation, spatial drift, and depth violations that are difficult to capture with purely perceptual rewards.

Table 3: Summary of preference-based and reinforcement learning methods for video generation. Entries are sorted chronologically by publication date. "-" indicates that the corresponding information is not reported or not clearly specified in the original paper.

| Model | Sub-category | Stages | Base Model | GPU | Venue | Year | Link |
|---|---|---|---|---|---|---|---|
| InstructVideo (Yuan et al., 2024a) | Preference-based Optimization | 1 | ModelScopeT2V | 4xA100 | CVPR | 2024 | ☉ ✗ |
| T2V-Turbo (Li et al., 2024) | Preference-based Optimization | 1 | VideoCrafter2 ModelScopeT2V | 8xA100 | NeurIPS | 2024 | ☉ ✗ |
| VADER (Prabhudesai et al., 2024) | Preference-based Optimization | 1 | VideoCrafter OpenSora ModelScopeT2V Stable Video Diffusion | 2xA6000 | arXiv | 2024 | ☉ ✗ |
| Prompt-A-Video (Ji et al., 2025b) | Preference-based Optimization | 2 | OpenSora CogVideoX | - | ICCV | 2024 | ☉ ✗ |
| PersonalVideo (Li et al., 2025b) | Video Reward Modeling | 1 | HunyuanVideo AnimateDiff | A800 | ICCV | 2025 | ☉ ✗ |
| VideoDPO (Liu et al., 2025h) | Preference-based Optimization | – | VideoCrafter2 T2V-Turbo CogVideo | 4xA100 | CVPR | 2025 | ☉ ✗ |
| VideoReward (Liu et al., 2025c) | Video Reward Modeling | 3 | – | 8xA800 | NeurIPS | 2025 | ☉ ✗ |
| MagicID (Li et al., 2025a) | Preference-based Optimization | 1 | HunyuanVideo | H100 | ICCV | 2025 | ☉ ✗ |
| DF-DPO (Cheng et al., 2025a) | Preference-based Optimization | 1 | CogVideoX | 8xH100 | arXiv | 2025 | – ✗ |
| AnimeReward (Zhu et al., 2025) | Video Reward Modeling | 3 | CogVideoX | 8xA800 | arXiv | 2025 | ☉ ✗ |
| Phys-AR (Lin et al., 2025c) | Reinforcement Learning | 3 | Llama3.1 | 32xA800 | arXiv | 2025 | – ✗ |
| DiffusionNPO (Wang et al., 2025c) | Preference-based Optimization | 1 | VideoCrafter2 | - | ICLR | 2025 | ☉ ✗ |
| DenseDPO (Wu et al., 2025f) | Preference-based Optimization | 1 | MAGVIT-v2 | 64xA100 | NeurIPS | 2025 | – ✗ |
| Seedance 1.0 (Gao et al., 2025d) | Reinforcement Learning | 4 | DiT | - | arXiv | 2025 | – ✗ |
| AlignHuman (Liang et al., 2025a) | Preference-based Optimization | 3 | MMDiT | - | arXiv | 2025 | – ✗ |
| RDPO (Qian et al., 2025) | Preference-based Optimization | 3 | LTX-Video | 32xH100 | arXiv | 2025 | – ✗ |
| BranchGRPO (Li et al., 2026b) | Preference-based Optimization | 1 | FLUX.1-Dev Wan2.1 | 16xH200 | arXiv | 2025 | ☉ ✗ |
| RLGF (Yan et al., 2025) | Reinforcement Learning | 1 | MagicDrive-V2 | 8xA100 | NeurIPS | 2025 | – ✗ |
| PhysMaster (Ji et al., 2025a) | Reinforcement Learning | 3 | DiT | 8xA800 | arXiv | 2025 | ☉ ✗ |
| IdentityGRPO (Meng et al., 2025c) | Reinforcement Learning | 2 | VACE | 8xA100 | arXiv | 2025 | ☉ ✗ |
| Epipolar-DPO (Kupyn et al., 2025) | Preference-based Optimization | 1 | Wan2.1 | 4xA6000 | arXiv | 2025 | ☉ ✗ |
| PhysCorr (Wang et al., 2025j) | Preference-based Optimization | 2 | Wan2.1 | 4xA800 | arXiv | 2025 | – ✗ |
| Ar-Drag (Zhao et al., 2025d) | Video Reward Modeling | 2 | Wan2.1 | 8xH200 | arXiv | 2025 | – ✗ |
| McSc (Yang et al., 2025a) | Reinforcement Learning | 3 | VideoCrafter2 Wan2.1 | 8xA100 | arXiv | 2025 | ☉ ✗ |
| ID-Crafter (Pan et al., 2025a) | Reinforcement Learning | 1 | Wan | 16xH20 | arXiv | 2025 | ☉ ✗ |
| BPGO (Liu et al., 2025g) | Preference-based Optimization | 1 | Wan2.1 Wan2.2 | 16xH100 | arXiv | 2025 | – ✗ |
| PRFL (Mi et al., 2025) | Video Reward Modeling | 2 | Wan2.1 | - | arXiv | 2025 | – ✗ |
| DPP-GRPO (Kazimi et al., 2025) | Preference-based Optimization | 2 | Wan2.1 CogVideoX | 4xL40S | arXiv | 2025 | – ✗ |
| Self-paced GRPO (Li et al., 2025g) | Reinforcement Learning | 1 | Wan2.1 HunyuanVideo | 16xH100 | arXiv | 2025 | – ✗ |
| NewtonRewards (Le et al., 2025) | Video Reward Modeling | 2 | OpenSora | 8xH100 | arXiv | 2025 | ☉ ✗ |
| IC-World (Wu et al., 2025a) | Reinforcement Learning | 2 | Wan2.1 | 8xH20 | arXiv | 2025 | ☉ ✗ |
| CamVerse (Wang et al., 2025u) | Reinforcement Learning | 2 | – | 32xH200 | arXiv | 2025 | – ✗ |
| DreaMontage (Liu et al., 2025b) | Preference-based Optimization | 3 | Seedance 1.0 | - | arXiv | 2025 | – ✗ |
| Euphonium (Zhong et al., 2026) | Reinforcement Learning | 1 | HunyuanVideo | 40xH800 | arXiv | 2026 | ☉ ✗ |
| HuDA (Ashutosh et al., 2026) | Preference-based Optimization | 1 | Wan2.1 | 32xH100 | arXiv | 2026 | – ✗ |
| PhysRVG (Zhang et al., 2026b) | Preference-based Optimization | 2 | Wan2.2 | 32xH20 | arXiv | 2026 | – ✗ |
| GT-SVJ (Shekhar et al., 2026) | Preference-based Optimization | 2 | CogVideoX | - | arXiv | 2026 | – ✗ |
| LocalDPO (Huang et al., 2026) | Preference-based Optimization | 1 | CogVideoX Wan2.1 | - | arXiv | 2026 | – ✗ |
| PISCES (Le et al., 2026) | Preference-based Optimization | 2 | VideoCrafter2 HunyuanVideo | 8xA100 | arXiv | 2026 | – ✗ |

# 6 Inference-Time Methods

**Takeaways**

- Inference-time alignment operationalizes post-trained signals by steering video generation during sampling, allowing alignment objectives to be enforced without further parameter updates.

- Guidance-based methods modify denoising trajectories using learned alignment signals or auxiliary models to control semantics, structure, motion, and physical plausibility while preserving the pretrained generative prior.

- Iterative refinement and self-editing regulate video generation through inference-time feedback loops or multi-stage refinement, enabling error correction, long-horizon consistency, and fine-grained control via closed-loop inference alone.

While Sections 3–5 examine alignment through post-training, alignment also extends to inference. Post-training produces alignment artifacts, such as reward models, learned critics, guidance modules, and specialized adapters, whose effects are realized during generation. At inference time, these signals steer, constrain, or refine video generation without further parameter updates. This is particularly important when residual temporal, physical, or semantic errors remain after training, or when user-specific control requirements and deployment constraints make additional retraining impractical. Rather than introducing new objectives, such mechanisms determine how post-trained signals are consumed during sampling, shaping trajectories, enforcing semantic or physical constraints, and regulating trade-offs among alignment goals.

## 6.1 Guidance-based Alignment

Guidance-based alignment directs the video generation process by injecting auxiliary control signals into the denoising trajectory at inference time. This approach influences generation towards specified semantics, structures, or dynamics without modifying the underlying model parameters. Guidance-based methods provide careful control over objects, motion, and style by shaping intermediate latent states throughout the denoising process, while maintaining the generative prior of the pretrained model.

A basic example is classifier-free guidance (CFG), a standard inference-time mechanism in diffusion models that strengthens conditional generation by combining conditional and unconditional denoising predictions (Ho & Salimans, 2022). CFG steers the generation process by extrapolating the difference between a conditionally generated output and an unconditionally generated one. Formally, at a given diffusion timestep $t$, let $\epsilon_\theta(x_t, c)$ represent the model's noise prediction conditioned on a signal $c$ (e.g., a text prompt), and $\epsilon_\theta(x_t, \varnothing)$ represent the unconditional (null) noise prediction. The guided noise prediction, $\hat{\epsilon}_{\text{cfg}}(x_t, c)$, is computed as:

$$\hat{\epsilon}_{\text{cfg}}(x_t, c) = \epsilon_\theta(x_t, \varnothing) + w\Big(\epsilon_\theta(x_t, c) - \epsilon_\theta(x_t, \varnothing)\Big), \tag{14}$$

where $x_t$ is the noisy latent at diffusion step $t$, and $w$ is the guidance scale. Intuitively, guidance amplifies the effect of the conditioning signal during sampling, trading off stronger prompt adherence against reduced diversity or potential artifacts. From this perspective, many guidance-based alignment methods can be viewed as extending this basic idea by replacing or augmenting the guidance term with richer semantic, structural, or physics-aware signals.

**Direct Trajectory Guidance.** Direct trajectory guidance steers video generation by modifying the denoising trajectory during inference, typically through latent or conditioning modulation, while keeping model parameters fixed (Esser et al., 2023; Liu et al., 2025a; Zhou et al., 2025c; Yuan et al., 2026). In practice, these methods differ in where the guidance enters sampling: some intervene through instance-aware or spatially localized guidance, while others reshape the conditioning path itself. InstanceV (Chen et al., 2025i) exemplifies the first case by combining instance-aware conditioning with spatially aware unconditional guidance to preserve instance-level consistency and reduce the distortion or disappearance of small objects during generation. ALG (Choi et al., 2025) illustrates the second case. It modifies the sampling path by

adaptively low-pass filtering the conditioning image in the early denoising stage, preventing the model from prematurely overfitting to static high-frequency appearance details and thereby encouraging more expressive motion. More fine-grained control is achieved by selective latent intervention methods such as Masked Latent Adaptation (Zheng et al., 2025b), which uses learned masks to confine guidance to task-relevant latent regions, enabling targeted alignment of motion or appearance while preserving the pretrained prior.

**Semantic and Structural Steering via Auxiliary Models.** Beyond direct trajectory perturbation, an alternative approach steers video generation using auxiliary models that provide semantic or structural signals at inference. A representative example is CSVC (Spyrou et al., 2025), which performs black-box causal steering without modifying generator parameters or requiring access to internal model mechanisms. Its core idea is to optimize text prompts using a vision-language-model-based objective under an assumed causal graph, so that the edited video is guided toward causally faithful counterfactual variations. This differs from direct trajectory guidance methods such as ALG, which intervene in the denoising path itself, because CSVC shifts the intervention point to external semantic feedback and prompt optimization. SynMotion (Tan et al., 2025a) similarly introduces auxiliary semantic guidance by decomposing textual descriptions into motion-relevant components with an auxiliary model, enabling finer control over motion during sampling. Together, these methods show that auxiliary models can guide generation not only by validating outputs, but also by providing structured semantic objectives that steer inference-time behavior.

## 6.2 Iterative Refinement and Self-editing

Complementing guidance-based steering, inference-time iterative refinement improves generation through feedback-driven updates. By cyclically updating intermediate representations without modifying model parameters, these methods enhance temporal consistency, motion accuracy, and structural coherence. This highlights the possibility of regulating generation behavior through closed-loop refinement alone.

**Inference-Time Iterative Refinement and Self-Correction.** Inference-time iterative refinement employs multi-step feedback loops to progressively revise intermediate representations or generation plans during sampling. By correcting intermediate states without updating model parameters, these methods reduce motion errors, temporal artifacts, and structural inconsistencies. Feedback may operate in latent space or at a higher-level planning stage, enabling refinement of both fine-grained spatiotemporal details and long-horizon behavior. At the latent level, DragVideo (Deng et al., 2024) applies iterative motion supervision on noisy latents to align motion with user-defined point trajectories. The user-provided drag signals are propagated through repeated updates of intermediate latent states during sampling, so alignment is enforced throughout the edited video. This makes the refinement loop explicit and helps the model progressively correct motion states as generation unfolds. DFVEdit (Cai et al., 2025a) instead performs cyclic latent updates for zero-shot video editing, repeatedly refining representations without retraining. Its refinement operates through repeated clean latent and flow transformation updates that improve editing consistency over the course of sampling. FlashI2V (Ge et al., 2025) revisits initialization by gradually shifting the noise distribution during inference, mitigating conditional image leakage and producing smoother motion.

Beyond latent manipulation, self-correction mechanisms introduce recursive feedback at the planning level. MotionAgent (Liao et al., 2025b) adopts an agentic framework with a "rethinking" step to verify motion alignment and iteratively adjust generation plans. It introduces an intermediate motion-planning process in which intended motion can be checked and revised before errors fully propagate through generation. As a result, self-correction operates not only on local denoising states, but also on higher-level motion planning.

**Cascaded and Multi-Stage Refinement.** Cascaded and multi-stage refinement structures inference into successive stages, where early stages establish coarse motion and layout and later stages focus on refining local interactions and visual details (Shen et al., 2025a; Tan et al., 2025b). By separating global dynamics from fine-grained refinement, this design reduces the accumulation of early motion errors that often degrade long and complex video generations (Lin et al., 2025b). iDiT-HOI (Shen et al., 2025c) exemplifies this paradigm with a two-stage diffusion transformer that first captures coarse motion patterns and then refines complex hand–object interactions. The first stage establishes the coarse interaction structure, while the second stage focuses on temporally coherent hand-object interaction dynamics, leading to improved temporal coherence and

physical plausibility. More generally, cascaded refinement architectures assign distinct semantic or temporal roles to different stages, allowing later stages to condition on stabilized intermediate representations, which improves robustness in long-horizon generation. Compared to iterative refinement methods that rely on cyclic feedback and correction, cascaded refinement adopts a feed-forward, stage-wise inference paradigm, trading iterative flexibility for improved stability and more predictable computational cost. Overall, staging inference in this way provides coarse-to-fine control without sacrificing inference-time efficiency.

Table 4: Summary of inference-time methods via post-trained signals for video generation.

| Model | Sub-Category | Base Model | Venue | Year | Link |
|---|---|---|---|---|---|
| Gen-1 (Esser et al., 2023) | Direct Trajectory Guidance | Stable Diffusion | ICCV | 2023 | ⌂ ✗ |
| MotionAgent (Liao et al., 2025b) | Iterative Refinement and Self-Correction | Stable Video Diffusion | ICCV | 2025 | ⌂ ✗ |
| AICL (Liu et al., 2025a) | Direct Trajectory Guidance | VideoCrafter VideoCrafter2 LVDM | ACM MM | 2025 | – ✗ |
| PAHA (Zhou et al., 2025c) | Direct Trajectory Guidance | VLDM | arXiv | 2025 | – ✗ |
| InstanceV (Chen et al., 2025i) | Direct Trajectory Guidance | Wan | arXiv | 2025 | – ✗ |
| ALG (Choi et al., 2025) | Direct Trajectory Guidance | CogVideoX Wan 2.1 HunyuanVideo LTX | arXiv | 2025 | ⌂ ✗ |
| CSVC (Spyrou et al., 2025) | Semantic and Structural Steering | Stable Diffusion | arXiv | 2025 | ⌂ ✗ |
| SynMotion (Tan et al., 2025a) | Semantic and Structural Steering | HunyuanVideo | arXiv | 2025 | – ✗ |
| DiffPhy (Zhang et al., 2025c) | Semantic and Structural Steering | Wan2.1 | arXiv | 2025 | – ✗ |
| DFVEdit (Cai et al., 2025a) | Iterative Refinement and Self-Correction | CogvideoX Wan2.1 | arXiv | 2025 | ⌂ ✗ |
| FlashI2V (Ge et al., 2025) | Iterative Refinement and Self-Correction | Wan2.1 | arXiv | 2025 | ⌂ ✗ |
| iDiT-HOI (Shen et al., 2025c) | Cascaded and Multi-Stage Refinement | Wan FLUX.1-Dev | arXiv | 2025 | – ✗ |
| Raccoon (Tan et al., 2025b) | Cascaded and Multi-Stage Refinement | – | arXiv | 2025 | – ✗ |
| WMReward (Yuan et al., 2026) | Direct Trajectory Guidance | MAGI-1 VLDM | arXiv | 2026 | – ✗ |

# 7 Cross-Family Comparison and Multi-stage Pipelines

**Takeaways**

- Backbone architecture affects how post-training objectives are expressed and implemented, but does not determine the taxonomy itself or make one post-training family inherently tied to a specific architecture.

- Different post-training families are useful under different alignment conditions: supervised tuning fits direct target supervision, self-training and distillation support scalable or efficient improvement, reward-based methods optimize evaluative goals, and inference-time methods provide flexible control without retraining.

- Cross-family combinations are usually realized as sequential multi-stage pipelines, where different stages separately handle adaptation, correction, evaluative refinement, and deployment-oriented compression.

Sections 3–6 organize post-training and alignment methods for video generation into four broad families: (1) supervised fine-tuning, (2) self-training and distillation, (3) preference- and reward-based optimization, and (4) inference-time methods. This taxonomy clarifies how alignment signals are introduced and enforced. At the same time, these families should not be interpreted as strictly competing alternatives. In practice, modern video generation systems often combine several post-training stages, in which different method families play distinct and complementary roles, typically built atop strong contemporary diffusion-based backbones.

### 7.1 How Architecture Shapes Post-training Interfaces

Although our taxonomy is organized by how alignment is enforced, the architectural design of the backbone model still affects how post-training is implemented in practice. Autoregressive video models generate discrete spatio-temporal tokens (Kondratyuk et al., 2024), while diffusion-based video models generate samples by iteratively refining continuous latent states through denoising steps (Ho et al., 2020). This difference affects how alignment methods are formulated. In autoregressive models, reinforcement learning or preference optimization can be defined more naturally over token-level policies (Yu et al., 2025a). In diffusion models, similar objectives are usually implemented over latent trajectories, denoising steps, or sampling-time guidance (Ho & Salimans, 2022; Shi et al., 2024; Tian et al., 2025b). This does not imply that any one architecture is inherently more suitable for post-training. Rather, different backbones make different forms of alignment easier to express and implement.

In the video generation literature covered by this survey, most post-training methods are developed on diffusion-based video generators, especially recent DiT-style models (as shown in Tables 1, 2, and 3), largely because these backbone models dominate the present video generation ecosystem. The pattern is therefore better understood as a consequence of the current backbone landscape, rather than evidence that particular post-training families are tied to diffusion architectures.

### 7.2 When Different Post-training Families Are More Appropriate

The four post-training families differ not only in how alignment is enforced, but also in the types of alignment problems they are best suited to address. Rather than viewing these families as interchangeable alternatives, their effectiveness depends on the available supervision, the target behavior, and the stage at which alignment is applied. We therefore compare them in terms of the settings in which each is most useful, along with their limitations.

**Supervised Fine-tuning Methods.** Supervised fine-tuning is most useful when high-quality paired or structured supervision is available, so that the model can directly learn the target behavior. Typical examples include supervised mappings from prompt and pose sequences to target videos, or from reference identity and motion conditions to personalized video outputs (Wang et al., 2023b; Ma et al., 2024a; Xu et al., 2025d). In such cases, supervised fine-tuning directly refines how the model maps user inputs and control signals to desired outputs. However, it becomes less suitable when the desired behavior cannot be easily specified as a direct target, such as when alignment depends on subtle human preferences, competing objectives, or long-horizon correctness that is easier to evaluate than to annotate.

**Self-training and Distillation Methods.** Self-training and distillation work best when high-quality aligned supervision is scarce, expensive, or noisy. These methods improve the generator by leveraging stronger teacher outputs or self-generated targets, instead of relying on newly curated aligned supervision or direct optimization against explicit evaluative signals (Lin et al., 2025b). They are especially useful when the goal is to make alignment more scalable, improve robustness, or produce a model that is easier to deploy, for example by distilling a large multi-step generator into a few-step or one-step student (Nie et al., 2026). Their limitations arise when the desired behavior cannot be reliably inferred from generated targets or teacher outputs and instead requires explicit evaluative feedback.

**Preference- and Reward-Based Methods.** Preference- and reward-based methods are particularly well suited to cases where the desired behavior is difficult to specify as a direct supervised target, but can still be expressed through comparative or evaluative feedback (Wallace et al., 2024). They are particularly useful when alignment depends on multiple objectives, such as semantic fidelity, temporal coherence, physical plausibility, and identity consistency, that are difficult to encode in a single supervised target (Liu et al., 2025c). In such cases, the generator is refined by optimizing toward signals that indicate which outputs are preferred or better aligned (Yang et al., 2024; Xu et al., 2026). However, their effectiveness depends critically on the quality of the reward or preference signal. If it is noisy, underspecified, or exploitable, optimization may favor narrow proxies or lead to reward-hacking behavior.

**Inference-Time Methods.** Inference-time methods are often most attractive when retraining is impractical, particularly under constraints on compute, data, or deployment. Instead of updating model parameters, they improve alignment by modifying the generation process at inference time, for example, through trajectory steering, constraint enforcement, or iterative refinement (Esser et al., 2023; Yuan et al., 2026). They are particularly useful when alignment must remain flexible at deployment time, such as across different users, prompts, or environments (Lee et al., 2025a; Choi et al., 2025). However, they are most effective when supported by strong pretrained components, such as reward models or guidance mechanisms, and are less reliable in their absence.

### 7.3 Family Intersections and Multi-stage Composition

Although the taxonomy separates post-training methods into distinct families, in practice, these methods can be combined within a single pipeline. The taxonomy, therefore, identifies the mechanism that plays the primary role in driving behavioral change, rather than implying mutually exclusive categories. In the literature covered by this survey, cross-family combinations are most commonly realized sequentially rather than within a single stage. Several recurring patterns emerge:

- **Supervised fine-tuning → preference- and reward-based alignment.** Prompt-A-Video (Ji et al., 2025b) provides a representative example of this pattern. It first uses a reward-guided prompt evolution process to construct improved prompt data and then applies supervised fine-tuning to train the prompt model. A second stage further aligns the model with DPO using pairwise data constructed from multi-dimensional rewards. The design reflects a clear division of labor: supervised fine-tuning establishes a stronger prompt generator, and preference optimization then refines it using comparative feedback that is harder to encode as a single supervised label.

- **Supervised fine-tuning → test-time self-training.** CustomTTT (Bi et al., 2025) illustrates this pattern by first training separate LoRA modules for appearance and motion customization, which is most naturally viewed as supervised fine-tuning. It then introduces a dedicated test-time training stage after LoRA combination, using the trained customized models as guidance to further update parameters and reduce artifacts caused by direct LoRA merging. The motivation is that appearance and motion customization can be learned separately through direct adaptation. However, once these separately learned controls are composed, their interaction may still introduce inconsistencies, which are better addressed through a subsequent correction stage based on self-training.

- **Distillation → preference- and reward-based alignment.** DOLLAR (Ding et al., 2025) provides a clear example of this combination. It first performs few-step video generation through a combination of variational score distillation and consistency distillation. It then applies latent reward model fine-tuning to further improve generation quality under specified reward metrics. The distillation stage improves efficiency by compressing generation into a few steps, while the reward-based stage recovers quality and alignment that may not be fully preserved by distillation alone.

- **Supervised fine-tuning → preference- and reward-based alignment → distillation.** Seedance 1.0 (Gao et al., 2025d) is a representative example of a larger multi-stage pipeline that spans several families. It explicitly describes a post-training sequence consisting of supervised fine-tuning, reinforcement learning with video-specific reward metrics, and multi-stage distillation. These stages play different roles: supervised fine-tuning strengthens the base model under curated supervision, reinforcement learning further improves behavioral alignment under evaluative feedback, and distillation transfers these gains into a faster and more deployable model. This example is especially illustrative because it shows how different families can be combined sequentially, with each stage addressing a different need rather than trying to optimize all objectives within a single stage.

Across these examples, a consistent pattern emerges: different post-training families play complementary roles within a larger system. Supervised stages establish or adapt behavior, self-training and distillation improve scalability and efficiency, and preference- or reward-based methods refine behavior using evaluative signals. By combining these stages sequentially, modern pipelines are able to address multiple aspects of alignment that are difficult to optimize within a single training paradigm.

# 8 Datasets, Benchmarks, and Evaluation Protocols

**Takeaways**

- Post-training and alignment datasets encode alignment objectives explicitly, providing targeted supervision for instruction following, temporal consistency, identity preservation, physical plausibility, and preference modeling beyond large-scale pretraining data.

- Benchmarks for video generation alignment are increasingly organized by alignment dimensions, separating instruction adherence, long-horizon temporal coherence, and physical plausibility to enable more diagnostic and complementary evaluation.

- Evaluation protocols are commonly grouped into three categories: automated metrics, learned evaluators, and human judgments, each serving distinct roles within the evaluation pipeline.

- Benchmark-wise quantitative summaries connect the post-training taxonomy with reported results on public benchmarks, covering both broad evaluation suites and more targeted benchmarks for specific alignment dimensions.

While post-training methods determine how video generation models are optimized, datasets, benchmarks, and evaluation protocols define what it means for a model to be aligned. Datasets encode alignment objectives through structured supervision, benchmarks translate them into concrete evaluation targets, and evaluation protocols specify how aligned behavior is measured. Rather than passive resources, these components shape how post-trained video generation models are developed, diagnosed, and evaluated.

## 8.1 Post-training Datasets

Post-training and alignment of video generation models depend not only on optimization methods, but also critically on the datasets that encode alignment signals. Unlike large-scale pretraining corpora that prioritize coverage and diversity, datasets used for post-training emphasize specific alignment objectives. Accordingly, they can be categorized by the type of alignment signal they provide, including instruction-following supervision, temporal consistency and identity preservation, physics- and reasoning-oriented constraints, and preference signals derived from either synthetic or real videos.

**Instruction-following datasets.** Instruction-following datasets aim to ensure that generated videos accurately reflect user intent expressed through textual descriptions and, when available, structured conditions (Bai et al., 2025a; Wang & Yang, 2025b; Ju et al., 2025). Representative examples include TIP-I2V (Wang & Yang, 2025a), which collects millions of real-world text and image prompts from user interactions, capturing realistic prompt distributions that differ substantially from those of curated captions. Such datasets are particularly valuable for aligning image-to-video models with user intent, as they expose failure modes arising from incomplete, underspecified, or noisy prompts. Beyond purely textual supervision, MMVideo (Xi et al., 2025) pairs text prompts with densely aligned multimodal annotations covering geometry, appearance, and semantics. This form of supervision translates instructions into executable constraints, enabling post-training methods to improve semantic adherence, controllability, and robustness under diverse instruction formulations. Together, these datasets support post-training strategies that improve instruction adherence while maintaining robustness under diverse prompt formulations.

**Temporal consistency and identity datasets.** A second class of datasets targets temporal alignment objectives such as long-range coherence, motion stability, and identity preservation (Yuan et al., 2025; Liu et al., 2025d; Xiao et al., 2025a). Since small frame-level errors can accumulate into perceptual artifacts, these datasets stress-test temporal consistency and penalize such failures. OpenHumanVid (Li et al., 2025c) focuses on human-centric videos requiring consistent appearance and articulation across diverse motions and viewpoints. EgoVid-5M (Wang et al., 2025p) extends this objective to egocentric generation, where first-person camera motion is tightly coupled with action dynamics, exposing overlooked temporal failure modes through kinematic signals and detailed annotations. ViMoGen-228K (Lin et al., 2026) instead emphasizes motion

Table 5: Datasets used for training in video generation post-training and alignment.

| Name | Size | Tasks | Link |
|---|---|---|---|
| ChronoMagic-Pro (Yuan et al., 2024b) | 460,000 | High resolution time-lapse video. | 🤗 |
| SafeSora (Dai et al., 2024) | 57,333 | Human preference text-video pairs for safety and value alignment. | 🤗 |
| CookGen (Xiao et al., 2025a) | 200,000 | Long-form narrative generation in the cooking domain. | 🤗 |
| HOIGen-1M (Liu et al., 2025d) | 1,000,000 | Human-object interaction videos. | 🤗 |
| TIP-I2V (Wang & Yang, 2025a) | 1,700,000 | User-driven text-image prompt dataset for image-to-video generation. | 🤗 |
| SynFMC (Shuai et al., 2025) | 62,000 | Camera-object motion control for video generation. | 🤗 |
| PhyWorld (Kang et al., 2025) | 6,000,000 | Physics-simulated video prediction dataset. | 🤗 |
| OpenS2V-5M (Yuan et al., 2025) | 5,000,000 | High resolution subject-text-video triples. | 🤗 |
| EgoVid-5M (Wang et al., 2025p) | 5,000,000 | Egocentric videos with action annotations. | 🧩 |
| VideoUFO (Wang & Yang, 2025b) | 1,091,712 | User-focused topic-aligned text-video pairs for text-to-video generation. | 🤗 |
| WISA-80K (Wang et al., 2025h) | 79,500 | Physics-aware text-to-video generation. | 🤗 |
| CI-VID (Ju et al., 2025) | 340,000 | Coherent sequence of video clips with text captions. | 🤗 |
| OpenHumanVid (Li et al., 2025c) | 52,300,000 | Human-centric text-video pairs with fine-grained appearance and motion. | – |
| TalkCuts (Chen et al., 2025c) | 164,000 | Multi-shot human speech videos. | – |
| GRADEO-Instruct (Mou et al., 2025) | 3,300 | Human-annotated video-rationale-score triples. | – |
| MMVideo (Xi et al., 2025) | 350,000 | Hybrid real-and-synthetic dataset aligned across modalities and captions. | – |
| Dprim (Sun et al., 2025b) | 32,000 | Primitive-level embodied video prediction for robotic world modeling. | – |
| DAVID-X (Gao et al., 2025c) | 747 | Defect-annotated explainable AI-generated video detection dataset with spatiotemporal evidence and rationales. | – |
| PairFS-4K (Chen et al., 2025d) | 4,000 | Two-person figure skating video dataset. | – |
| PNData (Bai et al., 2025a) | 296,960 | Prompt-random-noise-refined-noise triples. | – |

diversity and generalization while maintaining temporal stability. Together, these datasets provide alignment supervision for post-training methods aimed at reducing temporal drift while preserving motion realism and identity consistency.

**Physics and reasoning datasets.** Beyond perceptual coherence, an emerging class of datasets targets physical plausibility and causal consistency, reflecting the growing interest in video generation models as world simulators. These datasets encode alignment objectives that extend beyond appearance and motion, emphasizing whether generated videos adhere to basic physical laws, object interactions, and cause-and-effect relationships. Datasets such as WISA-80K (Wang et al., 2025h) introduce physics-aware supervision by constructing videos that reflect structured world dynamics, which helps post-training methods better align generated outputs with physical constraints. In more embodied, domain-specific scenarios, datasets such as Dprim (Sun et al., 2025b) go a step further by linking video generation to action-conditioned world transitions. In these settings, physical consistency is evaluated alongside downstream tasks such as robotics. These datasets play a crucial role in supporting reinforcement learning and preference-based alignment methods that rely on verifiable, rule-based signals rather than purely subjective judgments.

**Synthetic versus real preference datasets.** Finally, a distinct class of datasets provides preference-based alignment signals by contrasting synthetic and real videos, often with explicit failure annotations. Rather than prescribing how videos should be generated, they define undesirable or unacceptable outcomes, making them useful for alignment diagnosis, evaluation, and preference optimization. SafeSora (Dai et al., 2024) collects human preference annotations focused on safety and value alignment in text-to-video generation. DAVID-X (Gao et al., 2025c) instead pairs AI-generated and real videos with fine-grained spatio-temporal defect labels and natural language rationales. Although rarely used to directly train generators, these datasets provide valuable signals for post-training strategies, reward modeling, and evaluation. By annotating identity inconsistencies, motion anomalies, and physical implausibility, they help align automated objectives with human judgment.

## 8.2 Benchmarks by Alignment Dimensions

Unlike datasets, which mainly specify the source and structure of supervision, benchmarks translate alignment goals into concrete evaluation targets. In video generation, benchmarks are increasingly designed around specific alignment dimensions rather than comprehensive quality assessment. As a result, they tend to isolate particular aspects of aligned behavior, such as instruction adherence, temporal coherence, or physical plausibility. This dimension-oriented perspective clarifies how different benchmarks capture complementary aspects of alignment and enables more meaningful comparisons across methods.

**Instruction-following and controllability benchmarks.** Benchmarks in this category evaluate semantic correctness and controllability, assessing whether generated videos follow instructions about subjects, actions, scene setup, and audio outputs (Han et al., 2025; Chen et al., 2025g; Yu et al., 2025b; Pham et al., 2025; Ling et al., 2025; Sun et al., 2025a; Shi et al., 2025a). Such benchmarks are crucial for post-training evaluation, as instruction-following failures often persist despite high visual fidelity. OpenS2V-Eval (Yuan et al., 2025), for example, measures subject-to-video generation by testing whether identity and specified attributes are preserved. Domain-structured benchmarks like RecipeGen (Zhang et al., 2025h) and CineTechBench (Wang et al., 2025q) assess procedural and cinematic instruction execution, while TAVGBench (Mao et al., 2024) extends evaluation to multimodal settings by jointly considering audio and video outputs. Together, these benchmarks establish instruction adherence as a distinct alignment dimension for analyzing how effectively models translate intent into controlled generation.

**Temporal consistency and identity preservation benchmarks.** Temporal alignment benchmarks evaluate whether video generation models preserve coherent structure over long durations. Rather than prioritizing immediate semantic accuracy, these benchmarks emphasize temporal consistency and examine whether models maintain stable dynamics across frames, shots, or narrative segments. This perspective is reflected in benchmarks that target different forms of long-horizon coherence. ChronoMagic-Bench (Yuan et al., 2024b) evaluates text-to-time-lapse generation under strong physical priors, such as biological growth or physical transformations. It measures metamorphic amplitude and temporal coherence, rather than appearance stability alone. For multi-character interaction settings, DanceTogether (Chen et al., 2025d) introduces TogetherVideoBench. This benchmark specifically evaluates identity-action binding, assessing the model's ability to maintain distinct identities during complex, extended interactions. Together, these benchmarks highlight long-horizon temporal alignment as a multi-faceted objective, spanning physical progression, cinematic continuity, interaction stability, and narrative coherence.

**Physical plausibility and world-model benchmarks.** Physical plausibility benchmarks target alignment objectives that extend beyond perceptual coherence. Rather than asking whether a video looks consistent over time, these benchmarks assess whether the depicted dynamics match real-world expectations (Feng et al., 2025b; Bansal et al., 2025; Zhang et al., 2025l). This line of work is motivated by viewing video generation models as implicit world simulators. Representative benchmarks in this category are often grounded in task-oriented or embodied settings. WorldSimBench (Qin et al., 2025) evaluates whether generated videos support world simulation. It combines human feedback with downstream video-to-action or agent-centric evaluations to test whether the dynamics are actionable and physically meaningful. In a similar spirit, Drive&Gen (Wang et al., 2025f) evaluates physical plausibility through domain-specific tasks such as autonomous driving. In these

settings, violations of physical consistency directly degrade downstream performance. Unlike preference-based or purely perceptual benchmarks, these evaluations rely on structured criteria and verifiable outcomes. This makes them particularly compatible with reinforcement learning and verification-driven alignment methods.

### 8.3 Evaluation Protocols and Metrics

While benchmarks define which aspects of alignment are evaluated, evaluation protocols and metrics determine how those aspects are measured and compared. In video generation, evaluation typically draws on multiple sources of evidence, ranging from automated metrics to learned evaluators and human judgments. Each approach rests on different assumptions about perceptual quality, semantic correctness, and temporal coherence. As a result, no single evaluation protocol is sufficient to cover all alignment dimensions.

**Automated metrics.** Automated metrics remain central to video generation evaluation, providing scalable and reproducible assessment. Early methods borrow image and compression metrics such as FID (Heusel et al., 2017), SSIM and PSNR (Wang et al., 2004), and LPIPS (Zhang et al., 2018), which measure frame-level visual similarity or reconstruction quality. While effective for low-level fidelity, they fail to capture semantic correctness and temporal coherence, often ignoring cross-frame dynamics. Video-specific metrics such as FVD (Unterthiner et al., 2018) address this limitation by evaluating distributions of video features rather than individual frames, improving sensitivity to motion and temporal artifacts. However, FVD remains focused on visual realism and does not explicitly assess alignment with conditioning signals. With the rise of text-conditioned generation, later metrics incorporate vision-language representations (Liu et al., 2024b; Sharan et al., 2025; Quignon et al., 2025) to measure text–video correspondence at the clip level. VBench (Huang et al., 2024) and VBench2 (Zheng et al., 2025a) exemplify this pipeline-based approach, combining perceptual similarity, vision-language alignment, and motion-sensitive features into a unified evaluation framework. In practice, automated metrics are rarely used alone; instead, they are integrated into standardized pipelines to capture complementary aspects of quality.

More recent methods such as VideoScore (He et al., 2024) and VideoScore2 (He et al., 2025c) introduce learned scoring models trained to approximate human judgment across perceptual and semantic dimensions. Unlike hand-crafted metrics, they aggregate heterogeneous cues into a single signal, though they are primarily used for evaluation rather than optimization. Task-specific benchmarks further propose domain-aligned indicators—for example, ChronoMagic-Bench (Yuan et al., 2024b) introduces MTScore and CHScore to measure metamorphic amplitude and long-range temporal coherence—demonstrating how specialized metrics can supplement general-purpose evaluation.

**Learned evaluators.** Beyond fixed metrics and aggregation-based protocols, recent work increasingly adopts learned evaluators to approximate human judgment in video generation. These evaluators differ in supervision sources, outputs, and inference mechanisms but share a common goal: capturing alignment properties difficult to express with hand-crafted similarity measures. Existing methods broadly fall into two categories: reward-style evaluators that produce scalar scores (Qin et al., 2025; Wu et al., 2025b), and LLM-based evaluators that provide explicit reasoning or generative feedback. Representative reward-style evaluators include AnimeReward (Zhu et al., 2025) and VideoReward (Liu et al., 2025c), trained on human preference data to produce scalar scores reflecting perceptual quality and alignment. By aggregating heterogeneous visual and semantic cues into a single signal, they enable scalable evaluation more correlated with human judgment than traditional metrics. Although originally designed for optimization and post-training, such reward models are widely reused as evaluators due to their simplicity and effectiveness. However, their assessments remain largely opaque, as alignment is reduced to numerical scores without explicit reasoning.

More recent work leverages MLLMs as evaluators, treating evaluation as reasoning or generation rather than pure scoring (Wu et al., 2025e; Wang et al., 2025g; Bansal et al., 2025; Li et al., 2025d). ETVA (Guan et al., 2025) exemplifies a reasoning-based evaluator, assessing text–video alignment through question-driven evaluation to verify semantic attributes such as object existence, relations, and physical consistency beyond similarity metrics. In contrast, AIGVE-MACS (Liu & Zhang, 2025) adopts a generative approach, producing aspect-wise scores alongside natural language feedback and framing evaluation as structured generation. This improves interpretability and enables more diagnostic analysis of alignment quality.

**Human evaluation and hybrid protocols.** Despite advances in automated metrics and learned evaluators, human evaluation remains the reference standard for assessing alignment in video generation. This is especially true for visual realism, semantic precision, and overall preference. Common human evaluation protocols include absolute rating, pairwise comparison, and ranking-based judgments (Liu et al., 2023d; 2024b). In addition, recent work explores more efficient methods for scaling human feedback. Arena-style frameworks, such as K-Sort Arena (Li et al., 2025k), improve the robustness of preference-based benchmarking through structured comparison and probabilistic ranking. Importantly, these methods do not rely on automated evaluators. In practice, evaluation pipelines increasingly adopt hybrid protocols. Automated metrics and learned evaluators are used for large-scale screening and diagnostic analysis, while human evaluation is reserved for validation and final comparison. This hybrid paradigm balances scalability with reliability and reflects current best practices for evaluating aligned video generation models.

Table 6: Representative benchmarks used for video generation post-training and alignment evaluation.

| Name | Size | Tasks | Link |
|---|---|---|---|
| **FETV (Liu et al., 2023d)** | 618 | Fine-grained and temporal-aware evaluation of text-to-video generation. | 🤗 |
| **StoryBench (Bugliarello et al., 2023)** | 6,000 | Story-driven text-to-video generation evaluation | – |
| **VBench (Huang et al., 2024)** | – | Multi-dimensional video generation evaluation. | 🔺 |
| **ChronoMagic-Bench (Yuan et al., 2024b)** | 1,649 | Time-lapse T2V generation; temporal coherence and metamorphic change evaluation. | 🤗 |
| **EvalCrafter (Liu et al., 2024b)** | 700 | Text-to-video generation across diverse prompt types and multi-dimensional quality criteria. | 🤗 |
| **TAVGBench (Mao et al., 2024)** | 1,700,000 | Text to Audible-Video Generation. | 🤗 |
| **T2VSafetyBench (Miao et al., 2024)** | 4,400 | Text-to-video model safety assessment. | – |
| **MTBench (Shi et al., 2025b)** | 100 | Motion transfer task evaluation. | 🤗 |
| **FiVE (Li et al., 2025d)** | 100 | Fine-grained text-guided video editing evaluation. | 🤗 |
| **StoryEval (Wang et al., 2025t)** | 423 | Story-level multi-event text-to-video generation evaluation. | 🔺 |
| **MJ-BENCH-VIDEO (Tong et al., 2025)** | 10,842 | Fine-grained video preference evaluation. | 🤗 |
| **OpenS2V-Eval (Yuan et al., 2025)** | 180 | Subject-consistent video generation. | 🤗 |
| **Doc2Present (Shi et al., 2025a)** | 30 | Document-to-presentation video generation. | 🤗 |
| **VideoPhy (Bansal et al., 2025)** | 688 | Physical commonsense for real-world activities assessment. | 🤗 |
| **T2V-CompBench (Sun et al., 2025a)** | 700 | Compositional text-to-video generation. | 🤗 |
| **VEG-Bench (Yu et al., 2025b)** | 132 | Instructional video editing. | 🤗 |
| **VMBench (Ling et al., 2025)** | 1,050 | Human perception-aligned motion evaluation. | 🤗 |
| **VidCapBench (Chen et al., 2025g)** | 643 | Text-to-video generation video caption evaluation. | 🤗 |
| **VideoGen-RewardBench (Liu et al., 2025c)** | 26,500 | Annotated prompt-video pairs for reward model evaluation. | 🤗 |
| **Verse-Bench (Wang et al., 2025b)** | 600 | Joint audio-video generation evaluation. | 🤗 |
| **AIGC-LipSync (Peng et al., 2025)** | 615 | Audio-driven video lip synchronization evaluation. | 🤗 |
| **DisenStudioBench (Chen et al., 2024b)** | 1,500 | Customized multi-subject text-to-video generation. | – |
| **TC-Bench (Feng et al., 2025b)** | 270 | Temporal Compositionality of video generation assessment. | – |
| **HVEval (Wu et al., 2025c)** | 20,000 | Human-centric videos generation. | – |
| **PhyGenBench (Meng et al., 2025b)** | 160 | Evaluate physical commonsense correctness in text-to-video generation. | – |
| **Video-Bench (Han et al., 2025)** | 419 | Human-aligned video generation. | – |
| **AIGVQA-DB (Wang et al., 2025g)** | 36,576 | Text-to-video model capability assessment. | – |
| **ETVABench (Guan et al., 2025)** | 2,000 | Textvideo alignment evaluation. | – |

### 8.4 Benchmark-wise Quantitative Comparison of Post-training Methods

To complement the benchmark and evaluation protocol discussion above, we summarize quantitative results reported by representative post-training and alignment methods on selected public benchmarks. The goal of this comparison is not to establish a unified leaderboard, but to show how different post-training families are evaluated in practice and how their reported gains correspond to different alignment dimensions. In video generation, post-training, alignment, and inference-time adaptation methods often differ in task formulation, base model, model size, sampling budget, resolution, video length, and optimization objective (Li et al., 2024; Liu et al., 2025h; Yin et al., 2025; Jeong et al., 2025). Even when papers report results under the same benchmark name, they may use different prompt sets, submetrics, or evaluation subsets. Therefore, the numbers in this section should be interpreted as benchmark-wise evidence under heterogeneous protocols rather than as a universal ranking of post-training categories.

Tables 7 and 8 summarize reported results on VBench (Huang et al., 2024) and VBench2 (Zheng et al., 2025a), which cover broad evaluation dimensions such as visual quality, temporal consistency, subject consistency, controllability, commonsense, human fidelity, and physical plausibility. Table 9 further focuses on physics-oriented evaluation through VideoPhy (Bansal et al., 2025) and VideoPhy2 (Bansal et al., 2026). Only methods with publicly available reported results on the selected dimensions are included, and missing entries indicate scores that are not reported in the corresponding papers. Together, these tables provide a concrete view of how the proposed taxonomy connects to commonly used evaluation dimensions.

Table 7: **Reported quantitative results of representative post-training methods on VBench.** TF: Temporal Flickering, AQ: Aesthetic Quality, SC: Subject Consistency, IQ: Imaging Quality, MS: Motion Smoothness, BC: Background Consistency, DD: Dynamic Degree. Bold values indicate the best reported score within each post-training family. "-" indicates that the corresponding information is not reported in the original paper.

| Method | Backbone | | VBench | | | | | | |
|---|---|---|---|---|---|---|---|---|---|
| | Model | Size | TF | AQ | SC | IQ | MS | BC | DD |
| *Supervised Fine-Tuning Methods* | | | | | | | | | |
| **ReCapture (Zhang et al., 2025a)** | Stable Video Diffusion | – | 91.1 | 57.4 | 88.5 | 64.8 | 98.2 | 92.0 | 49.0 |
| **Phantom (Liu et al., 2025e)** | MMDiT | – | – | 58.0 | – | 70.6 | **99.3** | – | – |
| **Follow-Your-Creation (Ma et al., 2025e)** | Wan2.1 | – | 88.2 | – | 90.3 | – | 92.4 | 89.3 | – |
| **MinT (Wu et al., 2025g)** | OpenSora | – | – | 54.4 | 90.0 | 60.9 | 98.8 | 95.0 | **71.1** |
| **MoAlign (Bhowmik et al., 2026)** | CogVideoX | 2B | 99.0 | 64.5 | **95.8** | 64.5 | 98.4 | 96.4 | 42.2 |
| **3DreamBooth (Ko et al., 2026)** | HunyuanVideo-1.5 | 8B | – | 52.5 | – | **73.3** | 99.3 | – | – |
| **LTD (Wu et al., 2026b)** | Wan2.1 | – | **99.7** | **65.8** | 95.7 | 67.9 | 98.4 | **97.2** | 68.1 |
| *Self-Training and Distillation Methods* | | | | | | | | | |
| **Reangle-A-Video (Jeong et al., 2025)** | CogVideoX | 5B | 93.9 | 52.4 | 91.4 | 62.7 | 97.9 | 93.6 | 88.8 |
| **NFD (Cheng et al., 2025e)** | – | – | – | – | 86.1 | **68.4** | – | – | **99.5** |
| **MoGAN (Xue et al., 2025)** | Wan2.1 | 1.3B | – | 59.0 | – | 68.0 | **98.6** | – | 96.0 |
| **Neodragon (Karnewar et al., 2026)** | Pyramidal Flow DiT | – | **99.3** | 60.7 | – | 59.8 | – | – | – |
| **V.I.P. (Kim et al., 2025a)** | VideoCrafter2 | – | 98.1 | **62.9** | 96.8 | 67.6 | 97.9 | **97.7** | 45.8 |
| *Preference- and Reward-Based Methods* | | | | | | | | | |
| **VideoDPO (Liu et al., 2025h)** | CogVideoX | 2B | – | 58.6 | 94.7 | – | 88.6 | 96.6 | 38.9 |
| **PRFL (Mi et al., 2025)** | Wan2.1 | 14B | – | – | 95.5 | – | 98.1 | – | 84.7 |
| **Prompt-A-Video (Ji et al., 2025b)** | CogVideoX | 5B | – | 63.9 | 95.3 | 68.7 | 98.3 | 95.9 | 54.0 |
| **PhysCorr (Wang et al., 2025j)** | Wan2.1 | 14B | 99.4 | 62.0 | 96.8 | 67.3 | 97.2 | 97.5 | **94.8** |
| **T2V-Turbo (Li et al., 2024)** | VideoCrafter2 | – | 97.5 | 63.0 | 96.3 | **72.5** | 97.3 | 97.0 | 49.2 |
| **OnlineVPO (Zhang et al., 2026a)** | VideoCrafter2 | – | 97.5 | 62.3 | **98.0** | 68.9 | 98.9 | 98.2 | 47.0 |
| **Self-paced GRPO (Li et al., 2025g)** | Wan2.1 | 14B | 99.1 | **65.3** | 96.5 | 68.0 | 98.4 | **98.4** | 52.8 |
| **PhysRVG (Zhang et al., 2026b)** | Wan2.2 | 5B | **99.6** | 41.4 | 97.0 | 65.0 | **99.6** | 97.7 | 52.0 |
| *Inference-Time Methods* | | | | | | | | | |
| **MotionAgent (Liao et al., 2025b)** | Stable Video Diffusion | – | 97.5 | **64.5** | 96.1 | – | 98.9 | 96.8 | 16.7 |
| **ALG (Choi et al., 2025)** | Wan2.2 | 14B | **97.8** | 63.5 | 95.8 | **70.0** | 98.7 | – | – |
| **SynMotion (Tan et al., 2025a)** | HunyuanVideo | 13B | – | – | **98.3** | 69.5 | **99.5** | **97.6** | **88.2** |

Table 8: **Reported quantitative results of representative post-training methods on VBench2.** Bold values indicate the best reported score within each post-training family. "-" indicates that the corresponding information is not reported in the original paper.

| Method | Backbone | | VBench2 | | | | |
|---|---|---|---|---|---|---|---|
| | Model | Size | Creativity | Commonsense | Controllability | Human Fidelity | Physics |
| *Supervised Fine-Tuning Methods* | | | | | | | |
| **MoAlign (Bhowmik et al., 2026)** | CogVideoX | 2B | 52.8 | 65.5 | 25.7 | 86.7 | 48.8 |
| *Preference- and Reward-Based Methods* | | | | | | | |
| **Euphonium (Zhong et al., 2026)** | HunyuanVideo | 13B | **41.4** | **67.2** | 26.9 | **88.9** | 46.8 |
| **PhysCorr (Wang et al., 2025j)** | Wan2.1 | 14B | 29.6 | 42.3 | **35.4** | 85.1 | **49.1** |

Table 9: **Reported quantitative results on physical-plausibility benchmarks.** Bold values indicate the best reported score within each post-training family. "-" indicates that the corresponding information is not reported in the original paper.

| Method | Backbone | | VideoPhy | | VideoPhy2 | |
|---|---|---|---|---|---|---|
| | Model | Size | Semantic Adherence | Physics Consistency | Semantic Adherence | Physics Consistency |
| *Supervised Fine-Tuning Methods* | | | | | | |
| **MoAlign (Bhowmik et al., 2026)** | CogVideoX | 2B | 49.3 | 39.4 | **28.8** | **75.0** |
| **VideoREPA (Zhang et al., 2025i)** | CogVideoX | 5B | **72.1** | **40.1** | 21.0 | 72.5 |
| **WISA (Wang et al., 2025h)** | CogVideoX | 5B | 67.0 | 38.0 | – | – |
| *Preference- and Reward-Based Methods* | | | | | | |
| **PhysMaster (Ji et al., 2025a)** | DiT | – | 67.0 | 40.0 | – | – |
| *Inference-Time Methods* | | | | | | |
| **WMReward (Yuan et al., 2026)** | vLDM | 5B | 53.5 | 34.3 | – | – |

**Taxonomy-aware interpretation of benchmark patterns.** The reported results should be interpreted as dimension-specific evidence rather than as a single measure of overall model quality. Benchmarks such as VBench and VBench2 decompose video generation performance into different dimensions, including temporal consistency, visual fidelity, subject and background consistency, controllability, and motion strength. This decomposition is important for post-training analysis because these dimensions do not always improve together. For example, the VBench results show that high temporal consistency or motion smoothness scores do not necessarily imply strong motion generation: models with similarly smooth temporal evolution can still have substantially different Dynamic Degree scores. These observations highlight a central challenge in video alignment: post-training methods often optimize specific behavioral dimensions, and gains in one aspect of alignment may expose or even amplify weaknesses in another. Accordingly, the comparison provides a diagnostic perspective of how different post-training objectives shape model behavior across heterogeneous evaluation dimensions.

The benchmark patterns are also consistent with the proposed taxonomy. *Supervised fine-tuning methods* mainly achieve implicit alignment by refining the conditional mapping from prompts, reference images, trajectories, poses, domains, or other forms of structured supervision to generated videos (Wu et al., 2025g; Ko et al., 2026). Their reported results are therefore most informative for dimensions that closely correspond to the available supervision, such as subject consistency, background consistency, controllability, and domain-specific fidelity. However, supervised objectives alone are less well suited to optimizing alignment properties that are easier to evaluate than to annotate, such as subtle human preference, long-horizon physical correctness, or safety-sensitive failure avoidance. *Self-training and distillation methods* play a different role. They often transfer teacher behavior, improve trajectory consistency, increase robustness, or reduce inference cost (Xue et al., 2025; Kim et al., 2025a). Accordingly, their practical value should be interpreted alongside efficiency factors such as sampling steps, latency, and deployment costs, which are not fully captured by perceptual benchmark scores alone.

*Preference- and reward-based methods* correspond most directly to explicit alignment because they optimize evaluative signals that assess whether generated videos satisfy desired objectives (Li et al., 2024; Zhang et al., 2026a;b). Their benchmark results are especially relevant when the evaluation dimension matches the reward or preference signal, such as text-video alignment, human preference, dynamic motion, identity consistency, or physical plausibility. At the same time, variation across non-targeted dimensions highlights an important limitation: optimizing one reward can improve the measured objective while leaving other properties unchanged or even degraded. *Inference-time methods* provide a complementary mechanism by steering generation during sampling without updating model parameters (Choi et al., 2025; Yuan et al., 2026). Their reported performance is best understood as evidence of deployment time flexibility, but such methods also require evaluation of guidance conflicts, dependence on external critics, and sensitivity to prompt-specific sampling choices.

These patterns also explain why targeted benchmarks are particularly important for post-training and alignment methods. Unlike foundation video models (Wan et al., 2025; Kong et al., 2024), which are often evaluated as general-purpose systems and expected to improve broadly across many capabilities, many post-training methods are designed to optimize a specific alignment objective or solve a specific downstream task (Ma et al., 2024a; Xu et al., 2025d; Deng et al., 2024; Yin et al., 2025). A method may target physical plausibility, identity preservation, camera control, human motion, safety, personalization, or inference efficiency, without necessarily claiming uniform gains across all aspects of generation quality. Therefore, broad benchmarks such as VBench and VBench2 provide useful overall diagnostics, but they are not sufficient to determine whether a method succeeds at its intended alignment goal. VideoPhy and VideoPhy2 provide one example of this need in the context of physical plausibility: by separating semantic adherence from physics consistency, they can reveal cases where a video follows the prompt but violates physical constraints, or where physically plausible motion comes at the cost of weaker semantic alignment. Beyond physical plausibility, the emergence of targeted benchmarks for safety, compositional instruction following, and temporal or metamorphic coherence shows that objective-specific evaluation is becoming increasingly prevalent across alignment dimensions (Miao et al., 2024; Sun et al., 2025a; Yuan et al., 2024b).

## 9 Challenges and Future Directions

> **Takeaways**
>
> - Supervised fine-tuning is limited by tightly coupled spatial and temporal representations and weak long-horizon reasoning, making it difficult to jointly scale identity preservation, motion adaptation, and multi-stage instruction alignment.
>
> - Self-training and distillation risk error accumulation and trajectory mismatch, requiring verification and temporally aware objectives for robustness.
>
> - Preference-based and reinforcement learning suffer from coarse reward design and unstable optimization, often leading to conservative motion and limited scalability.
>
> - Inference-time alignment suffers from limited generality, guidance conflicts, and computational overhead, motivating more structured, reasoning-driven control.
>
> - Evaluation and benchmarking lack systematic cross-paradigm comparison and fail to thoroughly analyze objective-induced trade-offs and dynamic safety failures.

Despite rapid progress in post-training and alignment techniques for video generation models, significant challenges remain before these systems can achieve robust, controllable, and trustworthy deployment. Building on the taxonomy presented in Sections 3–8, we outline key open problems and promising research directions across supervised fine-tuning, preference learning and reinforcement learning, self-training and distillation, inference-time alignment, as well as cross-cutting challenges in evaluation, benchmarking, and safety.

### 9.1 Supervised Fine-tuning

**Decoupled Appearance and Motion Representation.** Supervised fine-tuning often exposes a tight coupling between appearance and motion representations in video generation models. Adapting models to new motion patterns can unintentionally alter identity and visual consistency (Zhang et al., 2025m; Xu et al., 2025c). To mitigate this issue, existing work seeks to decouple motion dynamics from appearance, typically via disentangled representations or specialized conditioning mechanisms (Wu et al., 2024; Wang et al., 2025m). However, these approaches reveal an inherent trade-off between preserving identity and learning new motion patterns. In modern video generators, spatiotemporal latent structures are highly shared. As a result, motion, texture, and identity remain tightly coupled during optimization, making full disentanglement difficult in practice. Future research may explore dual-branch or multi-stream fine-tuning methods that separate appearance and motion pathways.

**Long-horizon and Multi-stage Instruction Alignment.** Current supervised fine-tuning methods are typically optimized for short video clips of a few to tens of seconds. Consequently, models often experience challenges with following long-horizon or multi-stage instructions (Lin et al., 2024b). Over time, movements start to drift, details get blurry, and the background shifts unnaturally (Li et al., 2026a; Zhang et al., 2025f). This makes the video inconsistent and causes it to ignore the original prompt. Prior work has explored hierarchical planning or long-context tuning to extend temporal coherence (Long et al., 2024; Soni et al., 2025; Guo et al., 2025b). However, these methods are often computationally expensive and struggle with complex causal reasoning and long-term temporal dependencies. One possible direction is to use MLLMs as high-level planners that translate user intent into structured generation plans. In parallel, more efficient generation mechanisms are needed to scale to long videos. For instance, dynamic token routing could help reduce redundancy during long-context generation.

### 9.2 Self-training and Distillation

**Model Collapse and Error Accumulation in Self-training.** Self-training on model-generated videos carries a risk of model collapse. Repeatedly training on self-produced data can reduce diversity and reinforce existing biases. In long video generation, this problem becomes more severe. Small errors introduced early in the video can accumulate over time and gradually dominate later frames (Li et al., 2026a). A promising direction is verifier-guided self-evolution, where self-training is paired with an explicit verification stage (Liao et al., 2025b; Soni et al., 2025). In this setting, generated videos in earlier steps are first evaluated by a verifier, such as a critic or world model, and only verified or corrected samples are reused for training (Lin et al., 2025c; Zhang et al., 2025c). This closed-loop design refines model behavior while limiting error accumulation.

**Temporal Consistency Loss in Distillation.** Distilling diffusion-based video generators from hundreds of denoising steps to only a few inference steps often harms temporal smoothness and motion continuity. This typically leads to flickering artifacts or abrupt motion changes. Recent studies suggest that this issue arises because many distillation methods rely on distribution-level matching and overlook the temporal trajectories of the generation process (Sun et al., 2025c; Zheng et al., 2026). One promising direction is trajectory-aware adversarial distillation, which combines adversarial supervision on temporal coherence with trajectory- or flow-based matching objectives. Beyond accelerating inference, future distillation methods should aim to teach the student model a temporally consistent velocity field that stays aligned with the teacher's generation dynamics (Zhang et al., 2024b).

### 9.3 Preference-based and Reinforcement Learning

**Reward Limitations for Motion and Physical Plausibility.** Preference-based and reinforcement learning methods reveal systematic limitations in reward design for video generation. Existing reward formulations tend to favor visually sharp and temporally stable outputs, implicitly encouraging conservative generation behaviors that suppress motion dynamics in video generation models (Liu et al., 2025h). In addition, most reward signals operate at the video level and rely on holistic preference judgments, which limit their sensitivity to localized temporal failures (Wu et al., 2025f). As a result, subtle but important physical

errors, such as sliding motion or object interpenetration, are often missed, even though they greatly reduce physical plausibility (Meng et al., 2025b; Ji et al., 2025a; Wang et al., 2025j; Le et al., 2025). A promising direction is to incorporate physics-grounded reward signals that complement perceptual rewards (Wang et al., 2025j). These rewards can be derived from auxiliary physical constraints or verification signals. In parallel, supervision can move from holistic video-level labels to finer segment-level rewards (Chen et al., 2025a). This shift enables more localized credit assignment during optimization. With finer-grained rewards, optimization algorithms can penalize a specific time step rather than the entire generated video.

**Inefficiency and Instability in Video Reinforcement Learning.** Preference-based and reinforcement learning methods also face fundamental challenges in sample efficiency and training stability when applied to video generation. Video generation is inherently expensive. Constructing paired samples for preference optimization or performing online sampling for reinforcement learning quickly becomes prohibitive at scale (Wang et al., 2025c). In addition, video trajectories are long and high-dimensional. This often leads to high optimization variance, resulting in unstable training or convergence to overly conservative solutions (Li et al., 2026b; 2025g). Together, these issues make it difficult to directly apply standard reinforcement learning pipelines to video generation at scale. One promising direction is to improve the efficiency of reinforcement learning, for example, by reusing previously generated samples via off-policy optimization or by operating directly in the latent space to reduce decoding cost (Liu et al., 2025c).

## 9.4 Inference-Time Alignment

**Limited Generality of Gradient-based Inference Guidance.** Most inference-time guidance methods steer video generation using classifier-based gradients. This usually requires training task-specific evaluators, which limits flexibility. It also makes it hard to incorporate high-level semantic reasoning or physical constraints. Some recent work explores gradient-free alternatives, such as using VLMs for semantic steering or external rule checkers for physical filtering (Spyrou et al., 2025; Zhang et al., 2025c). However, these approaches are often limited to predefined constraints or simple rejection strategies. A more promising direction is agent-like control with powerful VLMs that monitor intermediate states and intervene during generation, adjusting prompts, attention, or motion plans in real time to enforce semantic and physical consistency without task-specific fine-tuning (Singh et al., 2025; Team et al., 2023).

**Guidance Conflicts and Computational Overhead.** Inference-time alignment methods often combine multiple forms of guidance, including text, object-level cues, and physical constraints. In practice, however, these signals can conflict with one another, which may lead to unstable generation or even collapsed outputs (Bi et al., 2025; Choi et al., 2025). In addition, many inference-time control strategies depend on iterative latent updates or repeated sampling, substantially increasing inference latency and limiting their use in interactive or real-time settings (Deng et al., 2024; Chen et al., 2025i). One potential direction is test-time search and planning, which approaches alignment through explicit lookahead or structured exploration instead of local gradient-based guidance. Rather than directly optimizing latent variables, future systems could first generate high-level plans or keyframes, assess their semantic and physical validity, and then selectively refine intermediate frames via backtracking or branching search (Soni et al., 2025). Such a strategy may offer more stable multi-objective alignment while keeping computational costs under control.

## 9.5 Evaluation and Benchmarking Challenges

**Cross-Paradigm Comparisons and Design Principles.** While post-training and alignment methods have shown promise, their relative strengths remain poorly understood. Most studies evaluate these paradigms in isolation, making it difficult to derive general design principles for video alignment (Wu et al., 2023; Li et al., 2026b; Gao et al., 2025d; Li et al., 2026a). Although recent benchmarks such as Video-Bench (Han et al., 2025), VMBench (Ling et al., 2025), and VideoGen-RewardBench (Liu et al., 2025c) provide useful common evaluation dimensions, they are rarely used for controlled cross-paradigm comparisons across supervised, self-training, preference-based, reinforcement learning, and inference-time approaches. A key future direction is therefore to move from heterogeneous reported results toward standardized comparison protocols that can evaluate different post-training families under matched conditions. Such protocols would not only support

more meaningful comparisons within clearly defined task settings but also reveal which alignment strategy is most effective for a given objective. These analyses would enable more principled choices among alignment strategies and help establish practical design principles for reliable and interpretable video-generation models.

**Failure Modes and Trade-offs.** Beyond benchmarking, post-training methods for video alignment introduce systematic trade-offs that are often insufficiently analyzed. Objectives that strongly penalize temporal inconsistency can lead models to minimize motion, producing overly static videos (Liu et al., 2025h; Wu et al., 2025f). Similarly, identity-preservation losses that tightly constrain appearance across frames can limit compositional generalization, hindering novel interactions or scene changes (Zhang et al., 2025m; Wang et al., 2025m; Meng et al., 2025c). Preference- and reward-based optimization brings additional risks. Optimizing a learned reward can bias generation toward narrow reward-aligned patterns, reducing diversity and sometimes leading to reward exploitation, where perceptual quality improves while long-term coherence degrades (Wang et al., 2025c; Chen et al., 2025a; Wang et al., 2025j). These are not isolated failures but recurring behaviors induced by the objectives themselves, including over-regularized motion, gradual temporal drift in long-horizon videos, and overspecialization to proxy metrics (Lin et al., 2024b; Gao et al., 2025b). Progress in video alignment therefore requires not only reporting aggregate gains but also systematically analyzing failure cases and regressions, especially outside controlled benchmarks.

**Safety-Aware Evaluation and Deployment Safeguards.** Safe deployment of video generation models requires more than addressing failures introduced during optimization. It also requires explicit safeguards against harmful use and misleading synthetic content. Recent benchmarks show that text-to-video models have safety risks that are not fully captured by standard quality evaluation alone Miao et al. (2024). Existing work in this area is still limited, especially for video-specific models, but it can be roughly grouped into a few directions. First, some methods aim to reduce unsafe generation directly. Many of these methods are training-free and work by blocking unsafe concepts employing multimodal risk detection (Ma et al., 2025a) or guiding the generation process during inference (Yoon et al., 2025; Xu et al., 2025a; Facchiano et al., 2026). A smaller number of works instead use post-training to adapt the generator itself for safer generation Ye et al. (2025a). Second, watermarking and source-tracing methods provide a way to identify AI-generated content (Su et al., 2025; Hu et al., 2025b;d). While traditional post-processing watermarks can degrade visual quality, recent advances focus on distortion-free, in-generation watermarking that embeds tracking keys directly into the diffusion noise space, preserving both temporal robustness and scalability (Shi et al., 2026). These can also be integrated into the generator through post-training Huang et al. (2025d). Third, deepfake detection remains important, but it is currently studied mostly as a separate downstream task rather than as post-training of the generator itself Pei et al. (2024). To bridge the gap with model alignment, newer forensic frameworks are incorporating explainable AI to identify specific spatiotemporal defects and provide natural language rationales for generated artifacts Gao et al. (2025c), which could serve as verifiable feedback for future post-training. Overall, these gaps suggest that video generation still lacks a strong body of work on video-specific post-training methods for safety, moderation, and control.

## 10 Conclusion

This survey reviews the emergence of post-training as a new trend in video generation, where the focus has gradually shifted from pure pre-training to targeted alignment and optimization. By combining supervised fine-tuning, preference- and reward-based methods, self-training, and inference-time control, recent approaches have improved controllability, temporal coherence, and alignment with user intent. Despite this progress, several fundamental challenges remain. Key issues include the tight coupling between appearance and motion during post-training, limitations in long-horizon and multi-stage instruction-following, reward and preference signals that fail to capture motion and physical plausibility, and the inefficiency and instability of preference-based and reinforcement learning on high-dimensional video trajectories. Future research will likely depend on more efficient optimization algorithms, stronger grounding signals, and a tighter integration between training-time alignment and inference-time computation. Progress along these directions will be crucial for building more robust and general-purpose video intelligence.

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
