# OpenReview forum: "Video Generation Models: A Survey of Post-Training and Alignment"
_TMLR — Accepted by TMLR_

### Review · Reviewer_N3LC · 2026-03-29

**Summary Of Contributions:**

This survey conducts a comprehensive review of post-training and alignment techniques for video generation models. It covers a wide range of techniques organized in four categories: (1) supervised fine-tuning methods, (2) self-training and distillation methods, (3) preference- and reward-based methods, and (4) inference-time methods. This survey also includes different dataset and benchmark, and common evaluation metrics used in the post training of video generation models.

Strengths:
1. Video generation models are growing rapidly and progressing constantly. Post training and alignment is a very important aspect, especially for academia where large scale pretraining is sometimes challenging. And this survey fills a genuine gap by listing a wide variety of modern techniques used in post training and alignment. Hence it is very valuable to the research community.
2. This survey is well structured and presented in a clear and logical manner. It groups techniques into four categories and each sections have easy-to-read takeaways. And it also includes very useful tables 1-6 listing 1) different methods' base models, publication time and links, 2) each datasets' sizes and tasks, etc. which makes it very easy for researchers to find relevant datasets and baseline methods.
3. This survey also lists some biggest challenges and research directions in the field, which is also useful.

Weaknesses:
1. While this survey is comprehensive and useful, it feels a bit like a catalogue and it would be better if it could include more discussions and insights like on when certain techniques should be preferred over others.
2. Lack of discussion on reward hacking: if using RL on some reward models, it is very common for the model to have reward hacking, especially for some simple metrics like motion score. It would be better to properly discuss this and how to mitigate reward hacking.
3. Lack of discussion on prompt enhancer / rewriter: arguably this should fall under inference-time methods. It's not fancy but is actually a standard but critical component in SOTA video generation models.
4. Lack of discussion on safety mechanism: it's not discussed in this survey how to add safety mechanisms to prevent harmful use of video generation models, e.g. safety classifier / filter, (invisible) watermarking and deepfake detection.

**Audience:**

Yes

**Audience Explanation:**

Researchers on video generation models will find this survey useful as it lists a wide variety of methods, datasets and evaluation metrics.

**Broader Impact Concerns:**

No. It's a survey and doesn't have ethical concerns.

**Claims And Evidence:**

Yes

**Claims Explanation:**

This is a survey which conducts a comprehensive review of post-training and alignment techniques for video generation models. As far as I know, it is indeed the first survey on this topic.

**Requested Changes:**

1. (good to have, not critical) Add more insights on when certain techniques should be preferred over others, e.g. depending on data volume, goal of the post training etc.
2. Add some discussion on reward hacking and how to mitigate (e.g. multi dimension in reward, finer granularity on segment level instead of video level, etc.)
3. Add some discussion on prompt enhancer / rewriter: what's the purpose and how is it trained.
4. Add some discussion on safety mechanism (see weaknesses) which is also an indispensable piece if the model gets released to the public.
5. Explain more clearly what does guidance mean in "Guidance-based Alignment": e.g. add some formula to explain the basic classifier free guidance which is a standard practice in almost every SOTA models
6. (good to have, not critical) It would be better to also include recent trends on audio-video joint generation (not just audio conditioned video generation) and how to post train such models.
7. (good to have, not critical) The survey lists this paper "Video diffusion models" (Ho et al. 2022) under "Among recent architectures, Diffusion Transformers (DiTs) (Peebles & Xie, 2023) have emerged as the dominant paradigm". But technically that paper is diffusion with U-Net but not Diffusion Transformer. And actually diffusion with U-Net is an intermediate state / period between the transformation from GAN to DiT.

---

> ### Author Response · Authors · 2026-04-19
> **Reply to the Reviewer**
>
> We thank the reviewer for their thoughtful and encouraging feedback, and for recognizing the survey's clear structure, comprehensive coverage of post training and alignment techniques for video generation, and its value to the research community.
>
> > Add more insights on when certain techniques should be preferred over others, e.g. depending on data volume, goal of the post training etc.
>
> We have added a discussion section (Section 7) to address this point. It provides practical guidance on when different post-training approaches are preferred, based on factors such as data availability, supervision level, alignment objectives, and deployment constraints. We also highlight key trade-offs and limitations across methods to support informed selection.
>
> > Add some discussion on reward hacking and how to mitigate (e.g. multi dimension in reward, finer granularity on segment level instead of video level, etc.)
>
> We expanded Section 5.4 to more explicitly discuss reward hacking and its implications. We highlight how coarse video-level rewards can encourage shortcut behaviors and overlook temporal inconsistencies, and we outline mitigation strategies through more structured supervision, including multi-dimensional reward design and finer-grained signals such as segment-level or hierarchical feedback.
>
>
> > Add some discussion on prompt enhancer / rewriter: what's the purpose and how is it trained.
>
> We agree that prompt enhancers/rewriters are commonly used in video generation pipelines and can play an important practical role. Prompt rewriting is usually an input preprocessing step performed at inference time by a separate language model, rather than a component that operates on the video generation model or its sampling dynamics. Since our survey focuses on post-training and inference-time methods that directly operate on the video generation model or its sampling process (e.g., model parameters or denoising dynamics), we do not treat prompt rewriting as a primary method family. For methods in which the prompt generator is integrated into training jointly with the video generator (e.g., [1]), these are already covered in our paper.
>
> [1] Video-as-Answer: Predict and Generate Next Video Event with Joint-GRPO. CVPR'26.
>
> > Add some discussion on safety mechanism (see weaknesses) which is also an indispensable piece if the model gets released to the public.
>
> We added a discussion in Section 9.5 on safety-aware evaluation and deployment safeguards. We emphasize that safe public release requires not only improving generation quality, but also incorporating explicit safety mechanisms, including unsafe-content mitigation, watermarking and provenance tracking, and deepfake detection. We further note that these aspects remain underexplored in current post-training work for video generation, particularly from a video-specific alignment perspective.
>
> > Explain more clearly what does guidance mean in "Guidance-based Alignment": e.g. add some formula to explain the basic classifier free guidance which is a standard practice in almost every SOTA models.
>
> We clarified the concept of guidance in Section 6.1. In particular, we added the standard formulation of classifier-free guidance as a representative example, and used it to illustrate how guidance-based methods steer the denoising trajectory at inference time without modifying model parameters.
>
> > It would be better to also include recent trends on audio-video joint generation (not just audio conditioned video generation) and how to post train such models.
>
> Thanks for the suggestion. We added a paragraph in Section 3.3 to cover this setting in more detail. It discusses recent trends in joint audiovisual synthesis, focusing on cross-modal alignment challenges, including temporal synchronization, semantic consistency, and unimodal quality preservation. It also highlights post-training strategies such as curriculum-based optimization (e.g., Apollo), where joint generation capability and synchronization quality are progressively improved through aligned audio–video training and constrained multi-task optimization.
>
> > The survey lists this paper "Video diffusion models" (Ho et al. 2022) under "Among recent architectures, Diffusion Transformers (DiTs) (Peebles & Xie, 2023) have emerged as the dominant paradigm". But technically that paper is diffusion with U-Net but not Diffusion Transformer. And actually diffusion with U-Net is an intermediate state / period between the transformation from GAN to DiT.
>
> Thanks for pointing this out. We revised the corresponding sentence in the Introduction to distinguish earlier diffusion/U-Net-based models from the later rise of DiT-based architectures.

---

> ### Author Response · Authors · 2026-04-27
>
> Dear Reviewer N3LC,
>
> Thank you again for your helpful feedback! We have submitted our response and revised manuscript addressing your comments accordingly. If you have additional suggestions or concerns to further improve our work, please don't hesitate to let us know.
>
> Thank you very much for your time and consideration!
>
> Sincerely,
>
> Authors

---

### Review · Reviewer_Q6WD · 2026-03-30

**Summary Of Contributions:**

Modern generation models typically undergo both pre-training and post-training stages. The pre-training stage trains the model with large-scale unlabelled dataset or corpus, so that the model can develop useful representations and generation capability; while post-training fine-tunes the model to adhere to human instructions and execute specific tasks effectively. This paper presents a comprehensive survey on post-training and alignment techniques for video generation models. In the established taxonomy, the authors categorized existing post-training methods into (1) supervised fine-tuning methods, (2) self-training and distillation methods, (3) preference- and reward-based methods, and (4) inference-time methods. Together, these categories span the entire spectrum of video model refinement. Beyond algorithmic design, this survey also explores critical technical dimensions, including architectural innovations, datasets, benchmarks, and evaluation protocols. By doing so, it provides a clear overview of current advancements for practitioners while establishing a rigorous foundation for researchers entering the field.

**Audience:**

Yes

**Audience Explanation:**

I think this paper will likely be of significant interest to researchers working on video generation, controlaable generation, and alignment. Besides, practitioners who seeks to fine-tune open-source video models such as Wan, CogVideoX, Cosmos-Predict2 to adapt for multiple downstream tasks (such as robotics) can refer to this paper for specific guidance. This survey also inspires people to design novel benchmarks and datasets for video generation.

**Broader Impact Concerns:**

No concerns. The paper has adequately discussed the relevant broader impact and ethical considerations.

**Claims And Evidence:**

Yes

**Claims Explanation:**

The major claim from this paper is the developed taxonomy of post-training methods for video generation models. The proposed categories comprehensively encompass the full spectrum of fine-tuning techniques currently utilized in video generation tasks, providing a structured and exhaustive overview of the field.

**Requested Changes:**

This is a well-structured, comprehensive survey that fills an important gap in the video generation litrerature. I would recommend the authors to add another section to discuss the differences and relationships between this survey and exsiting surveys on video generation, such as [1]. Besides, the paper would benefit from a more nuanced discussion on how different architectural designs (e.g., DiTs vs. ViTs vs. U-Nets) and generation paradigms (e.g., auto-regressive vs. bidirectional/diffusion-based) influence the selection and efficacy of post-training techniques.

[1]: Ma, Yue, et al. "Controllable video generation: A survey." arXiv preprint arXiv:2507.16869 (2025).

---

> ### Author Response · Authors · 2026-04-19
> **Reply to the Reviewer**
>
> We thank the reviewer for their encouraging comments and for recognizing the survey's well-structured taxonomy, its comprehensive treatment of post-training and alignment techniques for video generation, and its broad coverage of key technical dimensions including architectures, datasets, benchmarks, and evaluation protocols.
>
> > I would recommend the authors to add another section to discuss the differences and relationships between this survey and exsiting surveys on video generation, such as [1] (Ma, Yue, et al. "Controllable video generation: A survey." arXiv preprint arXiv:2507.16869 (2025)).
>
> We added a "Relationship to Existing Surveys" paragraph at the end of Section 1 to discuss the differences and connections between our survey and recent related surveys on video generation, including [1] on controllable generation, [2] and [3] on video diffusion models, and [4] on human video generation. While these surveys primarily focus on model architectures, conditioning strategies, or domain-specific applications, our survey instead organizes methods from the perspective of post-training and alignment, i.e., how alignment signals are incorporated into video generation models. We believe this perspective complements existing surveys and provides a clearer focus on post-training and alignment techniques.
>
> [1] Controllable video generation: A survey. arXiv'25.
>
> [2] Survey of video diffusion models: Foundations, implementations, and applications. TMLR'25.
>
> [3] A survey on video diffusion models. ACM Computing Surveys'24.
>
> [4] Comprehensive survey on human video generation: Challenges, methods, and insights. arXiv'24.
>
>
> > The paper would benefit from a more nuanced discussion on how different architectural designs (e.g., DiTs vs. ViTs vs. U-Nets) and generation paradigms (e.g., auto-regressive vs. bidirectional/diffusion-based) influence the selection and efficacy of post-training techniques.
>
> We added a discussion in Section 7.1 on how architectural design and generation paradigms affect post-training interfaces. We clarify that autoregressive models naturally support token-level optimization, whereas diffusion-based models typically express alignment objectives over denoising trajectories, latent variables, or sampling-time guidance. We also summarize that most existing post-training methods are currently built on diffusion-based generators, particularly DiT-based backbones, reflecting their dominance in today's video generation landscape rather than any inherent limitation of post-training techniques.

---

> ### Author Response · Authors · 2026-04-27
>
> Dear Reviewer Q6WD,
>
> Thank you again for your helpful feedback! We have submitted our response and revised manuscript addressing your comments accordingly. If you have additional suggestions or concerns to further improve our work, please don't hesitate to let us know.
>
> Thank you very much for your time and consideration!
>
> Sincerely,
>
> Authors

---

### Review · Reviewer_5LeY · 2026-04-07

**Summary Of Contributions:**

This paper presents the first comprehensive survey of post-training methodologies for video generation models. It categorizes the current literature into five distinct domains: Supervised Fine-Tuning (SFT), Self-Training and Knowledge Distillation, Preference and Reward-based Models, Inference-Time Methods, and Datasets/Benchmarks. The submission establishes a high-level taxonomy and maps existing research to key challenges within the field.

**Audience:**

No

**Audience Explanation:**

***For a review paper, I interpret "audience interest" as the value a reader gains from the synthesis and taxonomy, allowing them to grasp the field's current state and future directions without needing to read every underlying paper. Based on this interpretation, my current answer is no.***

**Rationale:** TMLR's audience, particularly junior scholars and practitioners seeking a quick entry point into post-training video generation, will find the proposed taxonomy highly valuable. The five-part categorization effectively organizes a rapidly expanding subfield. However, to fully serve this audience, the paper must articulate clear, actionable takeaways. Readers should learn the comparative advantages of these methods directly from the text, rather than using the survey merely as an annotated bibliography.

### Revision 1:
In the revision this issue has been addressed. There is more emphasize on explaining the different methods, their advantages, and key takeaways.

**Broader Impact Concerns:**

I foresee no broader impact concerns.

**Claims And Evidence:**

No

**Claims Explanation:**

***For a review paper, I interpret "accurate and clear evidence" as providing self-contained, faithful representations of the literature, and "convincing evidence" as presenting a robust, logical categorization of the field. Based on that interpretation my current answer is no (for the initial draft).***

**Rationale**: While the paper provides a convincing and well-structured categorization, it lacks the necessary self-containment to be fully accurate and clear.

1. Most prominent is the inconsistent technical depth: Section 5.1 details mathematical objectives for reinforcement learning, whereas Sections 3 and 4 omit objective functions entirely.

1. Maybe even more crucially is that the overview is very shallow. The conceptual description of most cited works is limited to a single sentence. And the mathematical description does not detail all the used notations. That makes the review heavy on name, notation and terminology dropping. I'd rather see even more emphasis on synthesizing methods, describing one or two key papers in more detail and then comparing other works briefly to these. The current manuscript is forcing the reader to consult the original papers to understand core mechanisms.

1. Minor but in the same category: the core claims in the introduction regarding current "mismatches" and "alignment strategies designed for video models" lack empirical backing, citations, or systematic follow-up across the subsequent sections.

### Revision 1:
In the revision the technical depth is increased, by adding objective functions and more descriptive explanation of some of the key works. While the review remains heavy on name/terminology dropping, it now also serves as good baseline overview of the field. Hence, I find the paper now to be `accurate, convincing and clear`.

**Requested Changes:**

If some of the major weaknesses are addressed, this paper is a contribution to the field. So, while currently my review is negative I feel positive that this could be (rather easily) adressed by the authors to make this paper *convincing* and of *broad interest*.

## Crucial and substantial change in the technical depth is required:
1. Ensure a deeper level of mathematical and conceptual detail across all sections. For example, add more details to the RL mathematics in Section 5 and introduce explicit objective functions in Sections 3 and 4 to avoid terminology dropping without context.
1. Expand descriptions of cited works beyond single-sentence summaries. Define and unpack ambiguous terminology (e.g., by explaining a few works in more detail, and contrasting other works to these) to ensure the survey is reasonably self-contained.

## Minor rephrasing to enhance clarity:
3.  Provide citations for the assertions regarding "objective mismatches" and "alignment strategies," and continuously map the categories back to these challenges throughout the paper.
4.  Clarify methodology intersections: The review defines 5 categories, do all papers nicely fit into a single category, or are there papers combining these? And if yes, how: stage 1 in category A and stage 2 in category B? Or are there ways to combine Category A&B in a single stage? If there are, please explicitly identify instances where different post-training methodologies are combined in practice!

## Minor suggestions for Tables 1-6
5.  Revise tables 1-4: Remove the "GPU" columns, as they offer minimal comparative value. Replace them with columns detailing methodology sub-categories or other comparative values. Clarify what the '-' is standing for and how the entries are sorted.
6.  [Tiny]: In table 2 I see SVD and Stable Video Diffusion models, I bet they correspond to the same base model.
7.  Expand dataset statistics in table 5 & 6: Augment the general "size" metric with specific statistics, including frame count, resolution, and types of conditional signals.

---

> ### Author Response · Authors · 2026-04-19
> **Reply to the Reviewer (1)**
>
> We thank the reviewer for their thoughtful comments and for recognizing the paper's convincing and well-structured categorization, as well as the value of its taxonomy for organizing this rapidly growing area.
>
> > Ensure a deeper level of mathematical and conceptual detail across all sections. For example, add more details to the RL mathematics in Section 5 and introduce explicit objective functions in Sections 3 and 4 to avoid terminology dropping without context.
>
> We added introductory subsections to Sections 3 and 4 (i.e., Sections 3.1 and 4.1) to present a unified optimization view of supervised fine-tuning, self-training, and distillation, including explicit objective functions and mathematical formulations. This aligns their presentation more closely with Section 5. We also refined Section 5.1 to improve notation consistency and expanded the RL discussion with additional mathematical detail for greater conceptual clarity.
>
> > Expand descriptions of cited works beyond single-sentence summaries. Define and unpack ambiguous terminology (e.g., by explaining a few works in more detail, and contrasting other works to these) to ensure the survey is reasonably self-contained.
>
> In the revision, we strengthened Sections 3–6 using an anchor-example style. Specifically, we now describe representative works in greater depth, covering their core mechanisms and contributions, and then relate other methods as extensions, alternatives, or complementary approaches. We also expanded previously brief descriptions to clarify key aspects such as supervision signals, control interfaces, and optimization roles. While not all works can be described in equal depth due to space constraints, we believe the revision substantially improves clarity and makes the survey more self-contained.
>
> > Provide citations for the assertions regarding "objective mismatches" and "alignment strategies," and continuously map the categories back to these challenges throughout the paper.
>
> Thank you for this helpful suggestion. In the revision, we strengthened the motivation in the Introduction by adding citations supporting both failure modes and the underlying objective mismatch, including VBench [1], VideoPhy [2], InstructVideo [3], SafeSora [4], and VBench-2 [5]. We also added citations (e.g., Atzmon et al. [6], RDPO [7], Improving Video Generation with Human Feedback [8], and OnlineVPO [9]) to support challenges such as competing objectives and limited high-quality supervision, as well as the need for video-specific alignment strategies.
>
> In addition, we refined the opening of Sections 3–6 to more explicitly connect each method family to these motivating challenges, clarifying how supervised fine-tuning, self-training and distillation, preference- and reward-based methods, and inference-time methods each address different aspects of the objective mismatch introduced in Section 1.
>
> [1] Vbench: Comprehensive benchmark suite for video generative models. CVPR'24.
>
> [2] VideoPhy: Evaluating Physical Commonsense for Video Generation. ICLR'25.
>
> [3] InstructVideo: Instructing Video Diffusion Models with Human Feedback. CVPR'24.
>
> [4] Safesora: Towards safety alignment of text2video generation via a human preference dataset. NeurIPS'24.
>
> [5] Vbench-2.0: Advancing video generation benchmark suite for intrinsic faithfulness. arXiv'25.
>
> [6] Identity-Motion Trade-offs in Text-to-Video Generation. BMVC'25.
>
> [7] RDPO: Real Data Preference Optimization for Physics Consistency Video Generation. arXiv'25.
>
> [8] Improving Video Generation with Human Feedback. NeurIPS'25.
>
> [9] Onlinevpo: Align video diffusion model with online video-centric preference optimization. WACV'26.
>
> > Clarify methodology intersections: The review defines 5 categories, do all papers nicely fit into a single category, or are there papers combining these? And if yes, how: stage 1 in category A and stage 2 in category B? Or are there ways to combine Category A&B in a single stage? If there are, please explicitly identify instances where different post-training methodologies are combined in practice!
>
> We first clarify that our taxonomy defines four post-training families rather than five. In the revision, we added Section 7 to explicitly address both the distinctions between method families and how they are combined in practice. In particular, Section 7.2 compares the settings in which each family is most appropriate, along with their strengths and limitations, while Section 7.3 discusses common cross-family integration patterns. We show that such combinations do exist in current pipelines, but are most commonly realized as sequential multi-stage pipelines rather than being jointly optimized within a single stage.

---

> > ### Author Response · Authors · 2026-04-19
> > **Reply to the Reviewer (2)**
> >
> > > Revise tables 1-4: Remove the "GPU" columns, as they offer minimal comparative value. Replace them with columns detailing methodology sub-categories or other comparative values. Clarify what the '-' is standing for and how the entries are sorted.
> >
> > We revised the tables accordingly. We merged the GPU model and number of GPUs columns into a single GPU column for compactness, and added a Sub-category column to align each method with our taxonomy. We also clarified that entries are sorted chronologically by publication date, and that "-" indicates information that is not reported or not clearly specified in the original papers.
> >
> > While we agree that GPU information alone is not a strong comparative metric, we retain it in a compact form as it still provides useful context regarding the approximate computational requirements of different methods.
> >
> > > In table 2 I see SVD and Stable Video Diffusion models, I bet they correspond to the same base model.
> >
> > Thank you for pointing this out. "SVD" and "Stable Video Diffusion" indeed refer to the same base model here. We have unified the notation to "Stable Video Diffusion" for consistency.
> >
> > > Expand dataset statistics in table 5 & 6: Augment the general "size" metric with specific statistics, including frame count, resolution, and types of conditional signals.
> >
> > We agree that richer dataset statistics would be valuable when consistently available. However, we found that most papers do not report key attributes such as resolution, frame count, and conditional signal types in a sufficiently consistent way across datasets and benchmarks. As a result, these attributes cannot be reliably standardized for inclusion in Tables 5 and 6. We therefore use the number of videos or video-prompt pairs as the primary size metric, since it is the most consistently reported statistic across the literature.

---

> ### Comment · Reviewer_5LeY · 2026-04-21
> **Reply to the Authors**
>
> Thanks for the revision and the responses. In the revision much of my concerns have been addressed. I'm happy to switch from no/no to yes/yes.

---

> > ### Author Response · Authors · 2026-04-21
> >
> > We are glad that your concerns were resolved in the revision, and we sincerely appreciate your time and thoughtful feedback throughout the review process. Thank you for your support.

---

### Decision · Action_Editor_rYu7 · 2026-05-28

**Recommendation:** Accept with minor revision

**Additional Comments:**

One limitation that should be addressed in the revision is that, although the paper summarizes existing benchmarks and evaluation protocols, it does not explicitly compare the concrete empirical performance differences among existing post-training methods. A more detailed benchmark-wise and dimension-wise comparison of quantitative results would help clarify the relative advantages and limitations of the proposed categories. This would also strengthen the connection between the taxonomy, open challenges, and future directions discussed in the paper. Such a quantitative comparison would make the survey more informative and practically useful for readers seeking to understand the current state and trends of post-training and alignment methods in video generation.

**Audience:**

Yes

**Audience Explanation:**

This paper would be of clear interest to researchers working on video generation, particularly those who are new to the field. By providing an organized overview that includes recent post-training methods, the paper helps readers quickly grasp the current landscape of video generation and its post-training researches and understand how different alignment approaches are positioned.

**Claims And Evidence:**

Yes

**Claims Explanation:**

This paper offers a comprehensive survey of video generation models, focusing specifically on post-training and alignment methods. In specific, rather than organizing existing methods primarily by architecture or conditioning signal, the paper categorizes them according to how alignment is enforced and proposes a structured taxonomy along this dimension. In addition, it summarizes existing datasets, benchmarks, and evaluation protocols for post-training and alignment in video generation, while also discussing open challenges and future research directions.

Overall, as a survey paper, this paper’s main claims are well supported. In particular, the claim that this is the first survey dedicated to post-training approaches for video generation is justified through a clear comparison with existing survey papers. The proposed four-category taxonomy is also presented in a logical and consistent manner, providing a coherent framework for organizing prior post-training methods. In addition, the paper adequately summarizes existing benchmarks and discusses remaining challenges, further supporting its contribution as a timely and useful survey.